# Optical generative models

Shiqi Chen[1,2,3], Yuhang Li[1,2,3], Yuntian Wang[1,2,3], Hanlong Chen[1,2,3] & Aydogan Ozcan[1,2,3 ✉]

Generative models cover various application areas, including image and video synthesis, natural language processing and molecular design, among many others[1–11]. As digital generative models become larger, scalable inference in a fast and energy-efficient manner becomes a challenge[12–14]. Here we present optical generative models inspired by diffusion models[4], where a shallow and fast digital encoder first maps random noise into phase patterns that serve as optical generative seeds for a desired data distribution; a jointly trained free-space-based reconfigurable decoder all-optically processes these generative seeds to create images never seen before following the target data distribution. Except for the illumination power and the random seed generation through a shallow encoder, these optical generative models do not consume computing power during the synthesis of the images. We report the optical generation of monochrome and multicolour images of handwritten digits, fashion products, butterflies, human faces and artworks, following the data distributions of MNIST[15], Fashion-MNIST[16], Butterflies-100[17], Celeb-A datasets[18], and Van Gogh's paintings and drawings[19], respectively, achieving an overall performance comparable to digital neural-network-based generative models. To experimentally demonstrate optical generative models, we used visible light to generate images of handwritten digits and fashion products. In addition, we generated Van Gogh-style artworks using both monochrome and multiwavelength illumination. These optical generative models might pave the way for energy-efficient and scalable inference tasks, further exploiting the potentials of optics and photonics for artificial-intelligence-generated content.

Generative digital models have recently evolved to create diverse, high-quality synthetic images[1–4], human-like natural language processing capabilities[5], new music pieces[6] and even new protein designs[7]. These emerging generative artificial intelligence (AI) technologies are critical for various applications ranging from large language models (LLMs)[5,11] to embodied intelligence[8] and AI-generated content[9,10]. With their success, these models are getting larger, demanding substantial amounts of power, memory and longer inference times[11]. The scalability and carbon footprint of generative AI models are also becoming a growing concern[13,14]. While several emerging approaches[20–39] have aimed to reduce the size and power consumption of such models, also improving their inference speed, there is still an urgent need to develop alternative approaches for designing and implementing power-efficient and scalable generative AI models.

Here we demonstrate optical generative models that can optically synthesize monochrome or colour images that follow a desired data distribution—that is, optically generating images that have never been reported before for a given distribution. Inspired by diffusion models[4], this concept uses a shallow digital encoder to rapidly transform random two-dimensional (2D) Gaussian noise patterns into 2D phase structures that represent optical generative seeds. This optical seed generation is a one-time effort that involves a shallow and fast phase-space encoder acting on random 2D noise patterns. The snapshot optical generation of each image or output data following

a desired distribution occurs on demand by randomly accessing one of these pre-calculated optical generative seeds. This broad concept can be implemented by different optical hardware, for example, integrated photonics or free-space-based implementations (Extended Data Fig. 1). Without loss of generality, here we report free-space-based reconfigurable optical generative models (Fig. 1 and Extended Data Fig. 1b). Each one of the optical generative seeds, once presented on a spatial light modulator (SLM) and illuminated by a plane wave, synthesizes an image through a reconfigurable diffractive decoder optimized for a given data distribution; the refresh rate is limited by the frame rate of the SLM that displays the pre-calculated optical generative seeds. The optical part of the computation required for snapshot image generation is carried out entirely through free-space propagation of light via an optimized and fixed (that is, static) diffractive decoder. We report image generation performance that is statistically comparable to digital neural-network-based generative models, which was confirmed through the generation of monochrome and multicolour images of handwritten digits, fashion products, butterflies, human faces and Van Gogh-style artworks, following the distributions of the Modified National Institute of Standards and Technology (MNIST)[15], Fashion-MNIST[16], Butterflies-100[17], Celeb-A datasets[18] and Van Gogh's paintings[19], respectively. To experimentally demonstrate snapshot and multicolour optical generative models, we built free-space hardware operating in the visible spectrum.

[1]Electrical and Computer Engineering Department, University of California Los Angeles, Los Angeles, CA, USA. [2]Bioengineering Department, University of California Los Angeles, Los Angeles, CA, USA. [3]California NanoSystems Institute (CNSI), University of California Los Angeles, Los Angeles, CA, USA. ✉e-mail: ozcan@ucla.edu

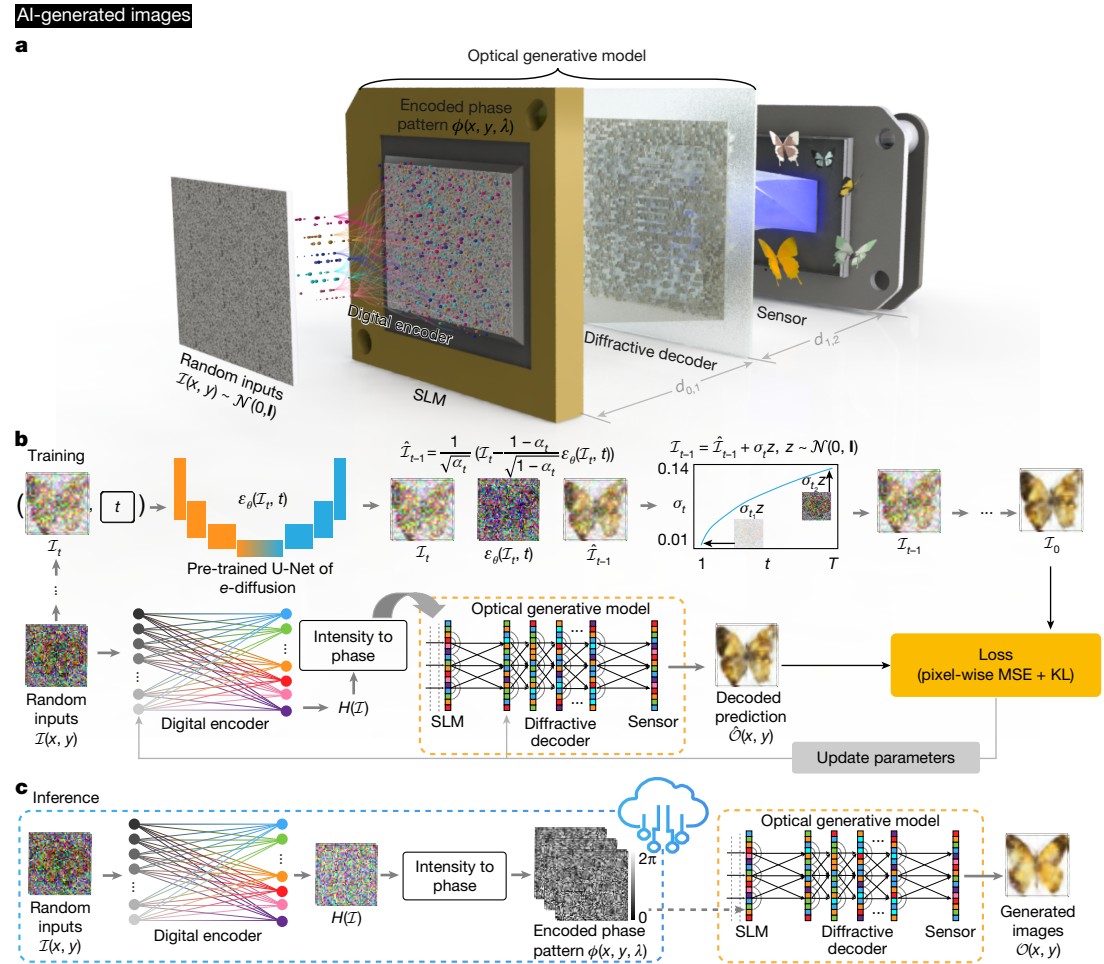

**Fig. 1 | Design of a snapshot optical generative model.** Images in panels **a**, **b** and **c** are AI-generated. **a**, Scheme of a snapshot optical generative model. Random Gaussian-noise-based inputs are first encoded by a shallow digital encoder, which creates numerous optical generative seeds that are randomly accessed by an SLM. After the input optical field propagates through a reconfigurable and optimized diffractive decoder, the generated images are recorded on a sensor array. For a given target data distribution, the generative optical model can synthesize countless images. The optical propagation of the input light generating the output image through the diffractive decoder takes <1 ns; however, the overall speed of image generation is limited by the input SLM refresh time. **b**, The snapshot optical generative model is trained by a learned DDPM, where the data pairs generated by the DDPM are used to guide the optimization of the snapshot optical generative model. **c**, For blind inference of images, the pre-calculated optical generative seeds are randomly accessed through, for example, a cloud-based server, where the snapshot image generation is locally realized by free-space optics and wave propagation. $d_{0,1}$, the distance between the SLM and the diffractive decoder; $d_{1,2}$, the distance between the diffractive decoder and the sensor; $\varepsilon_\theta$, the noise prediction model, where $\theta$ represents the parameters of the model; $\alpha_t$, a time-dependent noise scheduling coefficient in the diffusion process; $z$, a random variable sampled from the normal distribution; $\sigma_t$, the standard deviation of the noise added at the time step $t$; $H$, output of the shallow digital encoder; MSE, mean square error; KL, Kullback–Leibler divergence; $\hat{\mathcal{I}}$, intermediate denoising results; $\mathcal{O}$, generated intensity.

Our experimental results confirmed that the learned optical generative models successfully grasped the underlying features and relationships within each target data distribution.

The presented framework is highly flexible, as different generative models targeting different data distributions share the same optical architecture with an optimized diffractive decoder that is fixed or static for each task, synthesizing countless images using optical generative seeds phase-encoded from random noise. Therefore, the desired data distribution can be switched from one generative task to another by changing the optical generative seeds and the corresponding reconfigurable decoder surface, both of which can be performed without a change to the optical set-up. The energy efficiency, scalability and flexibility of optical generative models will stimulate further research and development, providing promising solutions for various AI-related applications, including, for example, AI-generated content, image and video processing and synthesis, among others[40].

## Snapshot image generation

Figure 1 shows a schematic of our monochrome snapshot image generative model. As depicted in Fig. 1a, random 2D inputs, each following the normal distribution, are encoded into 2D phase patterns by a digital encoder, which rapidly extracts the latent features and encodes them into the phase channel for subsequent analogue processing. These random-noise-generated phase-encoded inputs serve as our optical generative seeds and are loaded onto an SLM to feed information to our diffractive optical generative model. Under coherent illumination, the optical fields carrying these encoded phase patterns propagate and are processed by a diffractive decoder that is optimized for a given target data distribution. Finally, the generated images are captured by an image sensor, representing images following the target data distribution. The training procedure is shown in Fig. 1b, where we first train a teacher digital generative model based on the denoising diffusion probabilistic model (DDPM) to learn the target data distribution[4].

Once trained, the learned DDPM is frozen, and it continuously generates the noise–image data pairs used to train the snapshot optical generative model. The shallow digital phase encoder and the optical generative model are jointly trained, enabling the model to efficiently learn the target distribution with a simple and reconfigurable architecture. Figure 1c presents our blind inference procedure: the encoded phase patterns (that is, the optical seeds) generated by the digital encoder from random noise patterns are pre-calculated, and the optical generative model decodes these generative phase seeds through free space, using a fixed or static decoder. For rapid synthesis of optical generative phase seeds from random Gaussian noise, the digital encoder comprises three fully connected layers, where the first two are followed by a nonlinear activation function (Methods). The reconfigurable diffractive decoder is optimized with, for example, $400 \times 400$ learnable phase features, each covering the $0-2\pi$ range, and after its optimization it remains static for each target data distribution. Details about the snapshot image generation process and the jointly optimized decoding layers are described in the Methods and Supplementary Fig. 1.

After separately training the corresponding model on the MNIST dataset[15] and the Fashion-MNIST dataset[16], we converged on two different optical generative models. In Extended Data Fig. 2a,b, the snapshot generation of images of handwritten digits and fashion products, never seen before, are presented, showing high-quality output images for all data classes within these two data distributions. We used the inception score (IS)[41] and the Fréchet inception distance (FID)[42] as image quality metrics to evaluate the snapshot image generation performance (Extended Data Fig. 2c,d). Both of these metrics are measured with a batch size of 1,000 generated images, with the random integer seed $s \in [0, 10,000)$ controlling the sampling of random Gaussian inputs $\mathcal{I}(x, y)$. In the IS evaluation, we generated the same number of images as the original dataset, aiming to measure the whole data distribution. We also carried out a $t$-test[43] between the optically generated image data and the original dataset, with $P$ values used to assess the statistical significance of improvements in the IS metric (Extended Data Fig. 2c). The higher IS values, combined with small $P$ values[43] of <0.05, indicate that our snapshot optical image generative models created statistically more diverse images compared with the original datasets. Using FID-based evaluations, we also show the statistics of 100 repeated calculations, demonstrating the consistency between the optically generated images and the original data distribution.

To further evaluate the effectiveness of the snapshot optical generative models, we trained three sets of ten binary classifiers, each based on a convolutional neural network architecture[15]. The first set of classifiers was trained using only the standard MNIST training data; the second set was trained on a 50%–50% mixed dataset composed of standard and optically generated image data; and the third set was trained on 100% optically generated data (Methods). Each classifier was tasked with recognizing a specific handwritten digit, and all the training sets had the same number of samples. These classifiers were then blindly evaluated exclusively on the standard MNIST dataset, with the classification accuracies presented in Extended Data Fig. 2e. The classifiers trained using 100% generated image data achieved, on average, a classification accuracy of 99.18% (green curve in Extended Data Fig. 2e), showing a small average decrease of 0.4% compared with the standard MNIST-data-based training results (blue curve in Extended Data Fig. 2e). Together with the outstanding IS and FID performance metrics of our optical generative model reported for each class in Extended Data Fig. 2f, these analyses indicate that the snapshot optical generation produces images of new handwritten digits that follow the target distribution (revealed by the lower average FID) but have never appeared before in their style (as indicated by the higher average IS of the optically generated images).

Next, we evaluated the influence of the output diffraction efficiency ($\eta$) on the image generation performance of an optical generative model; $\eta$ is defined as the ratio of the total optical power distributed on the image sensor divided by the total input power illuminating the optical phase seed at the SLM plane. Depending on the illumination power available and the noise level encountered in the optical hardware, the diffraction efficiency can be optimized by adding an $\eta$-related loss term during training. By training several optical generative models targeting different levels of $\eta$, we report in Extended Data Fig. 2g an empirical relationship between FID and output diffraction efficiency of these models, measured with a batch size of 200 and repeated 100 times in different random seeds. Notably, for the optical generative models with a single decoding layer (blue line), $\eta$ can be increased to 41.8% on average with a minor compromise in the image quality, highlighting the capability of the snapshot optical generative model in achieving power-efficient image synthesis. We also trained additional optical generative models with 5 successive decoding layers (orange line in Extended Data Fig. 2g), demonstrating a further improved image quality for a given level of output diffraction efficiency; for example, $\eta$ could be increased to about 50% on average while keeping FID ≈ 100 (Extended Data Fig. 2g). These analyses indicate that, for a given image quality metric that is desired, a higher output diffraction efficiency can be achieved using a deeper decoder architecture, compared with the single-layer optical decoder.

We further extended the optical generative models for multicolour image generation using three illumination wavelengths (that is, $\lambda_R$, $\lambda_G$ and $\lambda_B$). Refer to Extended Data Fig. 3 and Methods for our results and analyses on multicolour image generation.

## Iterative optical generative models

The results and analyses performed so far were on snapshot optical generative models, where each phase-encoded optical generative seed was used to create an image through an optical decoder in a single illumination. We also devised an iterative optical general model for recursively reconstructing the target data distributions from Gaussian noise[4]. As depicted in Fig. 2a, the iterative optical generative model also operates at three illumination wavelengths, where the multi-channel phase patterns encoded by the shallow digital phase encoder are sequentially loaded onto the same SLM. To showcase the generative power of this iterative optical model, we used $L_o = 5$ decoding layers that are jointly optimized and fixed for the desired target data distribution. Different from the snapshot optical generative models discussed earlier, after recording the initial intensity image $\widehat{\mathcal{I}}_t$ on the image sensor plane, the measured $\widehat{\mathcal{I}}_t$ is perturbed by Gaussian noise with a designed variance, which is regarded as the iterative optical input $\mathcal{I}_{t-1}$ at the next timestep (timestep $t \in [0, T]$ and $\mathcal{I}_T \sim \mathcal{N}(\mathbf{0}, \mathbf{I})$), where $T$ is the total timestep and $\mathcal{N}$ represents the Gaussian distribution. The training process of such an iterative optical generative model is illustrated in Fig. 2b, where we sample a batch of timesteps ($t_1, t_2, \ldots$) and accordingly add noise to the original data $\mathcal{I}_0$ to get the noised samples ($\mathcal{I}_{t_1}, \mathcal{I}_{t_2}, \ldots$). These noised samples pass through the shallow digital encoder and the iterative optical generative model to get the successive outputs. Unlike the standard DDPM implementation, the iterative optical generative model directly predicts the denoised samples, with the loss function computed against $\mathcal{I}_0$. Figure 2c outlines the blind inference process of the iterative optical generative model, where the learned optical model recursively performs denoising on the perturbed samples from timestep $T$ to 0, and the final generated image is captured on the sensor plane (see Methods for details).

Two different iterative optical generative models were trained for multicolour image generation following the distributions of the Butterflies-100[17] and Celeb-A[18] datasets (see, for example, Extended Data Fig. 4a). Compared with the multicolour generation of images using snapshot illumination per wavelength channel (reported in, for example, Extended Data Fig. 3b), the iterative optical generative model produces higher-quality images with clearer backgrounds,

AI-generated images

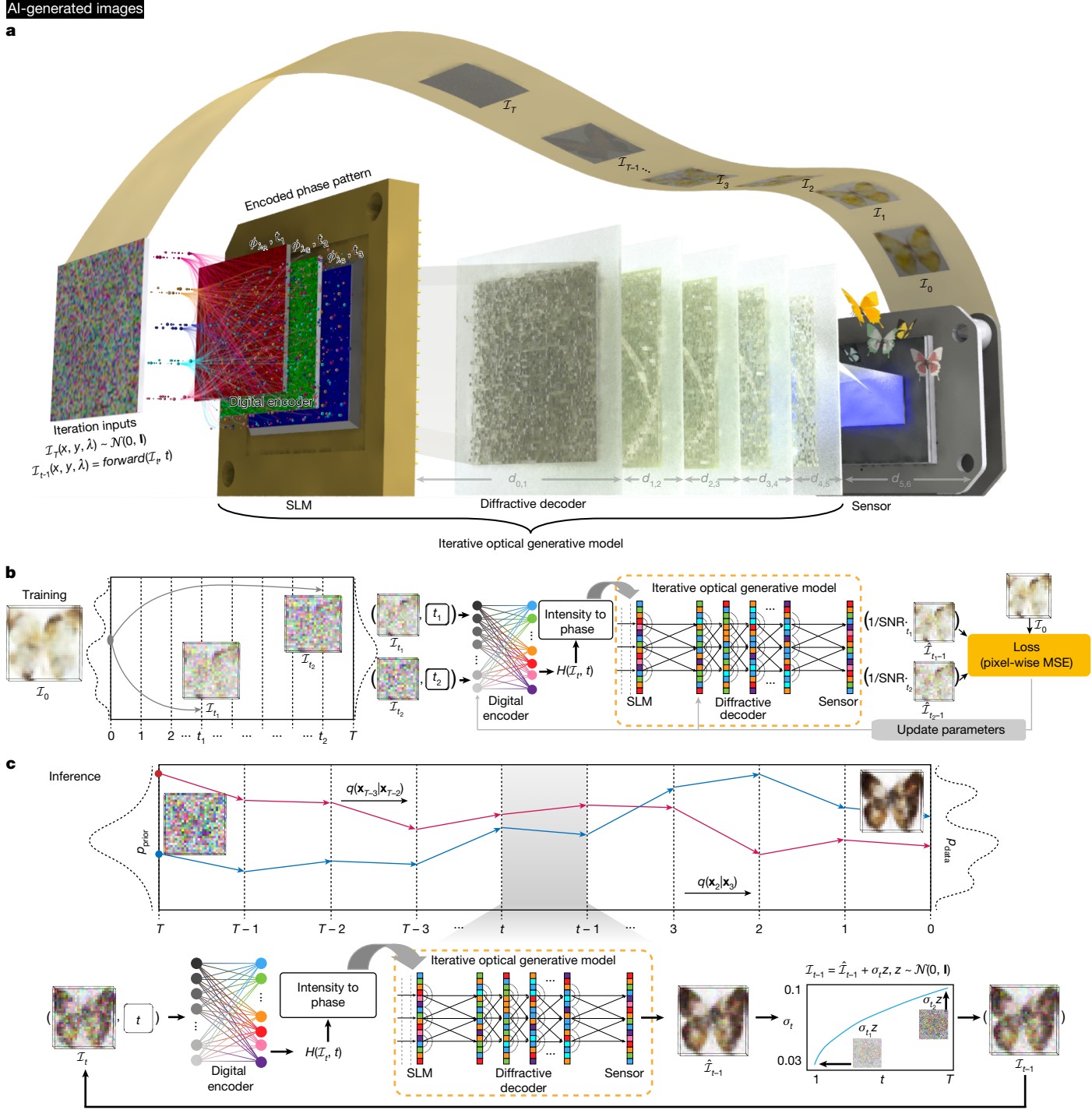

**Fig. 2 | Iterative optical generative models.** Images in panel **a**, **b** and **c** are AI-generated. **a**, Schematic of an iterative optical generative model. In each timestep, the noise-perturbed sample of the last timestep is input to the optical model. After the wave propagation, multicolour information is recorded to feed the next optical iteration with some scheduled noise added. For the last timestep, the image sensor-array records the output intensity for the final image generation. **b**, The iterative optical generative model is trained like a digital DDPM. **c**, Following the training, in its blind inference, the iterative optical generative model gradually reconstructs the target data distribution (generating an image at timestep 0) from Gaussian noise distribution (at timestep $T$). SNR, coefficient of distribution transformation; $p_{prior}$, normal distribution; $p_{data}$, original data distribution; $q$, approximate posterior probability; $\mathbf{x}_t$, state at time step $t$, that is, $\mathcal{I}_t$.

indicating its potential to realize diverse image generation without the digital diffusion guidance. Another key advantage of the iterative optical generative models is that we do not encounter mode collapse throughout the training process, as successive iterations divide the distribution mapping task into independent Gaussian processes controlled by different timesteps.

To better highlight the vital role of the collaboration between the shallow digital encoder and the diffractive decoder, we implemented an alternative iterative optical model trained on the Celeb-A dataset without using a digital encoder. This digital-encoder-free iterative optical generative model can also create multicolour images of human faces with different styles and backgrounds. This suggests that

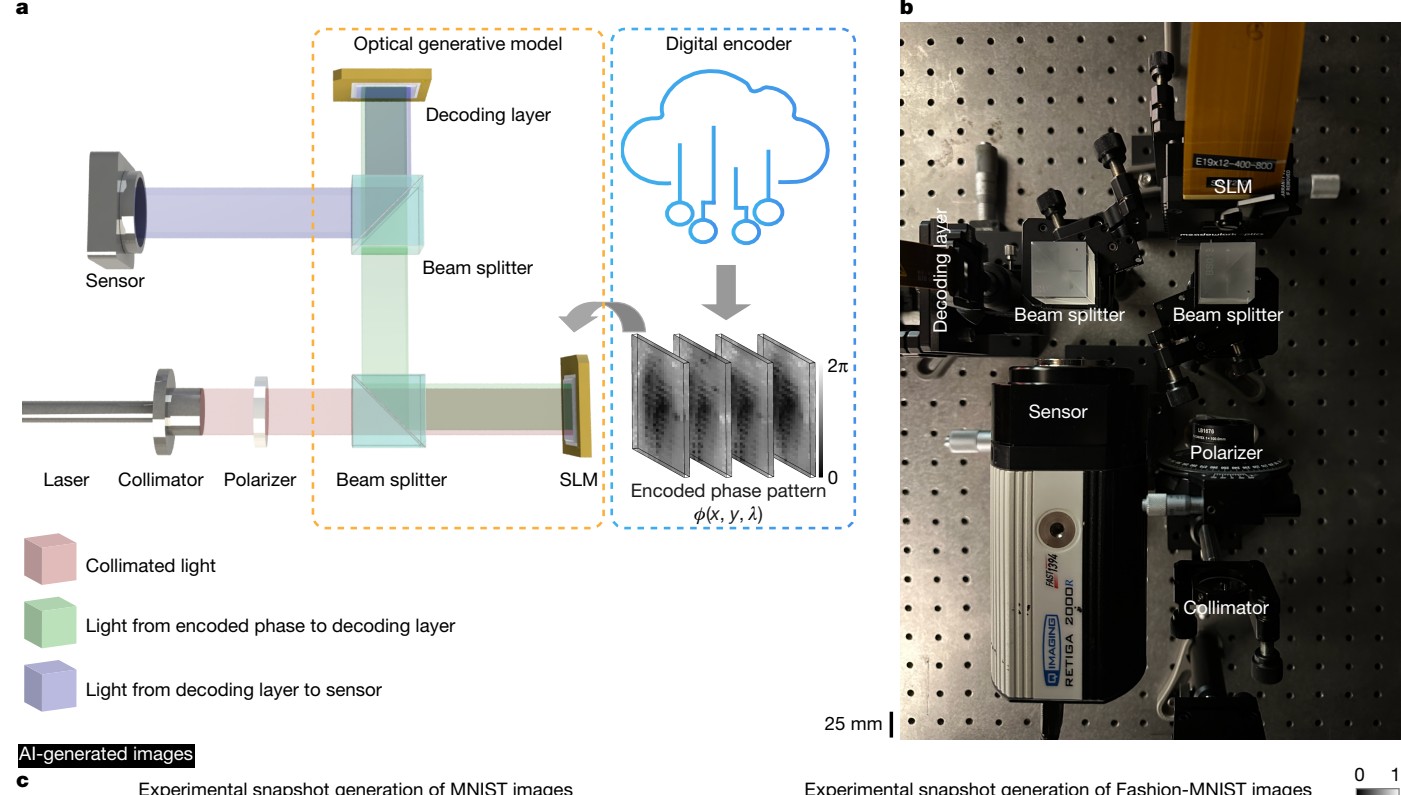

**a** Optical generative model — Decoding layer — Beam splitter — Light from encoded phase to decoding layer — Light from decoding layer to sensor — Collimated light — Sensor — Laser — Collimator — Polarizer — Beam splitter — SLM — Digital encoder — Encoded phase pattern $\phi(x, y, \lambda)$ — $2\pi$ — 0

**b** Decoding layer — SLM — Beam splitter — Beam splitter — Sensor — Polarizer — Collimator — 25 mm

AI-generated images

**c** Experimental snapshot generation of MNIST images

Experimental FID on MNIST snapshot generation: 131.08

Experimental snapshot generation of Fashion-MNIST images — 0 — 1

Experimental FID on Fashion-MNIST snapshot generation: 180.57 — 700λ

**Fig. 3 | Experimental demonstration of snapshot optical generative models. a**, Schematic of our experimental snapshot optical generative model. The encoded phase patterns, that is, the optical generative seeds, are pre-calculated and randomly accessed for each image inference task. **b**, Photograph of the snapshot optical generative model. **c**, This panel contains AI-generated images.

The experimental results of image generation using the optical generative models trained for handwritten digits and fashion products, following the target data distributions of MNIST and Fashion-MNIST, respectively. The colour bar (for normalized intensity) and the scale bar are also shown. Experimental FID evaluations compared to original datasets are also shown.

with the intensity-to-phase transformations directly implemented on an SLM without any encoder along with the photo-electric conversion at the image sensor plane, we can achieve complex domain mappings with an iterative optical generative model—although with reduced performance and image diversity compared with the results of an iterative optical generative model that uses a digital encoder.

The intermediate results $\mathcal{I}_{t-1}(t = 1{,}000,\ 800,\ \ldots,\ 20,\ 1)$ of the iterative optical generation model of Extended Data Fig. 4a are also shown in Extended Data Fig. 4b, which vividly demonstrate how the optical generative models gradually map the noise distribution into the target data domain. The FID and IS indicators of the iterative optical generative models are shown in Extended Data Fig. 4c,d, respectively, where the details of these performance assessments were the same as those used in Extended Data Fig. 3c,d. The results show an important improvement in the image generation performance of the iterative optical generative model, where the lower FID scores show that the generated images are closer to the target distribution. Furthermore, the higher IS values, along with the statistical $t$-test evaluations, indicate that the iterative optical generative model can create more diverse results than the original image dataset. We also report in the same figure the FID

and IS values of the iterative optical generative model that was trained without a digital encoder, which shows a relatively worse performance compared with the iterative optical generative model jointly trained with a shallow digital encoder.

## Experimental demonstration

We experimentally demonstrated snapshot optical generative models using a reconfigurable system operating in the visible spectrum (Fig. 3a). Laser light (520 nm) is collimated to uniformly illuminate an SLM. The SLM displays the generative optical seeds containing the pre-calculated phase patterns $\phi(x, y)$ processed by a shallow digital encoder. After passing through a beam splitter, the optical fields modulated by the encoded phase patterns corresponding to the generative optical seeds are processed by another SLM serving as a fixed or static decoder. For each optical generative model, the state of the optimized decoder surface is fixed, and the same optical architecture is switched from one state to another, generating images that follow different target distributions. At the output of the snapshot optical generative model, the intensity of the generated images is captured by an image sensor (see Fig. 3b and Methods for details).

For initial experiments, we trained two different models for generating images of handwritten digits and fashion products, following the MNIST[15] and Fashion-MNIST datasets[16], respectively. Figure 3c visualizes our experimental results for both models, which achieved experimental FID scores of 131.08 and 180.57 on the MNIST and Fashion-MNIST datasets, respectively. The successful generation of these images following the two target distributions highlights the versatility of the designed system, further validating the feasibility of snapshot optical generative models. The overall inference time is constrained by the SLM loading time, which can be minimized by using faster phase light modulators or SLMs with frame rates >1 kHz. Additional examples of optically generated snapshot images of handwritten digits and fashion products are reported in Supplementary Figs. 2 and 3 and Supplementary Videos 1 and 2.

To further explore the latent space of the snapshot optical generative model, we also performed experiments to investigate the relationship between the random noise inputs and the generated images (see Methods, Extended Data Fig. 5 and Supplementary Videos 3–9 for these latent space interpolation experiments and related analyses).

We also experimentally evaluated the snapshot optical image generation under a limited phase-encoding space (for example, $0–\pi/2$ versus $0–2\pi$) and a limited decoder bit-depth (for example, 4-bit-depth versus 8-bit-depth) using a restricted optical set-up (Supplementary Figs. 4a,b and 5). The performance comparisons between the experimental results of Fig. 3 and Supplementary Fig. 4 show the significance of employing a large phase bit-depth for the diffractive decoder as well as increasing the encoding phase range of the input SLM.

We further extended our experimental results to create higher-resolution images in the style of Van Gogh artworks, which used the same set-up shown in Fig. 3b. We experimentally demonstrated snapshot monochrome image generation for Van Gogh-style artworks using a digital encoder paired with a jointly trained diffractive decoder (Extended Data Fig. 6). The architecture of the digital encoder and the processing pipeline are shown in Supplementary Fig. 6. Additional comparisons in Supplementary Fig. 7 reveal the diffractive decoder's superior performance over free-space-based image decoding, using the same digital encoder architecture. Notably, although the free-space-based decoding completely failed in some cases, achieving a contrastive language-image pre-training score (CLIP score)[44] below 10–15, the diffractive decoder achieved stable image generation, with much better image quality at the output. As expected, we observed a numerical-aperture-related minor degradation in image resolution when increasing the SLM-to-decoder distance to match our experimental conditions (Supplementary Fig. 7 versus Supplementary Fig. 8); however, the diffractive decoder-based approach still maintains stable image generation compared with free-space-based decoding, which fails image generation in various cases despite using the same digital encoder architecture, as shown in Supplementary Fig. 8.

By further increasing the number of digital encoder parameters (Supplementary Table 1), we can improve the resolution and image quality of the optically generated Van Gogh-style artworks that are created in a snapshot; detailed comparisons are provided in Extended Data Fig. 7 for the number of trainable parameters spanning 44 million to 580 million. Figures 4 and 5 show our experimental results for higher-resolution monochrome and colour (red–green–blue (RGB)) image generation using a digital encoder with 580 million parameters. The monochrome images of Van Gogh-style artworks were generated with 520-nm illumination, whereas the colour images used sequential wavelengths of {450, 520, 638} nm for the B, G and R channels. In Fig. 4, the left three columns show the results where the snapshot images created by the optical generative model in a single pass closely resemble those produced by the digital diffusion model (that is, the teacher model with 1.07 billion trainable parameters and 1,000 inference steps per image), demonstrating the consistency of our image generation process with respect to the teacher diffusion model. Conversely, the three right columns, highlighted within the orange box, showcase the optical model's ability to generate diverse images that differ from those of the teacher digital diffusion model, illustrating its creative variability at the output (also see additional experimental results for Van Gogh-style artworks supporting our conclusions in Supplementary Figs. 9 and 10).

For the multicolour Van Gogh-style artwork generation, phase-encoded generative seed patterns at each wavelength channel were generated and sequentially loaded onto the SLM. Under the illumination of corresponding wavelengths, multicolour images were generated through a fixed or static diffractive decoder and merged digitally; stated differently, the same decoder state was shared across all the illumination wavelengths for all the image generation. Figure 5 presents the multicolour Van Gogh-style artwork generation results, including artistic examples that either match or differ from the outputs of the teacher digital diffusion model, which used 1.07 billion trainable parameters and 1,000 inference steps per image generation. Although slight chromatic aberrations were observed, the generated high-resolution colour images maintained high quality. Additional experimental results of Van Gogh-style colour artworks are provided in Supplementary Figs. 11 and 12.

To quantify the fidelity of the experimental optical generative model, Supplementary Fig. 13 reports the peak signal-to-noise ratio values calculated between the numerically simulated and the experimentally generated results. These quantitative comparisons for both the snapshot monochrome and the multicolour optical generative models demonstrate that the experimental outputs match simulations. In addition, Supplementary Fig. 14 presents the CLIP score evaluations corresponding to the results shown in Figs. 4 and 5, highlighting the semantic consistency achieved by the optical generative model.

## Discussion

This work demonstrated snapshot optical image generation from noise patterns by leveraging a diffractive network architecture. Earlier free-space-based optical networks primarily focused on tasks such as, for example, computational imaging and sensing, noise estimation and filtration, or data classification[22,35,36,45–51]. By contrast, our framework optically generates diverse images from noise, showcasing a highly desired 'creative' snapshot image generation capability that extends beyond the scope of previous studies. Moreover, without changing its architecture or physical hardware, optical generation for a different data distribution can be implemented by reconfiguring the diffractive decoder to a new optimized state. This versatility of optical generative models might be especially important for edge computing, augmented reality or virtual reality displays, also covering different entertainment-related applications.

Our results also showed that, guided by a teacher DDPM, the knowledge of the target distribution can be distilled, as illustrated by the images synthesized by our optical generative models. This distillation enables the optical generative model to capture semantic information effectively, as detailed in Extended Data Fig. 8 and Methods. Furthermore, mimicking the diffusion process, our iterative optical generative models can learn the target distribution in a self-supervised way, which avoids mode collapse, generating even more diverse results than the original dataset, as illustrated in Extended Data Fig. 4. Moreover, the iterative optical generative models have the potential to eliminate the use of a digital encoder and produce diverse outputs following different data distributions. A phase-encoding strategy also provides a critical nonlinear information encoding mechanism for optical generative models, as detailed in Extended Data Fig. 9 and Methods.

There are also some challenges associated with optical generative models in general. Potential misalignments and physical imperfections within the optical hardware and/or set-up present challenges.

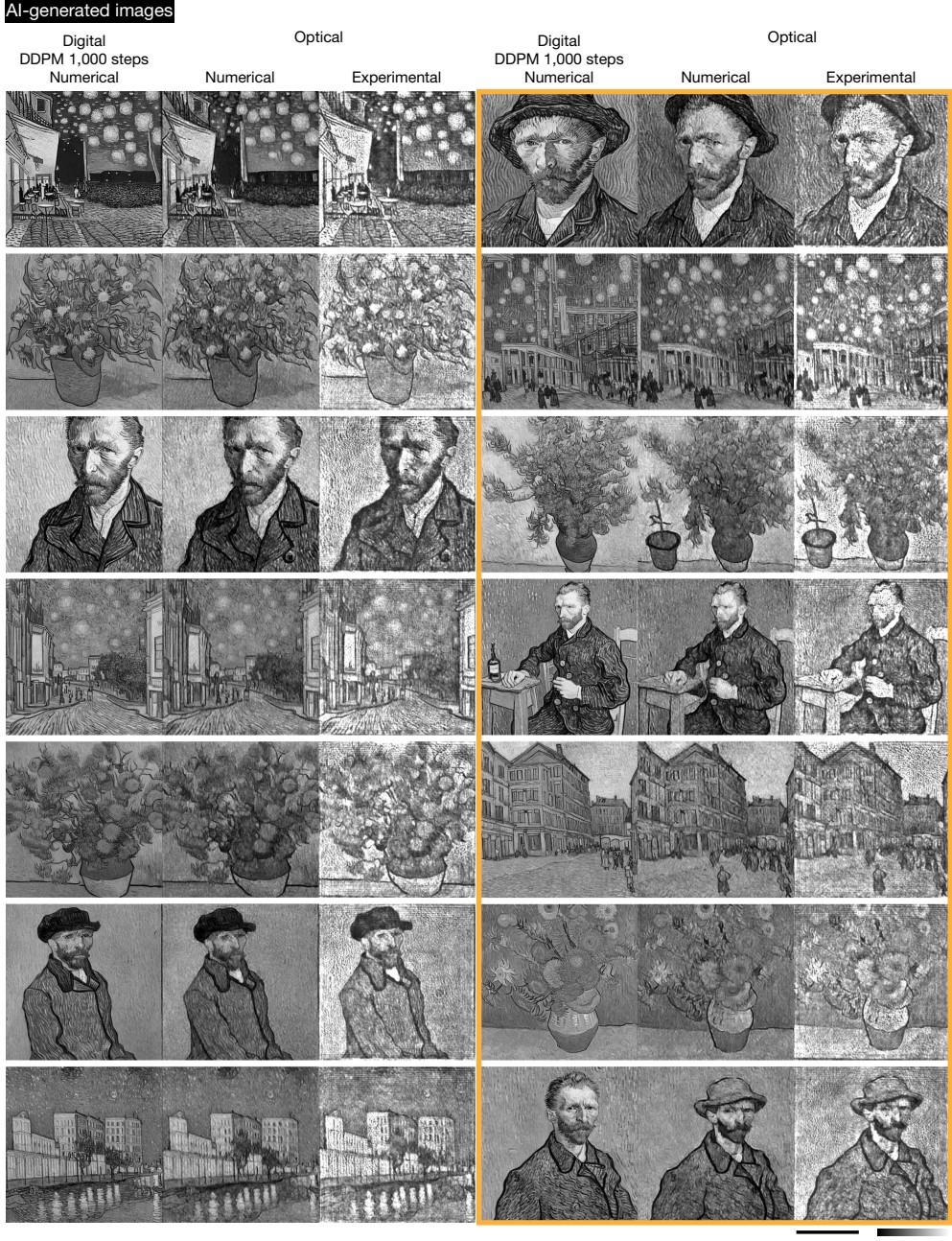

AI-generated images

Digital DDPM 1,000 steps — Numerical | Optical Numerical | Optical Experimental

2 mm    0    1

**Fig. 4 | Numerical and experimental results of a higher-resolution snapshot optical generative model for monochrome Van Gogh-style artwork generation compared against the teacher digital diffusion model with 1,000 steps.** This figure contains AI-generated images. We present comparative results on monochrome Van Gogh-style artwork generation for both the digital teacher diffusion model (with 1.07 billion trainable parameters and 1,000 steps used for each inference) and the snapshot optical generative model, along with the experimental results for the snapshot optical generative model. The orange box on the right reveals the obvious discrepancies observed between the digital DDPM teacher and the optical model, demonstrating the capability of the snapshot optical generative model to create diverse images beyond those produced by the digital teacher diffusion model. The digital phase encoder has 580 million trainable parameters and each snapshot optical image is generated by a unique random noise input. Input text ('architecture' or 'plants' or 'person') is used to generate different artworks. The colour bar and scale bar are also shown. More experimental results of Van Gogh-style artwork generation are provided in Supplementary Figs. 9 and 10.

Another challenge is the limited phase bit-depth of the optical modulator devices or surfaces that physically represent the generative optical seeds and the decoder layer(s). To investigate this, we numerically analysed three scenarios with varying phase bit-depth levels and evaluated their impact by imposing these constraints during the testing (Supplementary Fig. 15). Remarkably, models trained without such constraints were still able to generate handwritten digits despite the imposed phase bit-depth limitations during testing. To mitigate these challenges in the optical set-up, one can integrate these limitations directly into the training process, making the in-silico-optimized system align better with the physical limits and the capabilities of the local hardware. This strategy led to notable performance improvements compared with models that did not account for such bit-depth limitations during training (Supplementary Fig. 15). A key insight of this analysis is that a relatively simpler decoder surface with just three discrete phase levels (covering only 0, $2\pi/3$ or $4\pi/3$ per feature) would

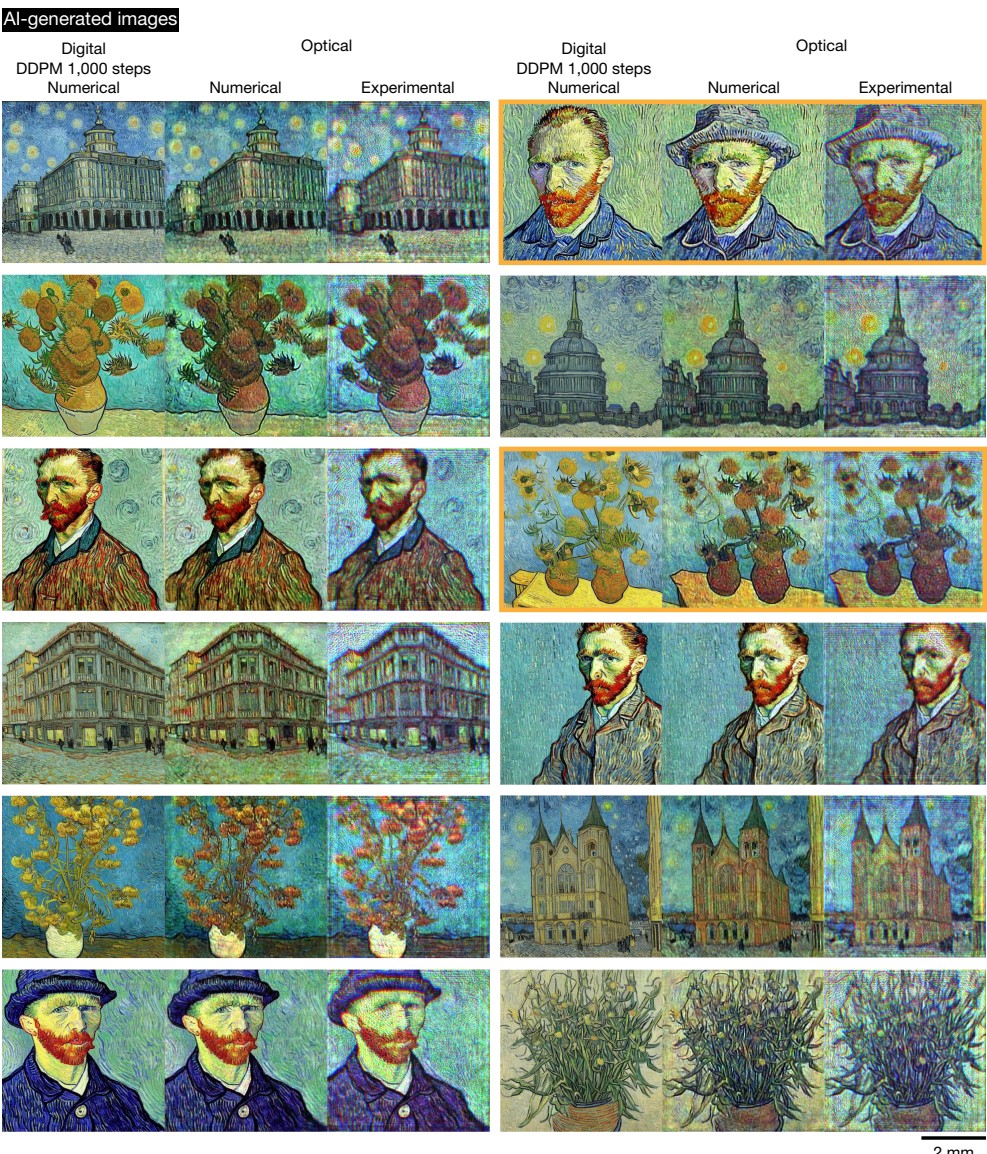

**Fig. 5 | Numerical and experimental results of a multicolour optical generative model for colourful Van Gogh-style artwork generation, compared against the teacher digital diffusion model with 1,000 steps.** This figure contains AI-generated images. We present numerical and experimental results of a multicolour optical generative model for colourful Van Gogh-style artwork generation, compared against the teacher digital diffusion model (with 1.07 billion trainable parameters and 1,000 steps used for each inference). The orange boxes on the right reveal the obvious discrepancies observed between the digital teacher and the optical model, demonstrating the capability of the multicolour optical generative model to create diverse images beyond those produced by the digital teacher diffusion model. The digital phase encoder has 580 million trainable parameters and each snapshot optical image of the RGB channels is generated by a unique random noise input. Input text ('architecture' or 'plants' or 'person') is used to generate different artworks. The scale bar is also provided. Additional experimental results of Van Gogh-style colour artworks are provided in Supplementary Figs. 11 and 12.

be sufficient, opening the door to replacing the decoder with a passive, thin surface fabricated by, for example, two-photon polymerization or optical lithography-based nanofabrication techniques[52,53]. This would further simplify the physical set-up of the local optical generative model, also making it more compact, lightweight and cost-effective.

Using the presented approach, spatially and/or spectrally multiplexed optical generative models can also be designed for the parallel generation of many independent images across different spatial and spectral channels (see, for example, Methods and Extended Data Fig. 10 for spectrally multiplexed optical generative models). Moreover, benefiting from the inherent advantages of diffractive decoders for rapid processing of visual information, optical generative models can also realize three-dimensional image generation across a volume, which could open up opportunities for augmented reality, virtual reality and entertainment-related applications, among others.

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

## Methods

### Snapshot optical image generation process

The snapshot optical generation procedure includes two parts: the digital encoder and the optical generative model. For image generation of MNIST, Fashion-MNIST, Butterflies-100 and Celeb-A, the digital encoder architecture that we selected is a variant of the multi-layer perceptron[54], which contained $L_d$ fully connected layers, each of which was followed by a subsequent activation function $\kappa$. For a randomly sampled input $\mathcal{I}(x,y) \sim \mathcal{N}(\mathbf{0}, \mathbf{I})$, where $x \in [1, h]$, $y \in [1, w]$ with $h$ and $w$ denoting the height and width of the input noise pattern, respectively, the digital encoder processes it and outputs the encoded signal $\mathcal{H}^{(l_d)}$ with a predicted scaling factor $s$ (refer to Supplementary Information section 1.1 for details). The one-dimensional output signal of the digital encoder is reshaped into a 2D signal $\mathcal{H}^{(L_d)} \in \mathbb{R}^{h \times w}$. For the Van Gogh-style artwork generation, the digital encoder consists of three parts: a noise feature processor, an in silico field propagator and a complex field converter (refer to Supplementary Information section 1.2 for details). The random sampled input is processed to a 2D output signal $\mathcal{H}^{(l_d)} \in \mathbb{R}^{h \times w}$ and is forwarded to the subsequent optical generative model as outlined below.

In our implementation, the optical generative model, without loss of generality, consisted of an SLM and a diffractive decoder, which included $L_o$ decoding layers. To construct the encoded phase pattern $\phi(x,y)$ projected by the SLM, the 2D real $\mathcal{H}^{(L_d)}$ was normalized to the range $[0, \alpha\pi]$, which can be formulated as:

$$\phi(x, y) = (\alpha\pi(\mathcal{H}^{(L_d)} + 1))/2 \tag{1}$$

Here, $\alpha$ is the coefficient controlling the phase dynamic range.

After constructing the incident complex optical field profile $\mathcal{U}^{(0)}(x,y) = \cos(\phi(x,y)) + i\sin(\phi(x,y))$ from the encoded phase pattern, the light propagation through air was modelled by the angular spectrum method[55], where $i$ is the imaginary unit. The free-space propagation of the incident complex field $\mathcal{U}(x,y)$ over an axial distance of $d$ in a medium with a refractive index $n$ can be written as:

$$\mathscr{O}(x, y) = \mathcal{P}_f^d(\mathcal{U}(x, y)) = \mathcal{F}^{-1}\{\mathcal{F}\{\mathcal{U}(x, y)\}\mathcal{M}(f_x, f_y; d; n)\} \tag{2}$$

where $\mathscr{O}(x,y)$ is the 2D output complex field, the operator $\mathcal{P}_f^d(\cdot)$ represents the free-space propagation over an axial distance $d$, $\mathcal{F}\{\cdot\}$ ($\mathcal{F}^{-1}\{\cdot\}$) is the 2D (inverse) Fourier transform, and $\mathcal{M}(f_x, f_y; d; n)$ is the transfer function of free-space propagation:

$$\mathcal{M}(f_x, f_y; d; n) = \begin{cases} 0, & f_x^2 + f_y^2 > \dfrac{n^2}{\lambda^2} \\ \exp\left\{ jkd\sqrt{1 - \left(\dfrac{2\pi f_x}{nk}\right)^2 - \left(\dfrac{2\pi f_y}{nk}\right)^2} \right\}, & f_x^2 + f_y^2 \le \dfrac{n^2}{\lambda^2} \end{cases} \tag{3}$$

where $j = \sqrt{-1}$, $\lambda$ is the wavelength of illumination in the air, $k = 2\pi/\lambda$ is the wavenumber, $f_x$ and $f_y$ are the spatial frequencies on the $x$–$y$ plane, and $+z$ is the propagation direction.

The decoding layers (one or more) were modelled as phase-only modulators for the 2D complex incident fields, where the output fields $\mathscr{O}(x,y)$ under the decoding phase modulation $\mathcal{P}_m$ of the $l_o$th decoding layer can be expressed as:

$$\mathscr{O}(x, y) = \mathcal{P}_m^{\phi^{(l_o)}}(\mathcal{U}(x, y)) = \mathcal{U}(x, y) \cdot \exp(j\phi^{(l_o)}(x, y)) \tag{4}$$

where $\phi^{(l_o)}(x,y)$ represents the phase modulation values of the diffractive features on the $l_o$th decoding layer, which were trained jointly with the shallow digital encoder to realize snapshot optical image generation. Therefore, the optical field $\mathscr{O}(x,y)$ at the output or image plane

can be calculated by iteratively applying the free-space propagation $\mathcal{P}_f$ and decoding phase modulation $\mathcal{P}_m$:

$$\mathscr{O}(x, y) = \mathcal{P}_f^{d_{L_o, L_o+1}}\left[ \prod_{l_o=1}^{L_o} \mathcal{P}_m^{\phi^{l_o}} \mathcal{P}_f^{d_{l_o-1, l_o}} \right](\mathcal{U}^{(0)}(x, y)) \tag{5}$$

where $d_{l_o-1, l_o}$ represents the axial distance between the $(l_o - 1)$th and the $l_o$th decoding layers. After propagating through all the components, the generated intensity $\mathcal{O}$ at the sensor plane can be calculated as the square of the complex amplitude, and the diffractive decoder's output intensity was normalized based on its maximum, $\max(\mathcal{O})$.

For the model operating across multiple wavelengths, for example, $\lambda_{(R,G,B)}$, the underlying logic of the forward propagation remained unchanged. During the forward procedure, the digital encoder converts the multi-channel random inputs $\mathcal{I}(x, y, \lambda)$ into the encoded phase patterns $\phi(x, y, \lambda)$ at different wavelengths, and these patterns were displayed on the SLM sequentially, for example, $\phi_{\lambda_R, t_1}, \phi_{\lambda_G, t_2}, \phi_{\lambda_B, t_3}$ for the red, green and blue illuminations, respectively. The phase modulation of the decoding layer for each $\lambda$ is calculated as follows:

$$\phi^{(l_o)}(x, y, \lambda) = \frac{\lambda_{\text{centre}}(n_\lambda - 1)}{\lambda(n_{\lambda_{\text{centre}}} - 1)} \phi^{(l_o)}(x, y, \lambda_{\text{centre}}) \tag{6}$$

where the $\phi^{(l_o)}(x, y, \lambda_{\text{centre}})$ is the phase modulation of the central wavelength $\lambda_{\text{centre}}$, which is the 2D optimizable parameters for each decoding layer. $n_\lambda$ is the refractive index of the optical decoder material as a function of wavelength. It is noted that equation (6) is for calculating the modulation shift of the decoding layer with a structured material. This condition does not need to hold for the reconfigurable decoder scheme using sequential multicolour illumination, where each wavelength can have its corresponding phase profile, fully utilizing all the degrees of freedom of the reconfigurable decoder surface.

### Training strategy for optical generative models

The goal of the generative model is to capture the underlying distribution of data $p_{\text{data}}(\mathcal{I})$ so that it can generate new samples from $p_{\text{model}}(\mathcal{I})$ that resemble the original data classes. $\mathcal{I}$, as the input, follows a simple and accessible distribution, typically a standard normal distribution. To achieve this goal, a teacher digital generative model based on DDPM[4] was used to learn the data distribution first. The details of training the DDPM are presented in Supplementary Information section 2. After the teacher generative model learns the target distribution $p_{\text{data}}(\mathcal{I})$, the training of the snapshot optical generative model was assisted by the learned digital teacher model, where the goal of the proposed model was formulated as:

$$\mathcal{L}(\theta) = \min_{\theta}\{\text{MSE}(\mathcal{O}_{\text{teacher}}, s\mathcal{O}_{\text{model}}) + \gamma \text{KL}(p_{\text{teacher}}\|p_{\text{model}}^{\theta})\} \tag{7}$$

where $p_{\text{teacher}}(\mathcal{I}) \sim p_{\text{data}}(\mathcal{I})$ represents the learned distribution of the teacher generative model, MSE is the mean square error, and $\text{KL}(\cdot\|\cdot)$ refers to the Kullback–Leibler (KL) divergence[2]. $p_{\text{model}}^{\theta}(\mathcal{I})$ is the distribution captured by the snapshot optical generative model and $\theta$ is the optimizable parameters of it. In the implementation, all the $p(\mathcal{I})$ were measured using the histograms of the generated images $\mathcal{O}(x, y)$. $\mathcal{O}_{\text{model}}$ refers to the generated images from the optical generative model, $\mathcal{O}_{\text{teacher}}$ denotes the samples from the learned teacher model, $s$ is a scaling factor predicted by the digital encoder and $\gamma$ is an empirical coefficient.

### Implementation details of snapshot optical generative models

For the MNIST and Fashion-MNIST datasets, which include the class labels, the first layer's input features in the digital encoder were formulated as:

$$\mathcal{H}^{(0)} = \text{concate}(\text{flatten}(\mathcal{I}(x, y)), \text{embedding}(\mathcal{C}, l)) \tag{8}$$

where $\mathcal{C}$ is the class label of the target generation and $l$ is the size of embedding($\cdot$) operation. This resulted in an input feature $\mathcal{H}^{(0)} \in \mathbb{R}^{xy+l}$. The Butterflies-100 and Celeb-A datasets do not have explicit class labels, so the operations were the same as illustrated before, that is, $\mathcal{H}^{(0)} = \text{flatten}(\mathcal{I}(x,y))$. The input resolutions $(x,y)$ for all datasets were set to 32 × 32, and the class label embedding size $l$ was also set to 32. The digital encoder contained $L_d = 3$ fully connected layers and the number of neurons $m_{l_d}$ in each layer was the same as the size of the input feature, that is, $xy + l$ for the MNIST and Fashion-MNIST datasets, and $3xy$ for the Butterflies-100 and Celeb-A datasets owing to three-colour channels. The activation function $\kappa$ uses LeakyReLU($\cdot$) with a slope of 0.2.

In the numerical simulations, the minimum lateral resolution of the optical generative model was 8 μm, and the illumination wavelength was 520 nm for monochrome operation and (450 nm, 520 nm, 638 nm) for multicolour operation. The number $L_o$ of the decoding layer was set to 1, the axial distance from the SLM plane to the decoding layer $d_{0,1}$ was 120.1 mm, and the distance from the decoding layer to the sensor plane $d_{1,2}$ was 96.4 mm. In the construction of the encoded phase pattern, that is, the optical generative seed $\phi(x,y)$, the coefficient $\alpha$ that controlled the phase range of normalization was set to 2.0, and the normalized profile was up-sampled in both the $x$ and the $y$ directions by a factor of 10. Hence, the size and the resolution of the object plane were 2.56 mm × 2.56 mm and 320 × 320, respectively. The number of optimizable features on the decoding layer was 400 × 400. On the image plane, the 320 × 320 intensity measurement was down-sampled by a factor of 10 for the loss calculations.

During the training of the teacher DDPM, the total timestep $T$ was set to 1,000. The noise prediction model $\epsilon_{\theta_{\text{proxy}}}(\mathcal{I}_t, t)$ shared the same structure profile as the general DDPM. $\beta_t$ was a linear function from $1 \times 10^{-4}$ ($t = 1$) to 0.02 ($t = T$). In the training of the snapshot optical generative models, the histogram of KL divergence was calculated using normalized integer intensity values within the range of [−1, 1], and the regularization coefficient $\gamma$ was set to $1 \times 10^{-4}$. All the generative models were optimized using the AdamW optimizer[56]. The learning rate for the digital parameters (digital encoder and DDPM) was $1 \times 10^{-4}$, and for the decoding layer was $2 \times 10^{-3}$, with a cosine annealing scheduler for the learning rate. The batch sizes were set as 200 for DDPM and 100 for the snapshot generative model. All the models were trained and tested using PyTorch 2.21[57] with a single NVIDIA RTX 4090 graphics processing unit.

For the Van Gogh-style artwork generation, three class labels were used as the conditions for image generation: {'architecture', 'plants' and 'person'}. The input resolution $(x,y)$ of latent noise was set to (80, 80) and the size of the class label embedding $l$ was 80. We added perturbations to the input noise so that the optical generative model was easier to cover the whole latent space[58]. The numerical simulations of monochrome and multicolour artwork generation shared the same physical distance and wavelength as the lower-resolution optical image generation. The object plane size was 8 mm × 8 mm with a resolution of 1,000 × 1,000. The number of optimizable features on the decoding layer was 1,000 × 1,000. On the image plane, the size and the resolution were 5.12 mm × 5.12 mm and 640 × 640, respectively. For the teacher DDPM used to generate Van Gogh-style artworks, we fine-tuned the pretrained Stable Diffusion v1.5[58] with the vangogh2photo dataset[19] and captioned it by a GIT-base model[59]. The total timestep $T$ was set to 1,000 steps and $\beta_t$ was a linear function from 0.00085 ($t = 1$) to 0.012 ($t = T$). The models were trained and tested using PyTorch 2.21 with four NVIDIA RTX 4090 graphics processing units. More details can be found in Supplementary Information section 3.

To evaluate the quality of snapshot optical image generation, IS[41] and FID[42] indicators were used to quantify the diversity and fidelity of the generated images compared with the original distributions. For the class-conditioned generation, for example, handwritten digits, we further examined the effectiveness of snapshot optically generated images

by comparing the classification accuracy of individual binary classifiers trained on different dataset compositions. As shown in Extended Data Fig. 2e, each binary classifier, based on the same convolutional neural network architecture, was trained to determine whether a given handwritten digit belongs to a specific digit or class. The standard MNIST dataset, the 50%–50% mixed dataset, and the optically generated image dataset each contained 5,000 images per target digit and 5,000 non-target digits, where the non-target digits were sampled uniformly from the remaining classes. To simulate variations in handwritten stroke thickness, we augmented the optically generated images by applying binary masks generated through morphological operations (erosion and dilation) with random kernel sizes. To evaluate the high-resolution image generation, the CLIP score was used to quantify the alignment between the generated images and the referenced text (detailed in Extended Data Fig. 7).

## Multicolour optical generative models

Extended Data Fig. 3a shows a schematic of our multicolour optical generative model, which shares the same hardware configuration as the monochrome counterpart reported in 'Results'. For multicolour image generation, random Gaussian noise inputs of the three channels $(\lambda_R, \lambda_G, \lambda_B)$ are fed into a shallow and rapid digital encoder, and the phase-encoded generative seed patterns at each wavelength channel $(\phi_{\lambda_R}, \phi_{\lambda_G}, \phi_{\lambda_B})$ are sequentially loaded $(t_1, t_2, t_3)$ onto the same input SLM (Extended Data Fig. 3a). Under the illumination of corresponding wavelengths in sequence, multicolour images following a desired data distribution are generated through a fixed diffractive decoder that is jointly optimized for the same image generation task. The resulting multicolour images are recorded on the same image sensor as before. We numerically tested the multicolour optical image generation framework shown in Extended Data Fig. 3a using 3 different wavelengths (450 nm, 520 nm, 638 nm), where 2 different generative optical models were trained on the Butterflies-100 dataset[17,60] and the Celeb-A dataset[18] separately. Because these two image datasets do not have explicit categories, the shallow digital encoder used only randomly sampled Gaussian noise as its input without class label embedding. For example, Extended Data Fig. 3b shows various images of butterflies produced by the multicolour optical generative model, revealing high-quality output images with various image features and characteristics that follow the corresponding data distribution. In Extended Data Fig. 3c,d, the FID and IS performance metrics on the Butterflies-100 and Celeb-A datasets are also presented. The IS metrics and the $t$-test results show that the optical multicolour image generation model provides a statistically significant improvement ($P < 0.05$) in terms of image diversity and IS scores compared with the original Butterflies-100 dataset, whereas it does not show a statistically significant difference compared with the original Celeb-A data distribution. In addition, some failed image generation cases are highlighted with red boxes in the bottom-right corner of Extended Data Fig. 3b. These rare cases were automatically identified based on a noise variance criterion, where the generated images with estimated noise variance ($\sigma^2$) exceeding an empirical threshold of 0.015 were classified as generation failures[61] (Supplementary Fig. 16). Such image generation failures were observed in 3.3% and 6.8% of the optically generated images for the Butterflies-100 and Celeb-A datasets, respectively. Extended Data Fig. 3e reveals that this image generation failure becomes more severe for the optical generative models that are trained longer. This behaviour is conceptually similar to the mode collapse issue sometimes observed deeper in the training stage, making the outputs of the longer-trained multicolour optical generative models limited to some repetitive image features.

## Performance analyses and comparisons

We conducted performance comparisons between snapshot optical generative models and all-digital deep-learning-based models formed

by stacked fully connected layers[62], trained on the same image generation task. Supplementary Figs. 17–21 present different configurations of these optical and all-digital generative models. In this analysis, we report their computing operations (that is, the floating-point operations (FLOPs)), training parameters, average IS values and examples of the generated images, providing a comprehensive comparison of these approaches. The digital generative models in Supplementary Fig. 18 were trained in an adversarial manner, which revealed that when the depth of the all-digital deep-learning-based generative model is shallow, the output image quality cannot capture the whole distribution of the target dataset, resulting in failures or repetitive generations. However, the snapshot optical generative model with a shallow digital encoder is able to realize a statistically comparable image generation performance, matching the performance of a deeper digital generative model stacked with nine fully connected layers (Supplementary Fig. 18). To provide additional comparisons, the digital models in Supplementary Figs. 19–20 were trained using the same teacher DDPM as used in the training of the optical generative models, and the results showed similar conclusions. In Supplementary Fig. 21, we also show comparisons using the digital DDPM, where the number of parameters of the U-Net in DDPM was reduced to match that of our shallow digital encoder, which resulted in some image generation failures in the outputs of the digital DDPM (despite using 1,000 denoising steps), which are exemplified with the red squares in Supplementary Fig. 21c. Overall, our findings reported in Supplementary Figs. 18–21 suggest that using a large DDPM as a teacher for the optical generative models can realize stable synthesis of images in a single snapshot through a shallow digital phase encoder followed by the optical diffractive decoder.

We also compared the architecture of our snapshot optical generative models with respect to a free-space propagation-based optical decoding model, where the diffractive decoder was removed (Supplementary Fig. 22a,b). The results of this comparison demonstrate that the diffractive decoder surface has a vital role in improving the visual quality of the generated images. In Supplementary Fig. 22c, we also analysed the class embedding feature in the digital encoder; this additional analysis reveals that the snapshot image generation quality of an optical model without class embedding is lower, indicating that this additional information conditions the optical generative model to better capture the overall structure of the underlying target data distribution.

To further shed light on the physical properties of our snapshot optical generative model, in Supplementary Fig. 23, we report the performance of the optical generative models as a function of the encoding phase range: [0–$\alpha\pi$]. Our analyses revealed that [0–$2\pi$] input phase encoding at the SLM provides better image generation results, as expected. In Supplementary Fig. 17a, we also explored the empirical relationship between the resolution of the optical generative seed phase patterns and the quality of the generated images. As the spatial resolution of the encoded phase seed patterns decreases, the quality of the image generation degrades, revealing the importance of the space-bandwidth product at the generative optical seed.

Furthermore, in Supplementary Fig. 15, we explored the impact of limited phase modulation levels (that is, a limited phase bit-depth) at the optical generative seed plane and the diffractive decoder. These comparisons revealed that image generation results could be improved by including the modulation bit-depth limitation (owing to, for example, inexpensive SLM hardware or surface fabrication limitations) in the forward model of the training process. Such a training strategy using a limited phase bit-depth revealed that the fixed or static decoder surface could work with 4-phase bit-depth and even 3 discrete levels of phase (for example, 0, $2\pi/3$, $4\pi/3$) per feature to successfully generate images through its decoder phase function (Supplementary Fig. 15). This is important as most two-photon polymerization or optical lithography-based fabrication methods[52,53] can routinely fabricate surfaces with 2–16 discrete phase levels per feature, which could help replace the decoder SLM with a passive fabricated surface structure.

We also investigated the significance of our diffusion model-inspired training strategy for the success of snapshot optical generative models (Supplementary Fig. 17b). When training an optical generative model as a generative adversarial network[1] or a variational autoencoder[2], we observed difficulty for the optical generative model to capture the underlying data distribution, resulting in a limited set of outputs that are repetitive or highly similar to each other—failing to generate diverse and high-quality images following the desired data distribution.

For the generation of colourful Van Gogh-style artworks, we also conducted performance comparisons for the optical generative model, reduced-size diffusion model (matching the size of our phase encoder) and the pretrained teacher diffusion model, as shown in Extended Data Fig. 8. Compared with our optical generative model, the reduced-size diffusion model that matches the size of our phase encoder produced inferior images with limited semantic details despite using 1,000 inference steps. The optical generative model outputs match the teacher diffusion model (which also used 1.07 billion trainable parameters with 1,000 inference steps). Furthermore, the CLIP score evaluations suggest that the optically generated images show good alignment with the underlying semantic content. Additional evaluations on the Van Gogh-style artwork generation are presented in Supplementary Figs. 13 and 14, where the peak signal-to-noise ratio and the CLIP scores are reported to demonstrate consistency at both the pixel level and semantic level. As there are only about 800 authenticated Van Gogh paintings available, computing IS or FID indicators against a limited data distribution is not meaningful and will be less stable.

## Phase encoding versus amplitude or intensity encoding

The phase-encoding strategy employed by the optical generative model provides an effective nonlinear information encoding mechanism as linear combinations of phase patterns at the input do not create complex fields or intensity patterns at the output that can be represented as a linear superposition of the individual outputs. In fact, this phase-encoding strategy enhances the capabilities of the diffractive decoding layer; for comparison, we trained optical generative models using amplitude encoding or intensity encoding, as presented in Extended Data Fig. 9, which further highlights the advantages of phase encoding with its superior performance as quantified by the lower FID scores on generated handwritten digit images. Similarly, for the generation of Van Gogh-style artworks, the optical generative models using amplitude encoding or intensity encoding failed to produce consistent high-quality and high-resolution output images, as shown in Extended Data Fig. 9, whereas the phase-encoding strategy successfully generated Van Gogh-style artworks. These comparisons underscore the critical role of phase encoding in the optical generative model.

## Implementation details of iterative optical generative models

DDPM is generally modelled as a Markovian noising process $q$, which gradually adds noise to the original data distribution $p_{\text{data}}(\mathcal{I})$ to produce noised samples $\mathcal{I}_1$ to $\mathcal{I}_T$. Our iterative optical generative models also employed a similar scheme to perform iterative generation, that is, training with the forward diffusing procedure and inferring with the reverse procedure. The reverse process performed two iterative operations: first, predicting the $\varepsilon_t$ to get the mean values of the Gaussian process $q(\mathcal{I}_{t-1} \mid \mathcal{I}_t, \mathcal{I}_0)$, then adding a Gaussian noise whose variance was predetermined. The target of the iterative optical generative model was to predict the original data $\mathcal{I}_0$.

Although the distribution of $\mathcal{I}_t$ and the mean $m_{t-1,t}$ of $q(\mathcal{I}_{t-1} \mid \mathcal{I}_t, \mathcal{I}_0)$ are different, they can be successively represented by the Gaussian process of $\mathcal{I}_0$ (see Supplementary Information sections 2.2 and 2.3 for details):

$$\mathcal{I}_t \sim \mathcal{N}(\sqrt{\bar{\alpha}_t}\mathcal{I}_0, \sqrt{1-\bar{\alpha}_t}\mathbf{I}) \qquad (9)$$

$$m_{t-1,t} \sim \mathcal{N}\left( \frac{\mathcal{I}_t}{\sqrt{\alpha_t}}, \frac{1-\alpha_t}{\sqrt{\alpha_t(1-\bar{\alpha}_t)}}\mathbf{I} \right) \tag{10}$$

Therefore, we introduced a coefficient of $SNR_t = \sqrt{\bar{\alpha}_t}/\sqrt{\alpha_t}$ to realize the transformation on the target distribution (see Supplementary Information section 2.4 for details). The loss function for iterative optical generative models was formed as:

$$\mathcal{L}(\theta) = \min_{\theta_{model}} E_{t \sim [1,T], \mathcal{I}_0 \sim p_{data}(\mathcal{D})}[\|SNR_t\mathcal{I}_0 - \mathcal{O}_{\theta_{model}}(\mathcal{I}_t, t)\|^2] \tag{11}$$

where $\theta_{model}$ is the parameter of the iterative optical generative model, $T$ is the total timestep in the denoising scheduler, and $\mathcal{O}_{model}(\mathcal{I}_t, t)$ is the output feature of the optical generative model predicted from the noised sample $\mathcal{I}_t$ and timestep $t$.

The iterative optical generative model consisted of a shallow digital encoder and an optical generative model, which worked jointly to generate high-quality images. In the training procedure, a batch of timesteps was sampled first, then the original data $\mathcal{I}_0$ was noised by the scheduler of timesteps to get $\mathcal{I}_t$. The noisy images, along with their corresponding timesteps, were fed into the digital encoder. It is noted that the timesteps were extra information, similar to the class labels used in snapshot optical generative models. As equation (7), the output intensity $\mathcal{O}_{model}(\mathcal{I}_t, t)$ was used to calculate the loss value for updating the learnable parameters.

In the inference stage, $\mathcal{I}_t$ started with Gaussian noises at timestep $T$, that is, $\mathcal{I}_T \sim \mathcal{N}(\mathbf{0}, \mathbf{I})$. After $\mathcal{I}_t$ passes through the digital encoder and the generative optical model, the resulting optical intensity image received on the image plane was normalized to $[-1, 1]$ range and then added with the designed noise:

$$\mathcal{I}_{t-1} = (\mathcal{O}_{model}(\mathcal{I}_t, t) - 0.5) \times 2 + \sigma_t z \tag{12}$$

where $\mathcal{O}_{model}(\mathcal{I}_t, t)$ is the normalized output intensity on the image sensor plane. $z \sim \mathcal{N}(\mathbf{0}, \mathbf{I})$ for $t > 1$, $z = 0$ when $t = 1$, and $\sigma_t^2 = (1 - \bar{\alpha}_{t-1})\beta_t/1 - \bar{\alpha}_t$. The measured intensity is perturbed by Gaussian noise with a designed variance, after which the resulting term, $\mathcal{I}_{t-1}$, is used as the optical generative seed at the next timestep. The iterative optical generative model was forwarded $T$ times, generating the final image $\mathcal{O}_{model}(\mathcal{I}_1, 1)$ on the image plane.

In the numerical implementations of the iterative optical generative models, two datasets for multicolour image generation were used separately: (1) Butterflies-100[17,60] and (2) Celeb-A[18]. The extrinsic parameters of the digital encoder and the optical generative model were similar to the snapshot optical generative models except for the number ($L_o$) of the decoding layers, which was set to 5. The distance between decoding layers $d_{l_o-1,l_o}$ was 20 mm. In the training of the iterative optical generative models, the total timestep $T$ was set to 1,000. $\beta_t$ is a linear function from $1 \times 10^{-3}/1 \times 10^{-3}$ ($t = 1$, Butterflies/Celeb-A) to $5 \times 10^{-3}/0.01$ ($t = T$, Butterflies/Celeb-A). The learnable parameters were optimized using the AdamW optimizer[56]. The learning rate for the digital parameters (digital encoder and DDPM) was $1 \times 10^{-4}$, and for the decoding layer was $2 \times 10^{-3}$, with a cosine annealing scheduler for the learning rate. The batch size was 200 for the iterative optical generative model. The models were trained and tested using PyTorch 2.21[57] with a single NVIDIA RTX 4090 graphics processing unit.

## Performance analysis of iterative optical generative models

We also investigated the influence of the number of diffractive layers and the performance limitations arising from potential misalignments in the fabrication or assembly of a multi-layer diffractive decoder trained for optical image generation. Our analysis revealed that the quality of iterative optical image generation without a digital encoder exhibits a degradation with a reduced number of diffractive layers. Supplementary Fig. 24 further demonstrates the scalability of the diffractive decoder: as the number of decoding layers increases, the FID score on the Celeb-A dataset drops, indicating the enhanced generative capability of the iterative optical generative model. Furthermore, as shown in Supplementary Fig. 25, lateral random misalignments cause a performance decrease in the image generation performance of multi-layer iterative optical models[63]. However, training the iterative optical generative model with small amounts of random misalignments makes its inference more robust against such unknown, random perturbations (Supplementary Fig. 25), which is an important strategy to bring resilience for implementing deeper diffractive decoder architectures in an optical generative model.

## Experimental set-up

The performance of the jointly trained optical generative model was experimentally validated in the visible spectrum. For the MNIST and Fashion-MNIST image generation (Fig. 3 and Extended Data Fig. 5), a laser (Fianium) was used for the illumination of the system at 520 nm. The laser beam was first filtered by a $4f$ system, with a 0.1-mm pinhole in the Fourier plane. Following the filtering, a linear polarizer was applied to align the polarization direction with the working direction of the SLM's liquid crystal. Then the light was modulated by the SLM (Meadowlark XY Phase Series; pixel pitch, 8 μm; resolution, 1,920 × 1,200) to create the encoded phase pattern, that is, the optical generative seed, $\phi(x, y, \lambda)$. For the reconfigurable diffractive decoder, another SLM (HOLOEYE PLUTO-2.1; pixel pitch, 8 μm; resolution, 1,920 × 1,080) was used to display the optimized $\phi^{(l_o)}(x, y, \lambda)$. After the diffractive decoder, we used a camera (QImaging Retiga-2000R; pixel pitch, 7.4 μm; resolution, 1,600 × 1,200) to capture the generated intensity of each output image $\mathcal{O}(x, y)$. The distance $d_{0,1}$ from the object plane to the optical decoder plane was 120.1 mm and the distance $d_{1,2}$ from the optical decoder plane to the sensor plane was 96.4 mm. The resolution of the encoded phase pattern, the decoding layer, and the sensor plane were 320 × 320, 400 × 400 and 320 × 320, respectively. After the image capture at the sensor plane, they were centre cropped, normalized and resized to the designed resolution. See Supplementary Videos 1–9 for the experimental images.

For the optical image generation experiments corresponding to monochrome Van Gogh-style artworks shown in Fig. 4 and Extended Data Fig. 6, the same set-up as in the previous experiments, was used, with adjustments made only to the resolution. The resolutions of the encoded phase pattern, the decoding layer and the sensor plane were 1,000 × 1,000, 1,000 × 1,000 and 640 × 640, respectively. For the multicolour artwork generation in, for example, Fig. 5, the same set-up was employed, with illumination wavelengths set to {450, 520, 638} nm, applied sequentially. All the captured images were first divided by the bit-depth of the sensor and normalized to [0, 1], and then we applied gamma correction ($\Gamma = 0.454$)[64] to adapt to human vision.

## Latent space interpolation experiments through a snapshot optical generative model

To explore the latent space of the snapshot optical generative model, we performed experiments to investigate the relationship between the random noise inputs and the generated images (Extended Data Fig. 5, Supplementary Fig. 26 and Supplementary Videos 3–9). As shown in Extended Data Fig. 5a, two random inputs $\mathcal{J}^1$ and $\mathcal{J}^2$ are sampled from the normal distribution $\mathcal{N}(\mathbf{0}, \mathbf{I})$ and linearly interpolated using the equation $\mathcal{J}^\gamma = \gamma \mathcal{J}^1 + (1-\gamma)\mathcal{J}^2$, where $\gamma$ is the interpolation coefficient. It is noted that the class embedding is also interpolated in the same way as the inputs. The interpolated input $\mathcal{J}^\gamma$ and the class embedding are then fed into the trained digital encoder, yielding the corresponding generative phase seed, which is fed into the snapshot optical generative set-up to output the corresponding image. Extended Data Fig. 5b shows the experimental results of this interpolation on the resulting images of handwritten digits using our optical generative set-up. Each row shows images generated from $\mathcal{J}^1$ (leftmost) to $\mathcal{J}^2$

(rightmost), with intermediate images produced by the interpolated inputs as $\gamma$ varies from 0 to 1. The generated images show smooth transitions between different handwritten digits, indicating that the snapshot optical generative model learned a continuous and well-organized latent space representation. Notably, the use of interpolated class embeddings demonstrates that the learned model realizes an external generalization: throughout the entire interpolation process, the generated images maintain recognizable digit-like features, gradually transforming one handwritten digit into another one through the interpolated class embeddings, suggesting effective capture of the underlying data distribution of handwritten digits. Additional interpolation-based experimental image generation results from our optical set-up are shown in Supplementary Fig. 26 and Supplementary Videos 3–9.

## Multiplexed optical generative models

We demonstrate in Extended Data Fig. 10 the potential of the optical generative model as a privacy-preserving and multiplexed visual information generation platform. In the scheme shown in Extended Data Fig. 10a, a single encoded phase pattern generated by a random seed is illuminated at different wavelengths, and only the correctly paired diffractive decoder can accurately reconstruct and reveal the intended information within the corresponding wavelength channel. This establishes secure content generation and simultaneous transmission of visual information to a group of viewers in a multiplexed manner, where the information presented by the digital encoder remains inaccessible to others unless the correct physical decoder is used (Extended Data Fig. 10b). This is different from a free-space-based image decoding, which fails to multiplex information channels using the same encoded pattern due to strong cross-talk among different channels, as shown in Extended Data Fig. 10c. By increasing the number of trainable diffractive features in a given decoder architecture proportional to the number of wavelengths, this privacy-preserved multiplexing capability can be scaled to include many wavelengths, where each unique decoder can only have access to one channel of information from the same or common encoder output. This secure multiplexing capability through diffractive decoders does not need dispersion engineering of the decoder material, and can be further improved by including polarization diversity in the diffractive decoder system. Without spatially optimized diffractive decoders, which act as physical security keys, a simple wavelength and/or polarization multiplexing scheme through free-space diffraction or a display would not provide real protection or privacy, as everyone would have access to the generated image content at a given wavelength and/or polarization combination.

Therefore, the physical decoder architecture that is jointly trained with the digital encoder offers naturally secure information processing for encryption and privacy preservation. As the encoder and the decoders are designed in tandem, and the individual decoders can be fabricated using various nanofabrication methods[52,53], it is difficult to reverse-engineer or replicate the physical decoders unless access to the design files is available. This physical protection and private multiplexing capability enabled by different physical decoders that receive signals from the same digital encoder is inherently difficult for conventional image display technologies to perform as they render content perceptible to any observer. For various applications such as secure visual communication to a group of users (for example, in public), anti-counterfeiting and personalized access control (for example, dynamically adapting to the specific attributes or history of each user), private and multiplexed delivery of generated visual content would be highly desired. Such a secure multiplexed optical generative model can also be designed to work with spatially partially coherent light by appropriately including the desired spatial coherence diameter in the optical forward model, which would open up the presented framework to, for example, light-emitting diodes.

## Energy consumption and speed of optical generative models

The presented optical generative models comprise four primary components: the electronic encoder network, the input SLM, the illumination light and the diffractive decoder, which are collectively optimized for image display. The electronic encoder used for MNIST and Fashion-MNIST datasets consists of three fully connected layers, and it requires 6.29 MFLOPs per image, with an energy cost of about 0.5–5.5 pJ FLOP$^{-1}$, resulting in an energy consumption of 0.003–0.033 mJ per image. This energy consumption increases to about 1.13–12.44 J and about 0.28–3.08 J per image for the Van Gogh-style artworks reported in Figs. 4 and 5 and Extended Data Fig. 6, respectively. The input SLM, with a power range of 1.9–3.5 W, consumes about 30–58 mJ per image at a 60-Hz refresh rate. This SLM-related energy consumption can be reduced to <2.5 mJ per image using a state-of-the-art SLM[65–67]. The diffractive decoder has a similar energy consumption if a second SLM is used; however, its contribution would become negligible if a static decoder (for example, a passive fabricated surface or layer) is employed. As for the illumination light, the energy consumption per wavelength channel can be estimated to be less than 0.8 mJ per image[68], which is negligible compared with other factors. If the generated images were to be digitized by an image sensor chip (for example, a 5–10 mega-pixel CMOS imager), this would also add an extra energy consumption of about 2–4 mJ per image. Consequently, the overall energy consumption for generated images intended for human perception—excluding the need for a digital camera—is dominated by the SLM-based power in lower-resolution image generation, whereas the digital encoder power consumption becomes the dominant factor for higher-resolution image generation tasks such as the Van Gogh-style artworks. In contrast, graphics processing unit-based generative systems using a DDPM model have different energy characteristics that are dominated by the diffusion and successive denoising processes (involving, for example, 1,000 steps). For example, the computational requirements for generating MNIST, Fashion-MNIST and Van Gogh-style artwork images using a digital DDPM model amount to approximately 287.68 GFLOPs and 530.26 TFLOPs, respectively, corresponding to about 0.14–1.58 J per image for MNIST and Fashion-MNIST and 265–2916 J per image for Van Gogh-style artworks. We also note that various previous works have focused on accelerating diffusion models to improve their inference speed and energy efficiency. For example, the denoising diffusion implicit model enabled content generation up to 20 times faster than DDPM while maintaining comparable image quality[69–71]. Under such an accelerated configuration, the estimated computational energy required for generating images using a digital denoising diffusion implicit model would be about 7–79 mJ per image for MNIST and Fashion-MNIST and 13.25–145.8 J per image for Van Gogh-style artworks. Furthermore, if the generated images must be displayed on a monitor for human perception, additional energy consumption is incurred—typically between about 13 mJ and 500 mJ per image at a 60-Hz refresh rate.

Overall, these comparisons reveal that if the image information to be generated will be stored and processed or harnessed in the digital domain, optical generative models would face additional power and speed penalties owing to the digital-to-analogue and analogue-to-digital conversion steps that would be involved in the optical set-up. However, if the image information to be generated will remain in the analogue domain for direct visualization by human observers (for example, in a near-eye or head-mounted display), the optical generative seeds can be pre-calculated with a modest energy consumption per seed, as detailed above. Furthermore, the static diffractive decoder surface can be fabricated using optical lithography or two-photon polymerization-based nanofabrication methods, which would optically generate snapshot images within the display set-up. This could enable compact and cost-effective image generators—such as 'optical artists'—by replacing the back-end diffractive decoder with

a fabricated passive surface. This set-up would allow for the snapshot generation of countless images, including various forms of artwork, using simpler local optical hardware. From the perspective of a digital generative model, for comparison, one could also use a standard image display along with pre-computed and stored images created through, for example, a digital DDPM model; this, however, requires substantially more energy consumption per image generation through the diffusion and successive denoising processes, as discussed earlier. Exploration of optical generative architectures using nanofabricated surfaces would enable various applications, especially for image and near-eye display systems, including head-mounted and wearable set-ups.

## Reporting summary

Further information on research design is available in the Nature Portfolio Reporting Summary linked to this article.

## Data availability

The authors declare that all data supporting the results of this study are available within the main text, Methods and Supplementary Information.

## Code availability

Deep learning models reported in this work used standard libraries and scripts that are publicly available in PyTorch. Training and testing codes can be found via Zenodo at https://zenodo.org/records/15446687 (ref. 72).

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

**Acknowledgements** A.O. acknowledges the support of the Volgenau Chair at UCLA and the V.M. Watanabe Award.

**Author contributions** A.O. and S.C. conceived of the research. S.C. performed the experiments, result analysis and statistical studies with help from Y.L., Y.W. and H.C. S.C. and A.O. prepared the paper, with contributions from Y.L., Y.W. and H.C. The research was supervised by A.O.

**Competing interests** A.O. and S.C. have a pending patent application on the presented technology filed through UCLA.

**Additional information**
**Correspondence and requests for materials** should be addressed to Aydogan Ozcan.

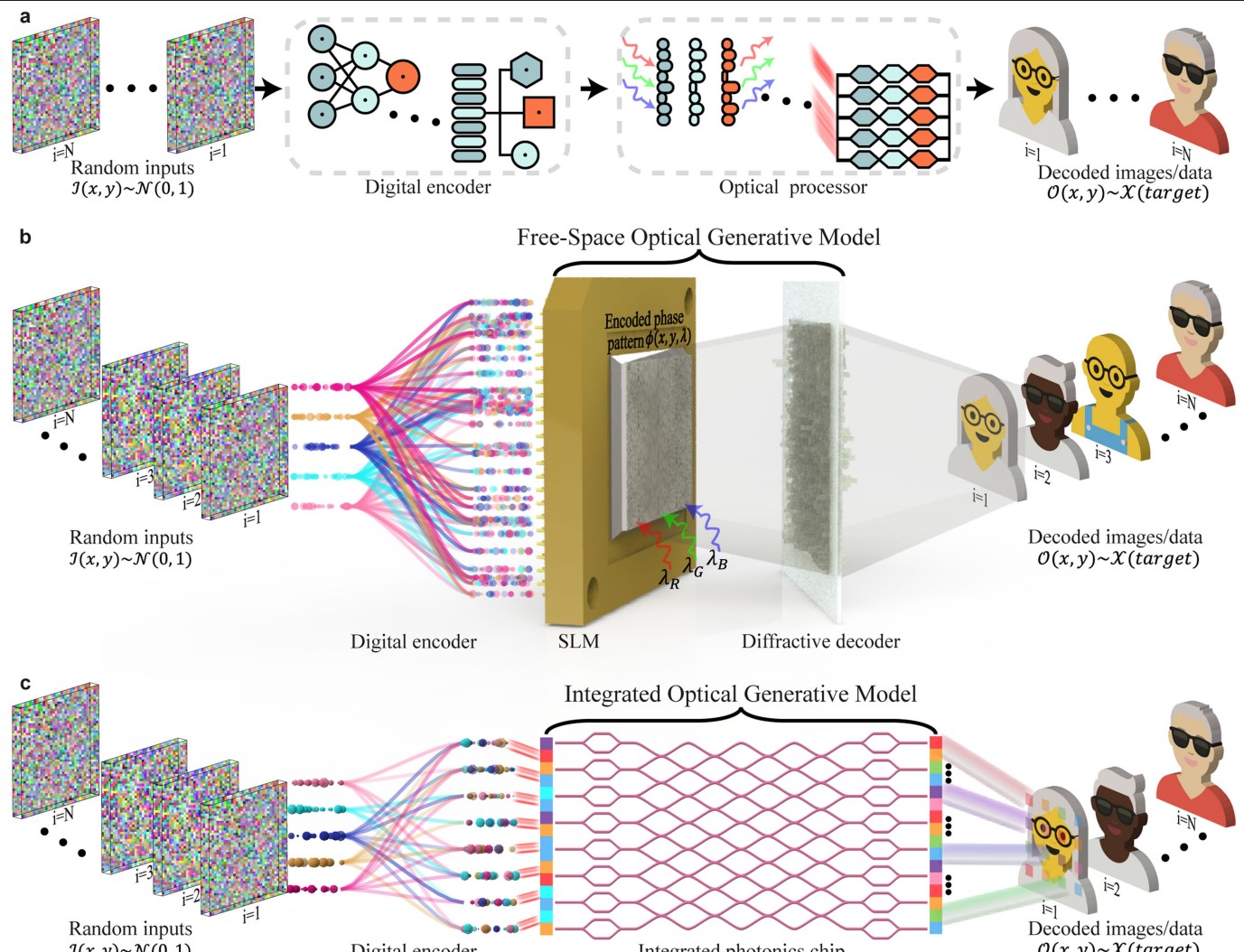

**Extended Data Fig. 1 | Illustration of optical generative models. a**, General concept of optical generative models. Random inputs sampled from Gaussian noise are sequentially processed by a shallow digital encoder and an optical processor, generating novel output images (never seen before) following a desired data distribution – for example, generating novel images of human faces. **b**, The design architecture of a free-space-based optical generative model. **c**, Illustration of an integrated photonics chip-based optical generative model.

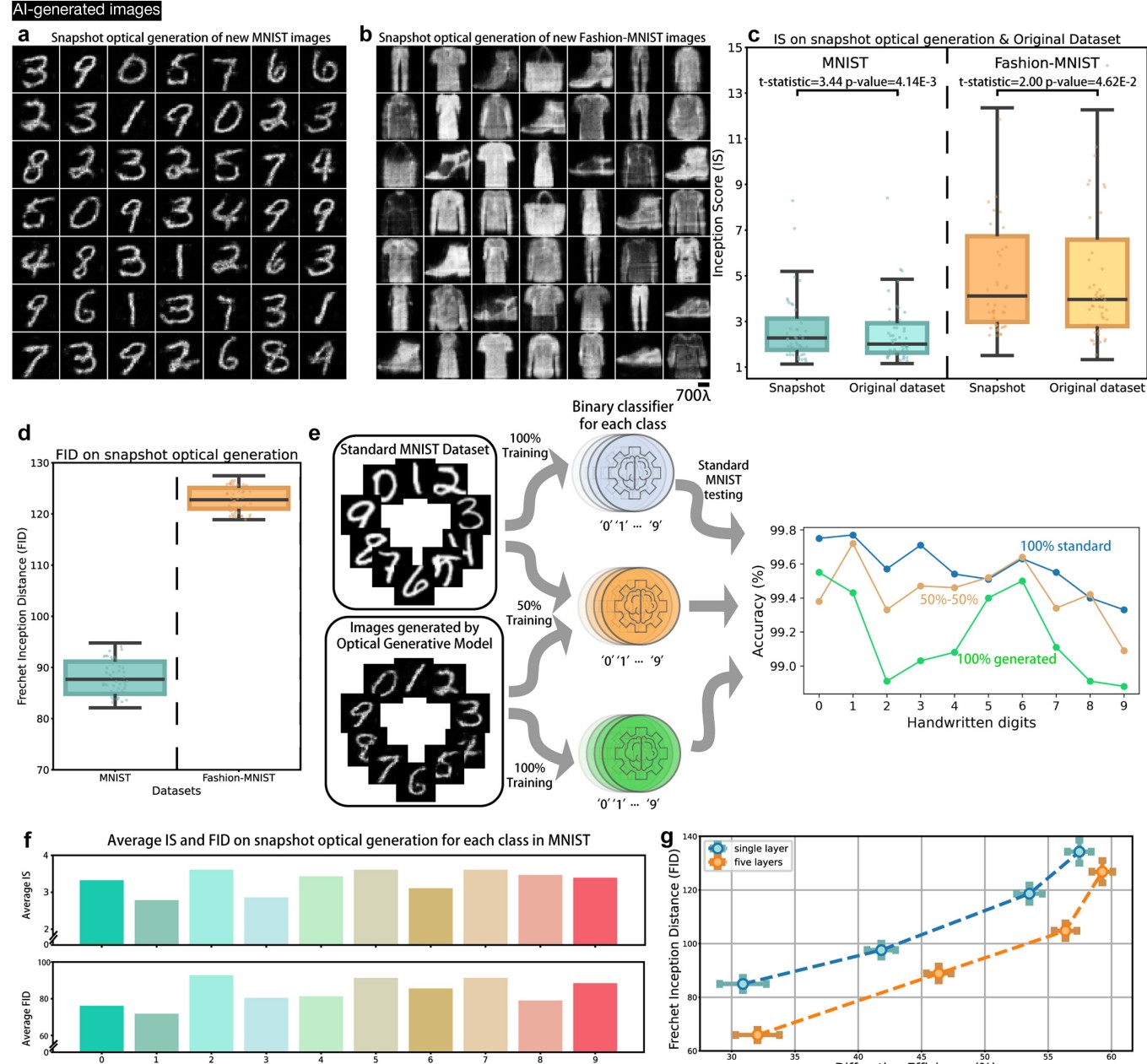

AI-generated images

**a** Snapshot optical generation of new MNIST images

**b** Snapshot optical generation of new Fashion-MNIST images

700λ

**c** IS on snapshot optical generation & Original Dataset

MNIST
t-statistic=3.44 p-value=4.14E-3

Fashion-MNIST
t-statistic=2.00 p-value=4.62E-2

Inception Score (IS)

Snapshot    Original dataset    Snapshot    Original dataset

**d** FID on snapshot optical generation

Frechet Inception Distance (FID)

MNIST    Fashion-MNIST
Datasets

**e** Standard MNIST Dataset

100% Training

Binary classifier for each class

Standard MNIST testing

'0' '1' ... '9'

Images generated by Optical Generative Model

50% Training

'0' '1' ... '9'

100% Training

'0' '1' ... '9'

Accuracy (%)

100% standard
50%-50%
100% generated

Handwritten digits

**f** Average IS and FID on snapshot optical generation for each class in MNIST

Average IS

Average FID

Handwritten digits

**g**

Frechet Inception Distance (FID)

single layer
five layers

Diffraction Efficiency (%)

**Extended Data Fig. 2 | Numerical performance evaluations of snapshot optical generative models.** This figure contains AI-generated images. **a**, Novel handwritten digit images (following the MNIST data distribution) generated by a snapshot optical generative model. **b**, Novel fashion product images (following the Fashion-MNIST data distribution) generated by a snapshot optical generative model. **c**, IS evaluation on snapshot image generation and the original target dataset, where the t-test results between two distributions are also reported. **d**, FID assessment of MNIST and Fashion-MNIST snapshot optical image generation processes. **e**, Classification accuracies of three different sets of 10 binary classifiers (one for each digit). The first set (blue curve in **e**) was trained using only the standard MNIST training data, the second set (orange curve in **e**)

was trained using a 50-50% mixed dataset composed of standard and optically generated image data, while the third set (green curve in **e**) was trained using 100% optically generated images. All of them were tested on the same test set of the standard MNIST dataset. The classifiers trained on 100% generated image data achieved 99.18% accuracy on average (green curve), showing a small decrease of 0.40% on average compared to the standard MNIST training (blue curve) across different digits. **f**, Average IS and FID metrics on snapshot optical image generation for each digit (0 to 9). **g**, Relationship between the FID scores and the output diffraction efficiency of optical generative models; one-layer vs. five-layer decoder-based snapshot optical image generation.

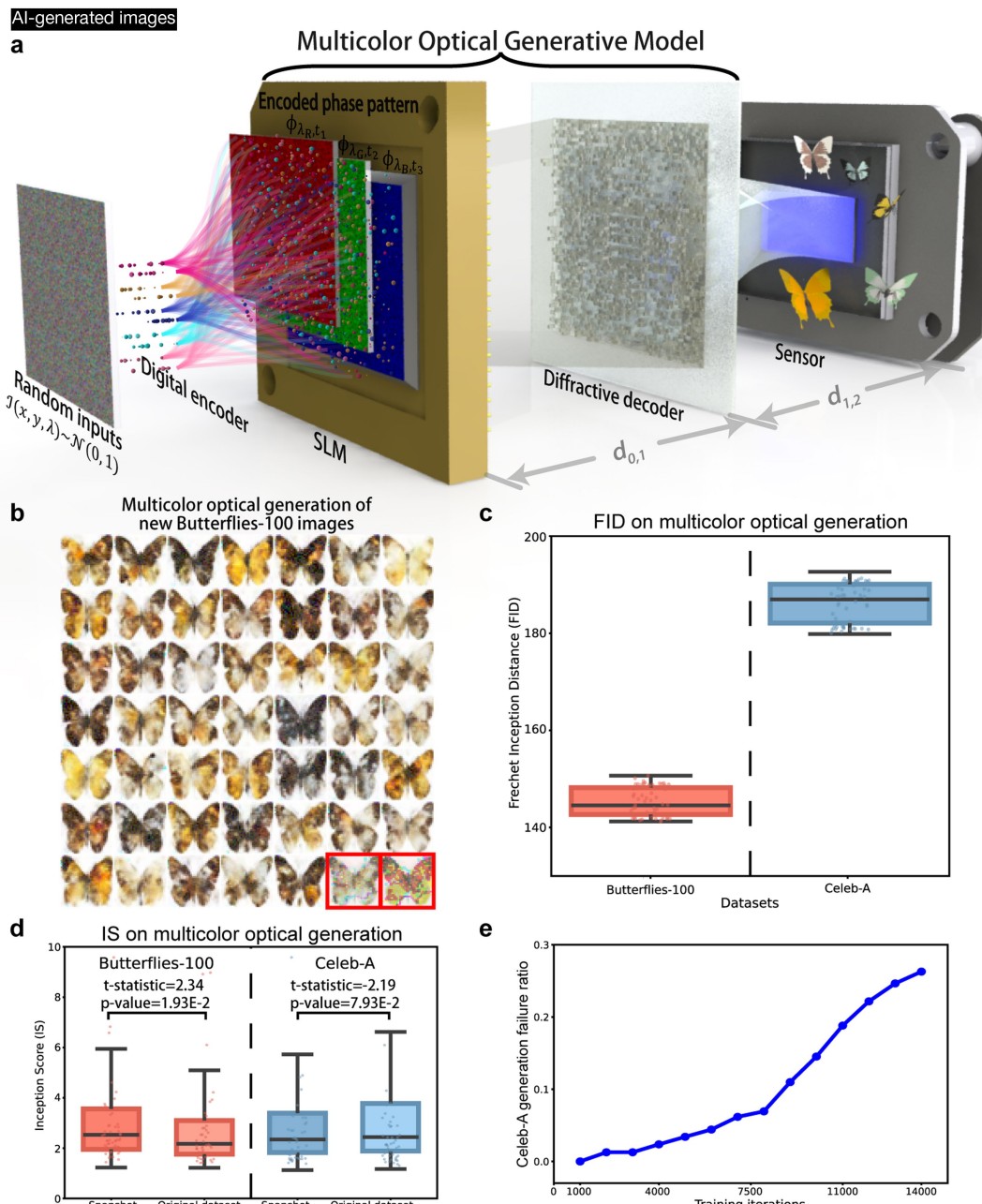

**Extended Data Fig. 3 | Numerical results of multi-color optical generative models.** This figure contains AI-generated images. **a**, Schematic of a multi-color optical generative model. **b**, Novel butterfly images (following the Butterflies-100 data distribution) generated by a multi-color optical generative model. We show some generation failure cases in the bottom-right of **b** with red frames. **c**, FID evaluation of multi-color optical generative models. **d**, IS evaluation of multi-color optical generative models against the original datasets, where the t-tests results between each pair of distributions are also listed. **e**, The ratio of image generation failure as the training on Celeb-A continues.

**a**  Iterative optical generation of new Buterflies-100 images

700λ

**b**  Intermediate results of iterative optical generation

Timestep:  999  799  599  399  299  199  99  59  39  19  0

**c**

FID on iterative optical generation

Frechet Inception Distance (FID)

Butterflies-100    Celeb-A    Celeb-A w/o Digital encoder

**d**

IS on iterative optical generation & original dataset

Inception Score (IS)

Butterflies-100
t-statistic=3.66
p-value=5.35E-4

Celeb-A
t-statistic=3.80
p-value=4.25E-2

t-statistic=1.89
p-value=3.21E-1

Iterative    Original dataset    Iterative    Original dataset    Iterative w/o. Digital encoder

**Extended Data Fig. 4 | Numerical performance evaluations of iterative optical generative models.** This figure contains AI-generated images. **a**, Novel butterfly images (following the Butterflies-100 data distribution) generated by an iterative optical generative model. **b**, Intermediate results of the iterative optical generative model at different timesteps. **c**, FID assessment on iterative optical generation of novel butterfly and human face images. **d**, IS comparisons of iterative optical generative model results and the original corresponding datasets (Butterflies-100 and Celeb-A). T-test results between each pair of distributions are also listed. The iterative optical generative models present higher IS values than the original datasets, which demonstrates that the optical models can generate more diverse images than the target data distributions.

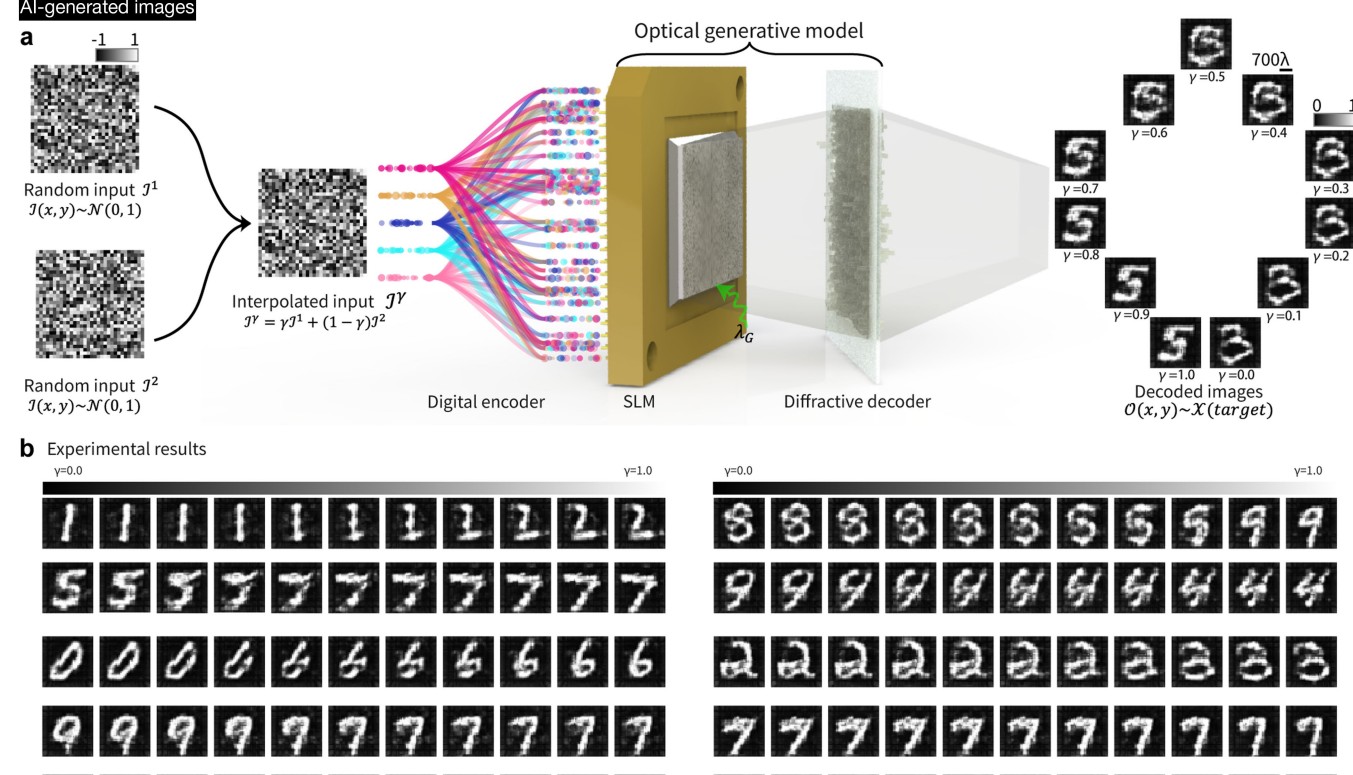

**Extended Data Fig. 5 | Experimental results of latent interpolation through a snapshot optical generative model.** This figure contains AI-generated images. **a**, We illustrate the latent interpolation carried out by a snapshot optical generative model, where two different random noise patterns (sampled from a normal distribution) and two class embeddings are first fused by weights, and then fed into the experimental optical generative model. The right part shows how the process of latent interpolation is controlled by the weights along with the interpolated class embeddings, gradually transforming one handwritten digit into another one. **b**, More experimental results of latent interpolations are shown. Also see Supplementary Fig. S26 and Supplementary Videos 3–9.

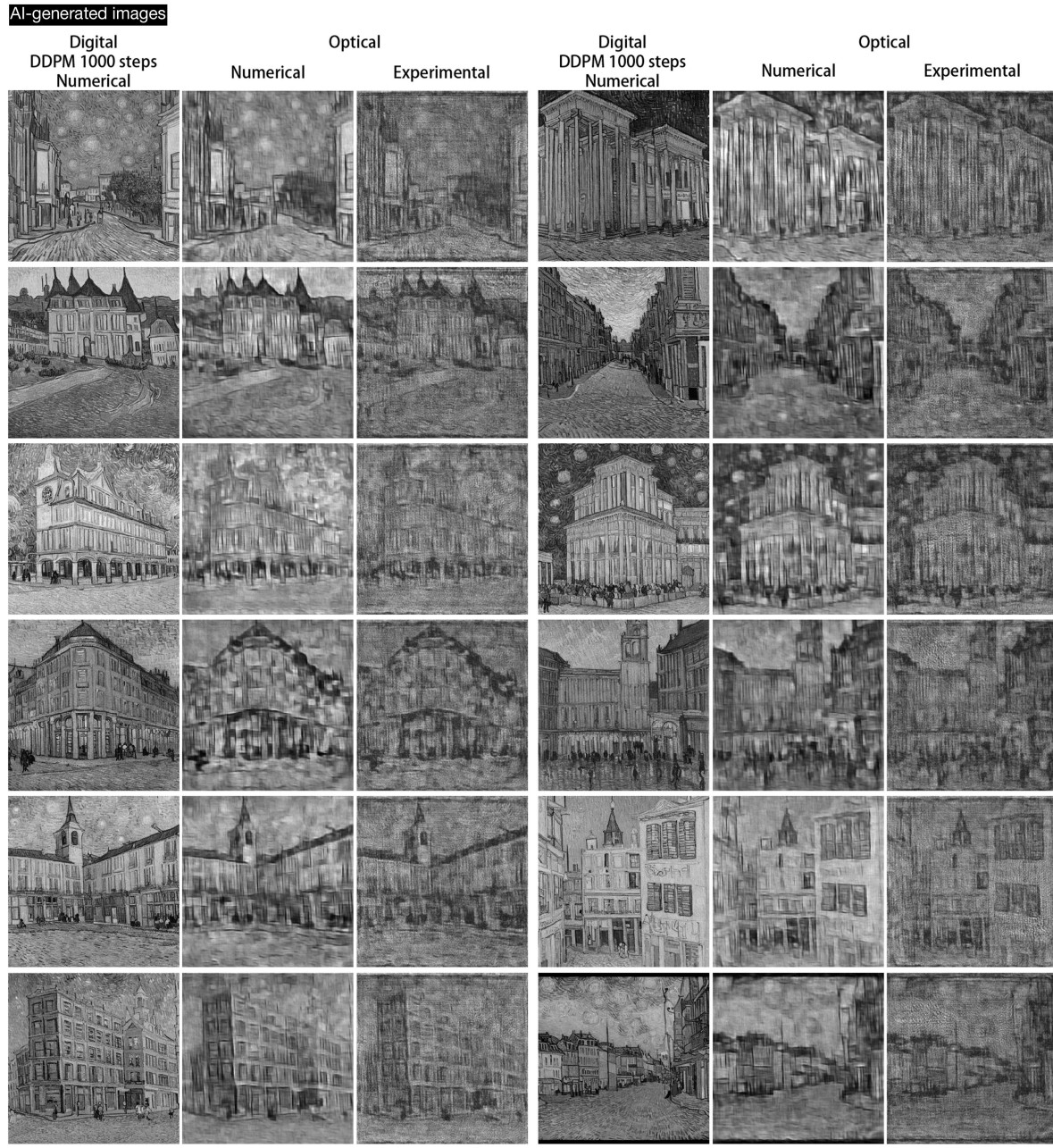

AI-generated images

| Digital DDPM 1000 steps Numerical | Optical | | Digital DDPM 1000 steps Numerical | Optical | |
|---|---|---|---|---|---|
| | Numerical | Experimental | | Numerical | Experimental |

1 mm   0 ▬▬ 1

**Extended Data Fig. 6 | Numerical and experimental results of a snapshot optical generative model for monochrome Van Gogh-style novel artwork generation compared against the teacher digital diffusion model with 1000 steps.** This figure contains AI-generated images. We present comparative results on monochrome Van Gogh-style novel artwork generation for both the digital teacher diffusion model (with 1.07 Billion trainable parameters and 1000 steps used for each inference) and the snapshot optical generative model, along with the experimental results for the snapshot optical generative model. The digital phase encoder has 85 M trainable parameters and each snapshot optical image is generated by a unique random noise input.

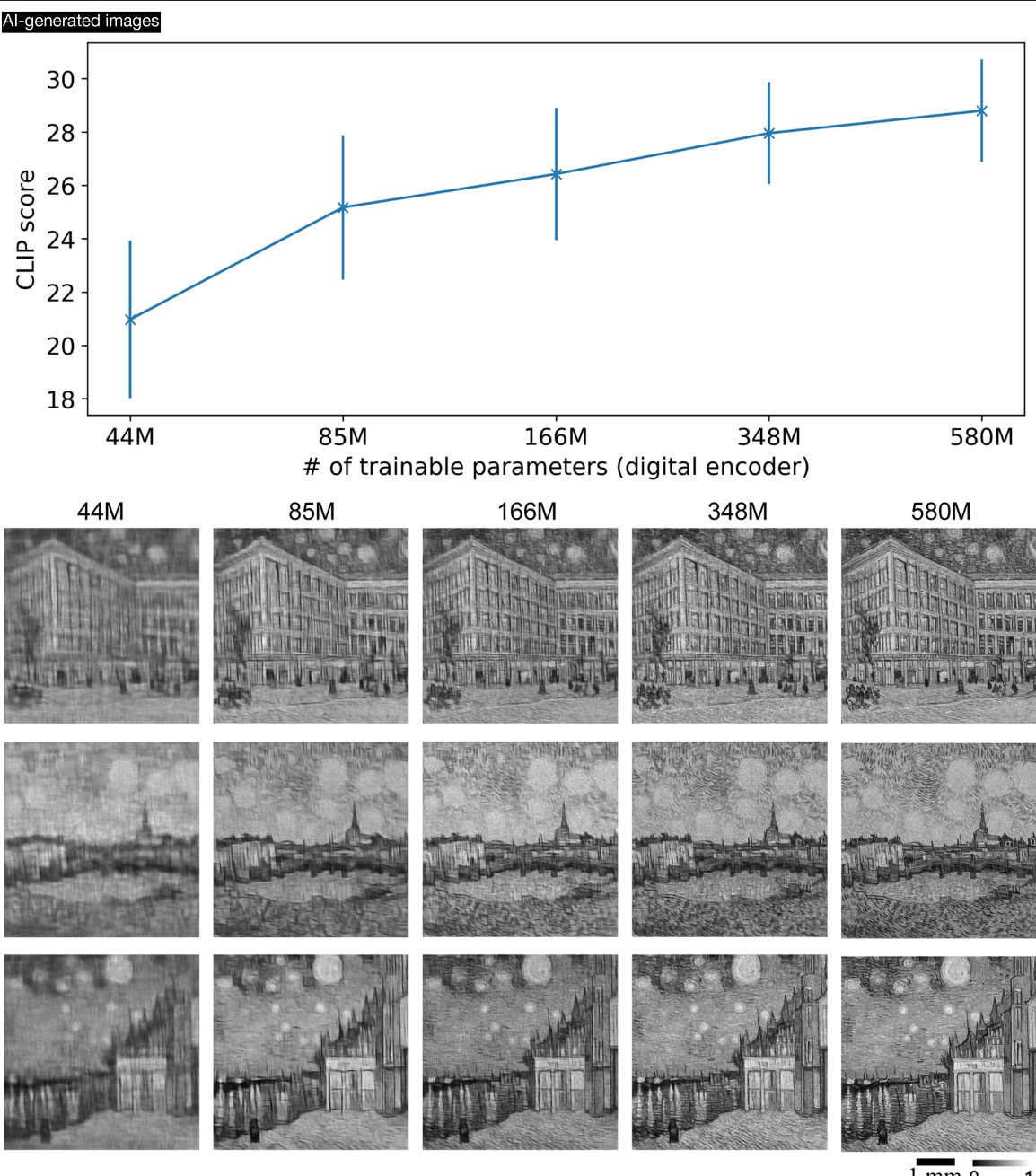

AI-generated images

**Extended Data Fig. 7 | CLIP score evaluation of the text-to-image alignment for Van Gogh style artworks created by the snapshot optical generative models with varying numbers of trainable parameters of the digital encoder.** This figure contains AI-generated images. The generated images are created by the snapshot optical generative models, and they are evaluated by calculating the contrastive language-image pre-training score (CLIP score), which quantifies the generated image alignment with respect to the reference text: "*Van Gogh style painting of architecture*".

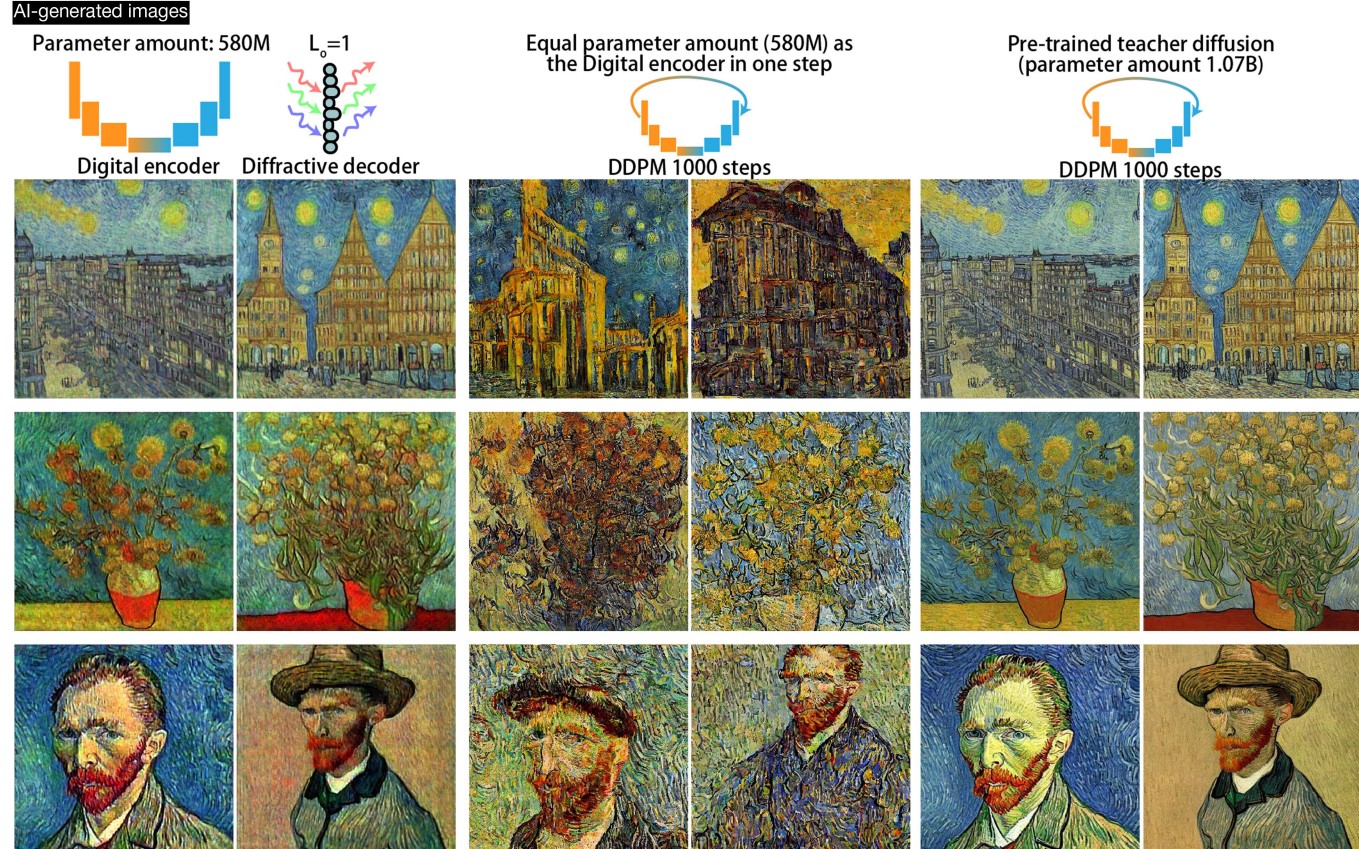

**Extended Data Fig. 8 | Generation of colorful Van Gogh-style artworks.**
This figure contains AI-generated images. Numerical simulation results of the multi-color optical generative model (580 M parameters) compared against a reduced-size diffusion model (shown in the middle) trained from scratch, whose U-Net has the same number of learnable parameters as the digital encoder in the optical generative model (580 M parameters), and a pre-trained teacher diffusion model (with 1.07B parameters) which is shown on the right. The average CLIP scores of the multi-color optical generative model, the reduced-size diffusion model, and the pre-trained teacher diffusion model are 28.25, 24.45, and 28.72, respectively (the reference text: "*Van Gogh style painting of {architecture, plants, person}*").

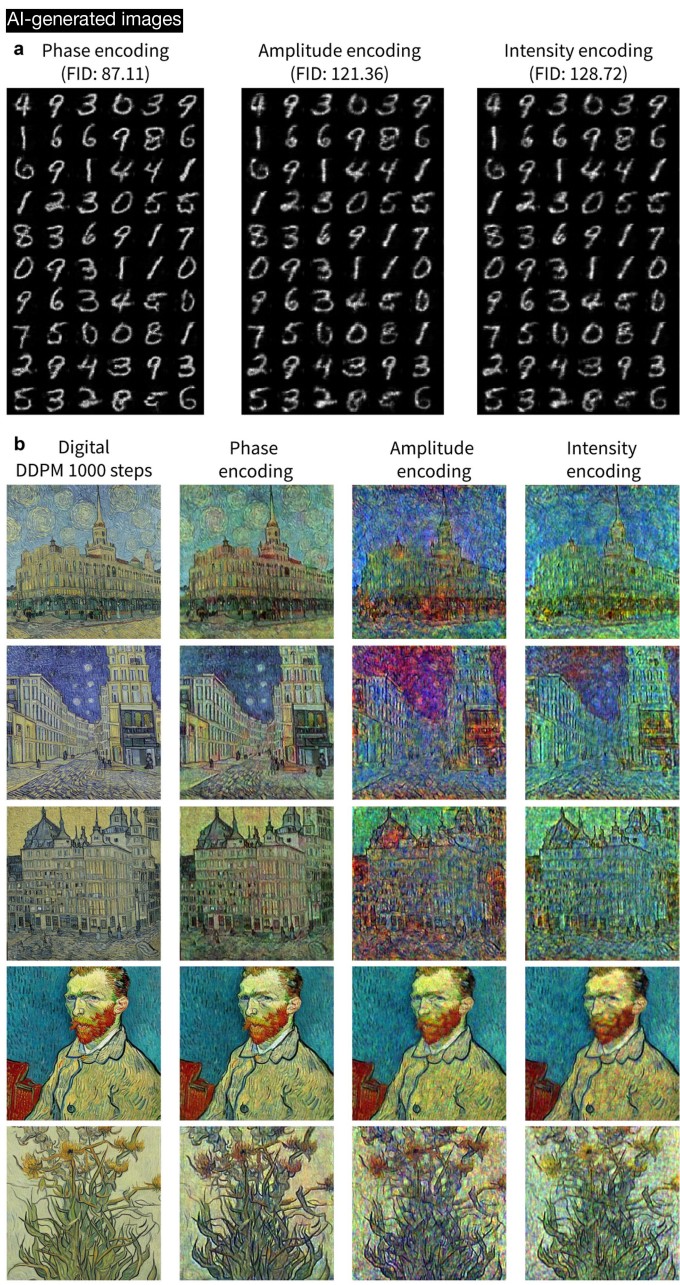

**a** Phase encoding (FID: 87.11)  Amplitude encoding (FID: 121.36)  Intensity encoding (FID: 128.72)

**b** Digital DDPM 1000 steps  Phase encoding  Amplitude encoding  Intensity encoding

**Extended Data Fig. 9 | Evaluation of different information encoding strategies for handwritten digits and colorful Van Gogh-style artwork generation using snapshot optical generation models.** This figure contains AI-generated images. **a**, Comparisons of different digital-to-optical information encoding methods are presented. The FID was calculated to quantify the quality and diversity of the image generation, where phase encoding achieves the best score. **b**, For Van Gogh-style artwork generation, phase encoding shows the best visual results, especially for scenes with sophisticated details.

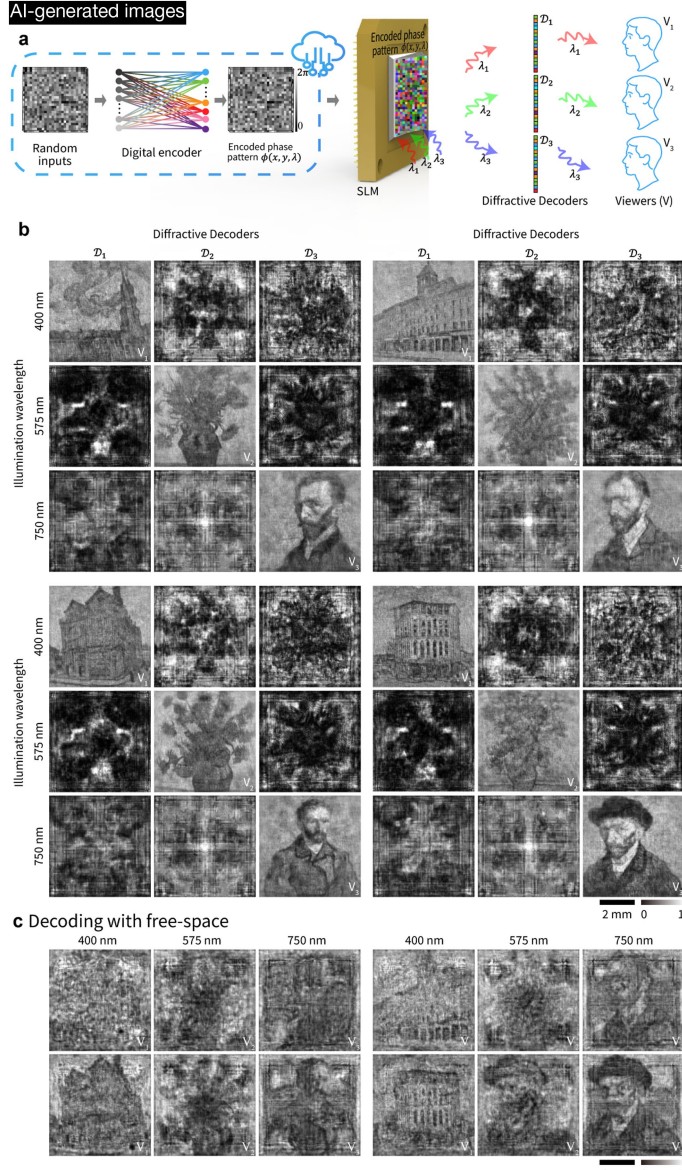

**Extended Data Fig. 10 | Optical generative models for private and multiplexed information generation.** This figure contains AI-generated images.
**a**, Demonstration of the privacy-preserving and multiplexed information/image projection using an optical generative model. A single encoded phase pattern generated from a random seed is illuminated with different wavelengths, and only the matching diffractive decoder can correctly reconstruct and reveal the intended information; with three unique diffractive decoders, each viewer can see a different optically generated image (*i.e.*, Van Gogh style paintings of architecture, plants, person) from the same encoded phase pattern. **b**, Confusion matrices showing the decoded images as perceived by each viewer ($V_1$, $V_2$ or $V_3$); a separate decoder ($D_1$, $D_2$ or $D_3$) is given to each viewer. Rows represent different illumination wavelengths, while columns correspond to different diffractive decoders, $D_1$, $D_2$, $D_3$. **c**, Free-space-based decoding (without a diffractive decoder) fails in multiplexed image generation. Furthermore, it does not enable protection or privacy of the displayed information since anyone can observe the resulting images at different wavelengths without the need for a physical decoder.

# Reporting Summary

## Statistics

For all statistical analyses, confirm that the following items are present in the figure legend, table legend, main text, or Methods section.

| n/a | Confirmed | |
|---|---|---|
| ☐ | ☒ | The exact sample size (*n*) for each experimental group/condition, given as a discrete number and unit of measurement |
| ☐ | ☒ | A statement on whether measurements were taken from distinct samples or whether the same sample was measured repeatedly |
| ☐ | ☒ | The statistical test(s) used AND whether they are one- or two-sided *Only common tests should be described solely by name; describe more complex techniques in the Methods section.* |
| ☒ | ☐ | A description of all covariates tested |
| ☒ | ☐ | A description of any assumptions or corrections, such as tests of normality and adjustment for multiple comparisons |
| ☐ | ☒ | A full description of the statistical parameters including central tendency (e.g. means) or other basic estimates (e.g. regression coefficient) AND variation (e.g. standard deviation) or associated estimates of uncertainty (e.g. confidence intervals) |
| ☐ | ☒ | For null hypothesis testing, the test statistic (e.g. *F*, *t*, *r*) with confidence intervals, effect sizes, degrees of freedom and *P* value noted *Give P values as exact values whenever suitable.* |
| ☒ | ☐ | For Bayesian analysis, information on the choice of priors and Markov chain Monte Carlo settings |
| ☒ | ☐ | For hierarchical and complex designs, identification of the appropriate level for tests and full reporting of outcomes |
| ☒ | ☐ | Estimates of effect sizes (e.g. Cohen's *d*, Pearson's *r*), indicating how they were calculated |

*Our web collection on statistics for biologists contains articles on many of the points above.*

## Software and code

Policy information about availability of computer code

| | |
|---|---|
| Data collection | MNIST, Fashion-MNIST, Butterflies-100, Celeb-A dataset, vangogh2photo datasets are open source image datasets. The experimental results were captured by a customized setup, consisting of a laser (Fianium), an SLM (HOLOEYE LC-R 2500), a PLM (DLP6750Q1EVM) and a camera (QImaging Retiga-2000R). Additional experimental results were also captured using another customized setup, using a laser, 2 SLMs (Meadowlark XY Phase Series, HOLOEYE PLUTO-2.1), and a camera (QImaging Retiga-2000R). |
| Data analysis | The neural networks were implemented and trained using Python 3.11.9 and PyTorch 2.21. The codes for our optical generative models can be found at: https://zenodo.org/records/15446687 |

For manuscripts utilizing custom algorithms or software that are central to the research but not yet described in published literature, software must be made available to editors and reviewers. We strongly encourage code deposition in a community repository (e.g. GitHub). See the Nature Portfolio guidelines for submitting code & software for further information.

# Data

Policy information about availability of data

All manuscripts must include a data availability statement. This statement should provide the following information, where applicable:

- Accession codes, unique identifiers, or web links for publicly available datasets
- A description of any restrictions on data availability
- For clinical datasets or third party data, please ensure that the statement adheres to our policy

> All data supporting the results of this study are available within the main text, Extended Data figures, the Supplementary Information and Supplementary Videos.
> MNIST: https://huggingface.co/datasets/ylecun/mnist
> Fashion-MNIST: https://huggingface.co/datasets/zalando-datasets/fashion_mnist
> Butterfly100: https://huggingface.co/datasets/huggan/smithsonian_butterflies_subset
> Celeb-A https://huggingface.co/datasets/nielsr/CelebA-faces
> vangogh2photo dataset: https://huggingface.co/datasets/huggan/vangogh2photo

# Research involving human participants, their data, or biological material

Policy information about studies with human participants or human data. See also policy information about sex, gender (identity/presentation), and sexual orientation and race, ethnicity and racism.

| | |
|---|---|
| Reporting on sex and gender | Not applicable |
| Reporting on race, ethnicity, or other socially relevant groupings | Not applicable |
| Population characteristics | Not applicable. |
| Recruitment | Not applicable |
| Ethics oversight | Not applicable |

Note that full information on the approval of the study protocol must also be provided in the manuscript.

# Field-specific reporting

Please select the one below that is the best fit for your research. If you are not sure, read the appropriate sections before making your selection.

☐ Life sciences ☐ Behavioural & social sciences ☐ Ecological, evolutionary & environmental sciences

For a reference copy of the document with all sections, see nature.com/documents/nr-reporting-summary-flat.pdf

# Life sciences study design

All studies must disclose on these points even when the disclosure is negative.

| | |
|---|---|
| Sample size | Describe how sample size was determined, detailing any statistical methods used to predetermine sample size OR if no sample-size calculation was performed, describe how sample sizes were chosen and provide a rationale for why these sample sizes are sufficient. |
| Data exclusions | Describe any data exclusions. If no data were excluded from the analyses, state so OR if data were excluded, describe the exclusions and the rationale behind them, indicating whether exclusion criteria were pre-established. |
| Replication | Describe the measures taken to verify the reproducibility of the experimental findings. If all attempts at replication were successful, confirm this OR if there are any findings that were not replicated or cannot be reproduced, note this and describe why. |
| Randomization | Describe how samples/organisms/participants were allocated into experimental groups. If allocation was not random, describe how covariates were controlled OR if this is not relevant to your study, explain why. |
| Blinding | Describe whether the investigators were blinded to group allocation during data collection and/or analysis. If blinding was not possible, describe why OR explain why blinding was not relevant to your study. |

# Behavioural & social sciences study design

All studies must disclose on these points even when the disclosure is negative.

| | |
|---|---|
| Study description | *Briefly describe the study type including whether data are quantitative, qualitative, or mixed-methods (e.g. qualitative cross-sectional, quantitative experimental, mixed-methods case study).* |
| Research sample | *State the research sample (e.g. Harvard university undergraduates, villagers in rural India) and provide relevant demographic information (e.g. age, sex) and indicate whether the sample is representative. Provide a rationale for the study sample chosen. For studies involving existing datasets, please describe the dataset and source.* |
| Sampling strategy | *Describe the sampling procedure (e.g. random, snowball, stratified, convenience). Describe the statistical methods that were used to predetermine sample size OR if no sample-size calculation was performed, describe how sample sizes were chosen and provide a rationale for why these sample sizes are sufficient. For qualitative data, please indicate whether data saturation was considered, and what criteria were used to decide that no further sampling was needed.* |
| Data collection | *Provide details about the data collection procedure, including the instruments or devices used to record the data (e.g. pen and paper, computer, eye tracker, video or audio equipment) whether anyone was present besides the participant(s) and the researcher, and whether the researcher was blind to experimental condition and/or the study hypothesis during data collection.* |
| Timing | *Indicate the start and stop dates of data collection. If there is a gap between collection periods, state the dates for each sample cohort.* |
| Data exclusions | *If no data were excluded from the analyses, state so OR if data were excluded, provide the exact number of exclusions and the rationale behind them, indicating whether exclusion criteria were pre-established.* |
| Non-participation | *State how many participants dropped out/declined participation and the reason(s) given OR provide response rate OR state that no participants dropped out/declined participation.* |
| Randomization | *If participants were not allocated into experimental groups, state so OR describe how participants were allocated to groups, and if allocation was not random, describe how covariates were controlled.* |

# Ecological, evolutionary & environmental sciences study design

All studies must disclose on these points even when the disclosure is negative.

| | |
|---|---|
| Study description | *Briefly describe the study. For quantitative data include treatment factors and interactions, design structure (e.g. factorial, nested, hierarchical), nature and number of experimental units and replicates.* |
| Research sample | *Describe the research sample (e.g. a group of tagged Passer domesticus, all Stenocereus thurberi within Organ Pipe Cactus National Monument), and provide a rationale for the sample choice. When relevant, describe the organism taxa, source, sex, age range and any manipulations. State what population the sample is meant to represent when applicable. For studies involving existing datasets, describe the data and its source.* |
| Sampling strategy | *Note the sampling procedure. Describe the statistical methods that were used to predetermine sample size OR if no sample-size calculation was performed, describe how sample sizes were chosen and provide a rationale for why these sample sizes are sufficient.* |
| Data collection | *Describe the data collection procedure, including who recorded the data and how.* |
| Timing and spatial scale | *Indicate the start and stop dates of data collection, noting the frequency and periodicity of sampling and providing a rationale for these choices. If there is a gap between collection periods, state the dates for each sample cohort. Specify the spatial scale from which the data are taken* |
| Data exclusions | *If no data were excluded from the analyses, state so OR if data were excluded, describe the exclusions and the rationale behind them, indicating whether exclusion criteria were pre-established.* |
| Reproducibility | *Describe the measures taken to verify the reproducibility of experimental findings. For each experiment, note whether any attempts to repeat the experiment failed OR state that all attempts to repeat the experiment were successful.* |
| Randomization | *Describe how samples/organisms/participants were allocated into groups. If allocation was not random, describe how covariates were controlled. If this is not relevant to your study, explain why.* |
| Blinding | *Describe the extent of blinding used during data acquisition and analysis. If blinding was not possible, describe why OR explain why blinding was not relevant to your study.* |

Did the study involve field work? ☐ Yes ☐ No

# Field work, collection and transport

| | |
|---|---|
| Field conditions | *Describe the study conditions for field work, providing relevant parameters (e.g. temperature, rainfall).* |
| Location | *State the location of the sampling or experiment, providing relevant parameters (e.g. latitude and longitude, elevation, water depth).* |
| Access & import/export | *Describe the efforts you have made to access habitats and to collect and import/export your samples in a responsible manner and in compliance with local, national and international laws, noting any permits that were obtained (give the name of the issuing authority, the date of issue, and any identifying information).* |
| Disturbance | *Describe any disturbance caused by the study and how it was minimized.* |

# Reporting for specific materials, systems and methods

We require information from authors about some types of materials, experimental systems and methods used in many studies. Here, indicate whether each material, system or method listed is relevant to your study. If you are not sure if a list item applies to your research, read the appropriate section before selecting a response.

### Materials & experimental systems

| n/a | Involved in the study |
|---|---|
| ☒ | ☐ Antibodies |
| ☒ | ☐ Eukaryotic cell lines |
| ☒ | ☐ Palaeontology and archaeology |
| ☒ | ☐ Animals and other organisms |
| ☒ | ☐ Clinical data |
| ☒ | ☐ Dual use research of concern |
| ☒ | ☐ Plants |

### Methods

| n/a | Involved in the study |
|---|---|
| ☒ | ☐ ChIP-seq |
| ☒ | ☐ Flow cytometry |
| ☒ | ☐ MRI-based neuroimaging |

## Plants

| | |
|---|---|
| Seed stocks | *Report on the source of all seed stocks or other plant material used. If applicable, state the seed stock centre and catalogue number. If plant specimens were collected from the field, describe the collection location, date and sampling procedures.* |
| Novel plant genotypes | *Describe the methods by which all novel plant genotypes were produced. This includes those generated by transgenic approaches, gene editing, chemical/radiation-based mutagenesis and hybridization. For transgenic lines, describe the transformation method, the number of independent lines analyzed and the generation upon which experiments were performed. For gene-edited lines, describe the editor used, the endogenous sequence targeted for editing, the targeting guide RNA sequence (if applicable) and how the editor was applied.* |
| Authentication | *Describe any authentication procedures for each seed stock used or novel genotype generated. Describe any experiments used to assess the effect of a mutation and, where applicable, how potential secondary effects (e.g. second site T-DNA insertions, mosiacism, off-target gene editing) were examined.* |

