## [Peer Review File · Nature]

Optical Generative Models

Corresponding Author: Dr Aydogan Ozcan

Version 0:

Reviewer comments:

Referee #1

(Remarks to the Author)

Authors proposed the concept of optical generative model, by which images can be generative in a snapshot. I understand the beauty of such a concept, especially given that authors provided multiple extensions of the core idea. However, I have several concerns:

1, The main body of the work is illustrations of the concept, which may be assisted by numerical calculations. Such an option greatly reduces the solidness of the work. Although in the end of the work, authors provided two experimental examples, the analysis regarding the performance is insufficient.

2, I guess this is a rush work that authors wanted to publish it ASAP. As a result, the structure of this paper is not well organized. There are many figures in this paper, but they turn to be similar that should be re-organized. Also, the writing should be compressed, as description of the network model and the performance is repeated several times throughout the paper.

3, Another concern lies in that, compared to the previous implementation of optical neural networks, what is new in this work? It looks like that they are all composed of several feed-forward layers. And if it is true, the credit should be purely given to diffusion model, and its hardware implementation is straightforward as previous works.

4, Regarding the model, I believe some concise equations explaining the details should be helpful to readers.

5, The diffractive decoder is the key in optical generative model. How did you prepare this decoder, and how much effort and cost are needed?

6, In all optical models, the optical-digital conversion should be considered when benchmarking the performance. Is it considered in this work? For instance, for the results in Fig. 4. BTW, the top-panel plot in Fig. 4 is really confusing, as all bars are the same for the two indexes.

7, When compared to electronic systems, the storage of parameters in memory is usually overlooked in optical computing systems. How should you consider this aspect, especially in the case of performance benchmarking?

8, Some references are not appropriate, e.g., those about emerging approaches for large generative AI models. Please check.

Referee #2

(Remarks to the Author)

In their manuscript Optical Generative Models, Chen et al report on the implementation of a digital-electronic / analog-optical generative network using off the shelf optical components. The results the authors report are indeed very surprising and fascinating in terms of generality. Using their concept, the authors demonstrate the generation of hand written digits, fashion items, butterflies and faces with high quality and robustness. The experimental results reported are sound, I do have some objections regarding claims made by the authors.

Regarding a recommendation I am undecided at the moment. The quality and novelty to the work contrasts with some of the claims and a lack of important analysis.

My major points:

1. Ultra fast inference time

The authors claim that inference time is extremely short – 1 ns. I strongly object to this claim. Firstly, this is only true for the optical propagation time and does not include (i) the implementation of the shallow encoder neither (ii) the response time of the SLM used to encode the injection phases and neither (iii) the camera frame rate. This would be like comparing the modulation speed of a transistor and ignoring the impact of the clock-network in an electronic chip. The contribution of (iii)

one might ignore since the image is created without the camera. However, (i) and (ii) are essential ingredients to image creation and hence have to be included. Hence, we are rather talking about 100ms.

2. Focus on FLOPs and Parameters for comparison

As another argument the authors claim that their system needs 3 times less FLOPs. However, FLOPs are only of indirect importance as they link to energy consumption. Considering a FLOP uses <1 pJ I think it is fair to assume that the SLM will use more power than one is able to safe.

3. Reason for the better performance in comparison to the digital decoder

The optical diffractive decoder implements a single layer with a linear matrix and an absolute square nonlinearity due to the photo-electric effect at the camera. This is a very simple operation – which is not necessarily a downside. However, I find it very confusing that using the electronic decoder the authors require more nonlinear layers to achieve the same results. I would like to ask the authors to compare to a single layer with the same amount of parameters as implemented by the phase-diffraction SLM and to compare these results with the optical decoder.

With these points I find that the technological relevance of the system is hard to justify for the highest level of publication targeted by the authors. However, I could be convinced if the authors make a clear case for their technology. Fundamentally speaking I find the work fascinating.

Minor comments and suggestions:

1. "This image generation process from the optical generative seed plane to the output image plane does not consume computing power except for the illumination wave.

I disagree, this needs the SLM too.

2. "To experimentally demonstrate the proof-of-concept[...]"

Starting from there till the end of the section I find the text quite repetitive from what was said before.

3. Figure. -> Figure?

Typo

4. and are projected onto a spatial light -> and are loaded onto a ...

5. a reconfigurable diffractive decoder

I suggest the authors remove jargon and simply call this another spatial light modulator. This is also true for the following paragraph. I would strongly prefer the authors call it here an SLM and the reconfigurable diffractive decoder in the sections that are more conceptual and less technical

6. p-values: what are p-values? I don't think they were introduced

7. Testing the accuracy of the optical output

In order to evaluate the validity of the newly optically generated data, the authors apply it to a network that was previously trained for MNIST classification and argue that the small drop in classification accuracy show that the generated data is representative of the original data.

I get the basic of the argument, and I would say it is kind of valid – yet it is a bit indirect. Why did the authors not take a generative adversarial approach? Did you try and what was the result?

8. 'on the original MNIST dataset exhibit some overfitting [...]"

I am not sure I agree with this line of argument. If the drop by 5.8% of the generated images is due to overfitting, then the same overfitting should happen using the original MNIST data. Otherwise this drop has to be associated to the fact that the generated data are not exactly as the original in their distribution?

9. Training of the electronic decoder

Did the authors train the decoder as well every time they trained an electronic decoder, or did they use the encoder optimized for optical decoding?

Referee #3

(Remarks to the Author)

A. Summary of the key results

This manuscript presents an optical neural network model implemented with a spatial light modulator (SLM) and phase light modulator (PLM), leveraging free-space light propagation for image generation.

The authors demonstrate novel image synthesis across several datasets and report a 3-fold reduction in floating-point operations (FLOPs) compared to fully-digital fully-connected (FC) neural networks of greater depth. They claim an inference time of less than 1 ns for generating a single 32×32 image. Additionally, the manuscript introduces the concept of multi-color and iterative optical generative models through numerical simulations.

The proposed architecture comprises a digital encoder that maps Gaussian noise to SLM phase patterns, followed by light propagation through an optimized diffractive optical layer (implemented with PLM) to produce images. The training strategy involves using a diffusion model to learn the data distribution, generating noise-image training pairs, and subsequently optimizing the digital encoder and diffractive layer in a supervised scheme.

B. Originality and significance

Conceptually, the model is very similar to the previous thread of work in diffractive optical neural network, e.g. as described

in this review:

*Hu, J., Mengu, D., Tzarouchis, D.C. et al. Diffractive optical computing in free space. Nat Commun **15**, 1525 (2024). <https://doi.org/10.1038/s41467-024-45982-w>*

Specifically, **Fig. 7: Hybrid optical-electronic networks for intelligent free-space processing** shows very similar model structure. All utilizing the combination of electronic neural network (FC perceptron) and optical neural network (diffractive layers).

As for significance, my primary concern lies in the framing of this work as “image generation.” The task presented here is more accurately described as learning a mapping from noise to images, with the hope of achieving some degree of generalization. While this is a valid machine learning task, it does not meet the standard expectations for true image generation in the context of modern generative models.

The diffusion model provides a theoretical foundation for approximating the target data distribution. However, the authors’ approach essentially adds another layer of approximation, this time using an end-to-end supervised training scheme with a basic FC perceptron.

The authors demonstrate advantages of the optical model in terms of reducing FLOPs for this specific task. However, this does not compensate for the lack of theoretical rigor and poor performance metrics. For example, for the MNIST dataset, the reported Fréchet Inception Distance (FID) scores in this study (~196 in Fig. 3d) are over an order of magnitude worse than those achieved by modern GAN models (~13 in table II):

Lazcano, D., Franco, N. F., & Creixell, W. (2021). Hgan: Hyperbolic generative adversarial network. IEEE Access, 9, 96309-96320.

See Sec. E for more questions about the claimed advantages regarding time performance, scalability, and energy efficiency.

C. Data & methodology: validity of approach, quality of data, quality of presentation

I think all results generated through numerical simulation should be explicitly labeled and clearly distinguished from experimental results. Are the results in Figs. 3, 5, and 7 based on digital simulations of the diffractive decoding behavior without hardware implementation? If it is the case, please label it; if not, then the section titled “Experimental demonstration of snapshot optical generative models” and the label in Fig. 8c as “Experimental snapshot generation” are confusing and misleading.

As for performance evaluation:

- **Fig. 3e***: The main text reported a 94.2% correctness, but the figure caption mentions “classification confidence,” yet the method of calculation is unclear. The authors should specify how this metric is defined .
- **Fig. 5f***: The criteria for defining “generation failure” are not sufficiently explained. While Fig. 5b and 5c qualitatively illustrate failure cases, some amount of color noise (not as strong) is also observed in the other generated samples. The main text does not clearly state how generation failure is defined, nor does it address whether these criteria are applied consistently. A quantitative criterion or threshold for failure should be provided for reproducibility and clarity.
- The claim in the main text that image inference time is less than 1 ns requires further justification. How was this value determined? Does it also include the time cost by the shallow digital encoder and SLM loading pattern, in addition to light propagation? Also, the A/D conversion time in the camera also potentially needs to be accounted for. If these additional factors were not explicitly discussed and addressed, the claim of <1 ns should be revised or clarified to avoid overstating the system’s performance.
- Is the FID score calculated against the original dataset used to train the DDPM? When you refer to “novel” images, are you comparing these to the data used to train the DDPM (denoising diffusion probabilistic model) or to the data generated by the DDPM that was subsequently used to train the optical generative model? I think it is essential to clearly separate the contributions of the DDPM and the optical generative model proposed in this work. If the optical generative model merely imitates the behavior of the DDPM and cannot generate new samples beyond the training samples provided by the DDPM, it would be misleading to describe it as a true generative model.

D. Appropriate use of statistics and treatment of uncertainties

The manuscript demonstrates an appropriate treatment of uncertainties. The supplementary material includes an ablation study and provides a discussion on the influence of the SLM bit depth and misalignment.

E. Conclusions: robustness, validity, reliability

Time Performance First, see the comment in Sec. C. Data & methodology to clarify the time performance. While the introduction acknowledges that the refresh rate is constrained by the frame rate of the spatial light modulator (SLM), this limitation is not addressed in the conclusions. For instance, the HOLOEYE LC-R 2500 used in the experiments operates at only 75 Hz, which is orders of magnitude slower than the claimed GHz inference time. For the iterative optical generative models that involve sampling 1,000 timestamps, the total computation time would be at least tens of seconds, excluding the additional overhead for data transmission between the camera and the digital encoder. The conclusions should accurately

reflect these constraints.

****Scalability:**** The conclusion of scalability is unsubstantiated given the limited scope of the demonstrations. The work is restricted to generating 32×32 images, which is far from representative of modern high-resolution image generation tasks. Furthermore, the scalability of the system is fundamentally constrained by the resolution and bit depth of the SLM and PLM. These hardware limitations significantly impact the system's ability to scale to larger image sizes or more complex tasks.

****Energy Efficiency:**** The claim of energy efficiency also requires reconsideration. While the optical component may theoretically operate with low power consumption, the overall system includes energy-intensive components such as the light source, SLM and camera. A more detailed analysis of the system's total power consumption, including both optical and electronic components, is necessary to validate this claim.

Additionally, although the main text does not state this explicitly, I suspect that the reported FID scores were measured only for the numerical simulations rather than the experimental results. This is suggested by Fig. 8, which shows a separate set of results labeled “experimental generation” without providing corresponding FID scores. If this is not the case, the authors should clarify this discrepancy, because the advantages of energy-efficiency and speed only hold true for the model implemented with real optics.

F. Suggested improvements: experiments, data for possible revision

Many suggestions for improvement have been provided in the earlier sections. The most critical issue is the last point in Section C—validating the capability of the proposed model to generate truly novel images by clearly separating its contribution from the pre-trained DDPM.

One additional point to consider: In Fig. 4, the noticeably low contrast in the results of the optical generative models compared to fully digital models may worth further exploration. Is this contrast limitation an inherent drawback of the system caused by the zero-order effect in diffractive optics? If so, it would be valuable to explicitly address this in the text and discuss any potential strategies to mitigate this issue.

**G. References: appropriate credit to previous work?***

The references seem appropriate, and no significant omissions were noted.

**H. Clarity and context: lucidity of abstract/summary, appropriateness of abstract, introduction and conclusions**

The abstract, introduction, and conclusions are generally clear and appropriately contextualized.

(Remarks on code availability)

The code simply requires the PyTorch framework and is reproducible. It provides a sample that utilizes the trained parameters to simulate the behavior of the proposed optical generative model. It does not include the implementation for training the DDPM from dataset and use DDPM to train the optical generative model. It remains unclear that how many noise/image pairs are required and what the computational overhead is for training this model. The authors should either provide the corresponding code or elaborate the training parameters in the manuscript.

Version 1:

Reviewer comments:

Referee #1

(Remarks to the Author)

I appreciate the authors' continued efforts on this work, including the addition of more experimental results and comprehensive discussions. However, several concerns remain:

Authors claimed scalability of this optical system. However, the parameter scale considered in this paper is only in the millions, which is much smaller than the commonly used generative AI model. How may it scaled up? In particular, authors have supplemented more experimental results, which, however, are apparently worse than the numerical results, e.g., for Van Gogh-style artworks. These findings suggest that inherent analog noise in the optical system may limit its scalability. The authors should address this issue explicitly.

Regarding the experimental results, the authors should also provide accuracy loss metrics, as done in the numerical section, to allow for better comparison between numerical, digital, and optical outcomes.

Regarding the cost for diffractive decoder, I meant the training cost. This aspect has not been discussed. This generative optics system is only used for inference, but what about the training process? It seems to be hidden, like a lot of accelerator work, but I think the training cost should be explicitly presented, especially when the system is to be scaled up.

Referee #2

(Remarks to the Author)

I would like to thank the authors for the extensive clarifications that are very transparent.

I am finally leaning now towards recommending publication. I think the technological relevance is given for some particular potential applications, not in the context of general image generation. However, the experiment illustrates in a beautiful manner what unconventional systems and optics in particular are capable of implementing. As such I expect the work to draw significant attention from a broad community.

Referee #3

(Remarks to the Author)

Thanks to the authors for carefully addressing the previous comments. The paper has improved significantly over the earlier version, demonstrating the proposed optical neural network as a properly and successfully distilled variant of a diffusion model for image generation tasks. I appreciate the high-resolution Van Gogh image generation demonstration, and the additional details on training and evaluation make the work more robust. It is encouraging to see that the proposed model can indeed generate novel images that are meaningfully different from those produced by the teacher model, and the empirical validation of the phase encoding strategy adds strength to the contribution. I now find the method itself valid; however, I remain somewhat unconvinced about its practical advantages or necessity.

Fundamentally, even if we acknowledge that the distilled network structure is efficient, one must ask: if the ultimate goal is to digitally generate novel images, why not simply use the numerical version without implementing it physically with a laser and SLM? The phase encoding can be simulated digitally and integrated into the network structure. The FLOPs would be nearly identical (given that the digital encoder has ~100M params and the diffractive layer only has ~1M), and without relying on the physical SLM, the inference frame rate could actually be higher. In that case, the energy consumption would also remain comparable.

Viewed this way, the core value of the paper becomes the proposal of a network architecture with a physical interpretation to approximate or accelerate diffusion-based generative models. There is already a substantial body of literature on accelerating diffusion models, and I believe it is necessary to acknowledge these efforts and, ideally, compare the proposed approach with them—particularly if the framing is focused on inference speed or energy efficiency.

Overall, the concept is interesting and the current demonstration is valid. However, I remain concerned that its practical value is limited, as it lacks compelling advantages over existing generative models. The need for a physical optical layer is not fully justified, especially given that the use of diffractive layers in optical neural networks is already a well-established technique. Without a clearer argument for why physical implementation is necessary or superior, adapting this structure to a new application may fall short of the novelty and impact expected for a top-tier journal.

Furthermore, some of the comparisons raise logical concerns. Specifically, in Figs. S18–S19, the authors state:

> “Models without an optical decoder (i.e., utilize free-space propagation only) fail to generate novel images despite the presence of the digital diffusion model-based teacher and the same digital encoder, which demonstrates that the diffractive decoder surface plays a vital role in improving the visual quality of the snapshot generated novel optical images.”

First, in Fig. S18, I wonder why Design II produces a completely gray output in the first row, yet manages to generate some meaningful structure in the second row? The comparison is also not exactly fair, because the two models do not have equivalent capacity. While both designs use the same digital encoder architecture, Design I includes an optimizable diffractive decoder layer, whereas Design II lacks additional trainable component.

Fig. S25's conclusion regarding is also questionable. There is no guarantee that a reduced-size diffusion model will learn the same noise-to-image mapping as the larger teacher model. Since the smaller model is trained from scratch, rather than distilled from the full model in an end-to-end manner, it is unsurprising that it fails to match the output distribution of the original. To support the claims made in this section, quantitative metrics such as FID scores should be reported. In fact, based on visual inspection alone, the results from the reduced-size DDPM do not appear significantly degraded to me.

(Remarks on code availability)

Pretty similar to the last submission if I remember correctly. Provided checkpoint and test code but not training.

Version 2:

Reviewer comments:

Referee #1

(Remarks to the Author)

I appreciate and thank the authors for the extensive supplementary work they have done, and I believe the work is worthy of publication. However, I hope the authors can pay attention to their wording and remain objective. The CLIP score obtained in the experiment is significantly lower than that of the numerical calculation and the lightweight numerical calculation, rather than demonstrating "high alignment".

Referee #3

(Remarks to the Author)

Thanks to the authors for the clarifications. The reduction in FLOPs makes sense, it is intuitive that using a FC structure leads to a more compact generative neural network with fewer parameters, and that the optical implementation reduces the computational overhead of the FC layers. The methodology now presents a compelling story to me. (though I imagine the training process must be very costly given the FLOPs argument?)

The potential of the optical generative model as a platform for privacy-preserving and multiplexed visual information generation is also quite interesting, and I appreciate the authors' efforts in conducting such experiments.

My only remaining question is about the evaluation metrics: why is the CLIP score used for the Van Gogh dataset, rather than standard metrics like FID or IS as used for other datasets? Also, was any comparison made against the teacher diffusion model in terms of FID or IS? As far as I know, CLIP was originally introduced for text-to-image generation, where a target text description is provided. However, in this paper all generative models are performing unconditional generation without text guidance. While I agree it is a valid demonstration that the generated images align with the intended semantic content, the use of CLIP score alone feels unusual to me. I wonder is there a specific reason for not using standard metrics like FID or IS for the Van Gogh dataset, as was done for MNIST and CelebA in Extended Data Figs. 2?

Overall, I believe this work is suitable for publication, provided that the above points are addressed. Finally, I appreciate the authors for releasing the training code.

We sincerely thank the referees for their reviews and the constructive feedback that we have received on our manuscript “**Optical Generative Models**” submitted to *Nature* (Manuscript ID: 2024-09-20762).

As detailed below, we have revised our manuscript in response to the reviewers’ comments. The original referee comments are shown in black color, whereas for ease of communication, our answers are provided in blue. Our revisions have also been marked in the main text and supplementary information files using yellow highlighting.

Summary of our Revisions:

To address the questions raised by the reviewers and significantly enhance the clarity, depth, and impact of your manuscript, we have conducted additional analyses and image generation experiments. At a top level, we have made the following revisions, the details of which will be provided in our specific responses to each referee comment:

- **Higher Resolution Novel Artwork Generation:** The manuscript now includes experiments on generating higher resolution, *Van Gogh-style* monochrome and multicolor images. These results, reported in several figures as listed below, showcase the model's ability to generate detailed and aesthetically pleasing novel artworks using optical generative models. The image generation results of the optical model exhibit high-quality and valid semantic information while differing meaningfully from the teacher diffusion model's outputs. These experimental and numerical results demonstrate that our framework achieves true image generation beyond a mere imitation of the digital teacher diffusion model.
- **Additional Analyses and Experiments:** We have also added improved experimental results on MNIST and Fashion-MNIST datasets, including detailed numerical and experimental FID scores that validate the robustness of your approach.
- **New Figures and Supplementary Information:** Several new main text figures and supplementary figures, as well as supplementary videos, have been added to the manuscript to provide visual evidence and detailed quantitative comparisons of our model's performance against digital models; see the complete list of these new figures and videos below.
- **Revised Text Sections and existing Figures:** We have made substantial revisions to the Abstract, Introduction, Results, Discussion, and Methods sections, along with the Supplementary Information and several existing figures, to incorporate the new data and clarify the novel contributions and significance of our work. We have also clarified misunderstandings and eliminated potentially misleading arguments in our manuscript by revising them to reflect a more accurate and balanced description of our results and analyses.
- **Improved Model Descriptions:** The manuscript now includes more concise and clear descriptions of the network model, including the diffusion model training details and equations moved to supplementary information for clarity.

As a quick summary, the following items have been revised and added to our manuscript's main text and SI, highlighted in yellow:

Changes to text:

- Abstract
- Introduction
- Results
- Discussion
- Methods
- Supplementary Information

New Figures Added:

- **Figure 9.** Numerical and experimental results of a snapshot optical generative model for monochrome Van Gogh-style novel artwork generation compared against the teacher digital diffusion model with 1000 steps
- **Figure 10.** Numerical and experimental results of a higher resolution snapshot optical generative model for monochrome Van Gogh-style novel artwork generation compared against the teacher digital diffusion model with 1000 steps
- **Figure 11.** Numerical and experimental results of a multi-color optical generative model for colorful Van Gogh-style novel artwork generation, compared against the teacher digital diffusion model with 1000 steps
- **Supplementary Fig. S2.** Performance comparison of snapshot optical generative models against digital-only GAN-based generative models trained on MNIST dataset.
- **Supplementary Fig. S3.** Performance comparison of MNIST snapshot optical generative models against digital-only generative models guided by diffusion.
- **Supplementary Fig. S4.** Performance comparison of Fashion-MNIST snapshot optical generative models against digital-only generative models guided by diffusion.
- **Supplementary Fig. S5.** Performance comparison of MNIST snapshot optical generative models against DDPM-based generative models
- **Supplementary Fig. S7.** Performance investigation of snapshot optical generative models as a function of the SLM phase range.
- **Supplementary Fig. S9.** The impact of limited phase modulation levels on snapshot optical generative models.
- **Supplementary Fig. S10.** Determination of image generation failures using noise variance.
- **Supplementary Fig. S14.** Experimental demonstration of snapshot optical generative models with a limited phase range and decoder bit depth
- **Supplementary Fig. S15.** Different views of the optical generative model set-up with a limited phase range and decoder bit depth used in Supplementary Fig. S14
- **Supplementary Fig. S17.** The simulation pipeline for higher resolution snapshot optical image generation
- **Supplementary Fig. S18.** Comparison of diffractive decoder and free-space decoder on Van Gogh style artwork generation.

- **Supplementary Fig. S19. Comparison of diffractive decoder and free-space decoder on Van Gogh style artwork generation.**
 - **Supplementary Fig. S20. CLIP score evaluation of the text-to-image alignment for Van Gogh style artworks created by the snapshot optical generative models with varying numbers of trainable parameters of the digital encoder.**
 - **Supplementary Fig. S21. Numerical and experimental results of a higher resolution snapshot optical generative model for monochrome Van Gogh-style novel artwork generation compared against the teacher digital diffusion model with 1000 steps**
 - **Supplementary Fig. S22. Numerical and experimental results of a higher resolution snapshot optical generative model for monochrome Van Gogh-style novel artwork generation compared against the teacher digital diffusion model with 1000 steps.**
 - **Supplementary Fig. S23. Numerical and experimental results of a multi-color optical generative model for colorful Van Gogh-style novel artwork generation, compared against the teacher digital diffusion model with 1000 steps.**
 - **Supplementary Fig. S24. Numerical and experimental results of a multi-color optical generative model for colorful Van Gogh-style novel artwork generation, compared against the teacher digital diffusion model with 1000 steps.**
 - **Supplementary Fig. S25. Generation of colorful Van Gogh-style artworks**
 - **Supplementary Fig. S26. Evaluation of different information encoding strategies for handwritten digits and colorful Van Gogh-style artwork generation using snapshot optical generation models.**
 - **We have also added Supplementary Videos 1-9**
-

Referee #1:

Authors proposed the concept of optical generative model, by which images can be generative in a snapshot. I understand the beauty of such a concept, especially given that authors provided multiple extensions of the core idea.

-- We sincerely thank the referee for the positive assessment and constructive, valuable feedback, which helped us to enhance the quality and clarity of our manuscript.

However, I have several concerns:

1, The main body of the work is illustrations of the concept, which may be assisted by numerical calculations. Such an option greatly reduces the solidness of the work. Although in the end of the work, authors provided two experimental examples, the analysis regarding the performance is insufficient.

-- We thank the referee for the valuable suggestion regarding the inclusion of additional analyses and experimental results to strengthen the manuscript further. Accordingly, we have updated our revised manuscript with several new figures in our main text and supplementary information file, as summarized on Pages 1-3 of this Response Letter. For example, we reported new experimental results on the MNIST and Fashion-MNIST datasets, as shown in the **updated Fig. 7**. Additionally, we measured 1000 images for each dataset and incorporated the experimental FID metric to quantitatively evaluate our

image generation quality. We achieved experimental FID scores of 131.08 and 180.57 on the MNIST and Fashion-MNIST datasets, validating the robustness of our approach.

To further extend the demonstration of our model's capabilities, we conducted a comprehensive evaluation on generating Van Gogh-style higher-resolution images (i.e., **novel artwork**) using the proposed optical generative models. We first generated monochrome high-resolution images in the Van Gogh style, as illustrated in the **newly added Figs. 9 and 10 (as well as Supplementary Figures S21-S22)**. Furthermore, as shown in the **newly added Fig. 11 (as well as Supplementary Figures S23-S24)**, we also conducted experiments for the generation of color images in the Van Gogh style, validating its ability to generate high-resolution multi-color images. These new experimental results collectively provide a more comprehensive evaluation of the generation capabilities of our optical generative models.

Accordingly, these new results have been added to the **Abstract, Introduction section** and **Results section** of our revised manuscript:

*“...**Figure 7c** visualizes our experimental results for both models, which achieved experimental FID scores of 131.08 and 180.57 on the MNIST and Fashion-MNIST datasets, respectively.”*

“...To experimentally demonstrate optical generative models, we used visible light to generate novel images of handwritten digits and fashion products. Additionally, we generated Van Gogh-style novel artworks using both monochrome and multi-wavelength illumination.”

“...To experimentally demonstrate the proof-of-concept for snapshot and multi-color optical generative models, we built free-space hardware operating in the visible spectrum. These optical generative models synthesized novel images on-demand, including handwritten digits (MNIST), fashion products (Fashion MNIST), and Van Gogh-style novel artworks. Our experimental results confirmed that the learned optical generative models successfully grasped the underlying features and relationships within each target data distribution.”

*“...We further extended our experimental results for the optical generative models to create higher resolution images in the style of Van Gogh artworks, which were also demonstrated using the same setup shown in **Fig. 7b**. As illustrated in **Fig. 9**, we experimentally demonstrated snapshot monochrome image generation for Van Gogh-style novel artworks using a digital encoder paired with a jointly-trained diffractive decoder. The architecture of the digital encoder and the processing pipeline are shown in **Supplementary Fig. S17**. Additional comparisons in **Supplementary Fig. S18** reveal the diffractive decoder's superior performance over free-space-based image decoding, both using the same digital encoder architecture. Notably, while the free-space-based decoder completely failed in some cases, the diffractive decoder achieved stable image generation, with much better image quality at the output. As expected, we observed a numerical aperture-related minor degradation in image resolution when increasing the SLM-to-decoder distance to match our experimental conditions (see **Supplementary Fig. S18** vs. **Supplementary Fig. S19**); however, the diffractive decoder-based approach still maintains stable image generation performance compared to free-space-based decoding, which fails*

image generation in various cases despite using the same digital encoder, as shown in **Supplementary Fig. S19**.

By further increasing the number of digital encoder parameters (see **Supplementary Table S1**), we can improve the resolution and image quality of the optically generated Van Gogh-style novel artworks that are created in a snapshot; detailed comparisons are provided in **Supplementary Fig. S20** for the number of trainable parameters spanning 44M to 580M. **Figures 10 and 11** show our experimental results for higher-resolution monochrome and color (RGB) image generation using a digital encoder with 580M parameters. The monochrome images of Van Gogh-style novel artworks were generated with an illumination wavelength of 520 nm, while the color images used sequential wavelengths of {450,520,638} nm for B, G and R channels, respectively. In **Fig. 10**, the left three columns display the results where the snapshot images created by the optical generative model in a single pass closely resemble those produced by the digital diffusion model (i.e., the teacher model with 1.07 Billion trainable parameters and 1000 inference steps per image), demonstrating the consistency of our image generation process with respect to the teacher diffusion model. Conversely, the three right columns, highlighted within the orange box, showcase the optical model's ability to generate diverse images that differ from those of the teacher digital diffusion model, illustrating its creative variability at the output; also see additional experimental results for more Van Gogh-style novel artworks supporting our conclusions in **Supplementary Figs. S21-S22**.

For the multi-color Van Gogh-style novel artwork generation, phase-encoded generative seed patterns at each wavelength channel were generated and sequentially loaded onto the SLM. Under the illumination of corresponding wavelengths, multi-color images were generated through a fixed/static diffractive decoder and merged digitally; stated differently, the same decoder state was shared across all the illumination wavelengths for all the novel image generation. **Figure 11** presents the multi-color Van Gogh-style novel artwork generation results, including artistic examples that either match or differ from the outputs of the teacher digital diffusion model, which used 1.07 Billion trainable parameters and 1000 inference steps per image generation. Although slight chromatic aberrations were observed likely due to the achievable color space of the selected wavelengths and the camera's response function — the generated high-resolution color images maintained high quality. Additional experimental results of Van Gogh-style novel color artworks are provided in **Supplementary Figs. S23-S24**.

For the generation of colorful Van Gogh-style artworks, we also conducted performance comparisons for the optical generative model, reduced-size diffusion model (matching the size of our phase encoder), and the pretrained teacher diffusion model, as shown in **Supplementary Fig. S25**. Compared to our optical generative model, the reduced-size diffusion model that matches the size of our phase encoder produced inferior images with limited semantic details despite using 1000 inference steps. The optical generative model outputs closely match the teacher diffusion model (which also used 1.07B trainable parameters with 1000 inference steps).”

2, I guess this is a rush work that authors wanted to publish it ASAP. As a result, the structure of this paper is not well organized. There are many figures in this paper, but they turn to be similar that should

be re-organized. Also, the writing should be compressed, as description of the network model and the performance is repeated several times throughout the paper.

-- We thank the reviewer for this constructive suggestion. Following the referee’s suggestions, we have reorganized the figures in the manuscript to improve clarity and reduce redundancy. Specifically, **Figure 4** was moved to the Supplementary Information and expanded into **Supplementary Figs. S2–S5** for better clarity. Additionally, we refined the writing to remove repetitive descriptions of the network model and performance, making the manuscript more concise while maintaining the necessary technical details. The description of the digital encoder and the training target of DDPM can be found in **Supplementary Information Sections 1 and 2**. Transitions between sections were also improved to create a more logical flow of information. We believe these revisions enhance the overall readability and structure of the paper.

Accordingly, the following parts were reorganized to the **Supplementary Information** of the revised manuscript:

“...For lower-resolution optical image generation, we use a variant of the multi-layer perceptron as the digital encoder. For a random sampled input $\mathcal{J}(x, y) \sim \mathcal{N}(0, I)$, the digital signal processed by the l_d^{th} layer can be calculated by:

$$\mathcal{H}^{(l_d)} = \kappa(W^{(l_d)}\mathcal{H}^{(l_d-1)} + b^{(l_d)}) \quad (1),$$

where the $\mathcal{H}^{(l_d-1)}$ is the output of the $(l_d - 1)^{th}$ layer, and $\mathcal{H}^{(0)} = \text{flatten}(\mathcal{J}(x, y))$ is the input of the first layer. $W^{(l_d)} \in \mathbb{R}^{m_{l_d} \times m_{l_d-1}}$ is the weight matrix, $b^{(l_d)} \in \mathbb{R}^{m_{l_d}}$ is the bias, and m_{l_d} is the number of neurons in the l_d^{th} layer. Note that the last layer predicts a scaling factor s , so the digital encoder’s output $\mathcal{H}_{out}^{(L_d)} \in \mathbb{R}^{hw+1}$ is split into 1D output signal $\mathcal{H}^{(L_d)} \in \mathbb{R}^{hw}$ and $s \in \mathbb{R}^1$. Then, the 1D output signal $\mathcal{H}^{(L_d)} \in \mathbb{R}^{hw}$ is reshaped to the 2D dimension of $\mathbb{R}^{h \times w}$ (denoted as $\text{reshape}(\cdot)$), which can be represented by the operation that reshapes a vector $v \in \mathbb{R}^{hw}$ into a matrix $\tilde{\phi} \in \mathbb{R}^{h \times w}$:

$$\tilde{\phi}_{a,b} = v_{(a-1) \times w + b} \quad (2),$$

where $a = 1, 2, \dots, h$ is the row index, $b = 1, 2, \dots, w$ is the column index.

*For higher-resolution optical image generation (e.g., for artwork generation), we utilized a field propagation-based processing pipeline. As shown in **Supplementary Fig. S17**, the digital encoder here consists of three parts: noise feature processor, in silico field propagator, and complex field converter that is conducted by a shallow U-Net. In the noise feature processor, the latent noise and the embedded class information are concatenated and processed by FC layers. Similar to lower-resolution optical image generation, the FC layer outputs the latent image feature and the scaling factor. Then, the latent features are unsampled and processed by the convolutional layers to produce a complex field ψ_o with the desired image dimension, which can be formulated as follows:*

$$\psi_o = \text{upconv}(\text{reshape}(\text{split}(\text{FC}(\mathcal{H}^{(0)})))) \quad (3),$$

where the $\text{FC}(\cdot)$ maps the input feature $\mathcal{H}^{(0)} \in \mathbb{R}^{xyc+l}$ into $\mathcal{H}^{(1)} \in \mathbb{R}^{xyc \cdot n_{xy}^2 \cdot n_c}$, x, y are the spatial dimensions of the latent feature, c is the channel dimension of the latent feature, n_{xy} is the spatial compression factor, and n_c is the channel compression factor. The $\text{split}(\cdot)$ operation separates the latent image features and the scaling factor. The $\text{reshape}(\cdot)$ is the same operation as in Eq. S2. The $\text{upconv}(\cdot)$ is an up-convolutional architecture composed of $1/(n_{xy}^2 \cdot n_c)$

sequential up-sampling and convolutional layers (with $64k_c$ channels), interleaved with $\text{LeakyReLU}(\cdot)$ activation functions with a slope of 0.2. After the noise feature processor, an *in silico* field propagation is conducted from the image plane to the SLM plane, aiming to let the digital encoder interpret the processing conducted by the diffractive decoder. The complex field ψ_o at the image plane was numerically backward-propagated through the trainable diffractive decoder ϕ^{l_0} to the SLM plane ψ_{SLM} . The virtual complex field at the SLM plane ψ_{SLM} is calculated by:

$$\psi_{SLM}(x, y) = \mathcal{P}_f^{d_{0,1}} \mathcal{P}_m^{\phi^{l_0}} \mathcal{P}_f^{d_{1,2}}(\psi_o(x, y)) \quad (4),$$

where $d_{0,1}$ and $d_{1,2}$ are the axial distance from the SLM plane to the decoding layer and from the decoding layer to the sensor plane, respectively. $\mathcal{P}_m^{\phi^{l_0}}$ is the decoding phase modulation. Finally, the complex field is converted to the encoded optical random seeds ϕ_{SLM} on the SLM plane:

$$\phi_{SLM} = \text{Unet}(\psi_{SLM}) \quad (5).$$

The complex field converter is a U-Net model that has 3 down-sampling and 3 up-sampling stages, with channel dimensions of $32k_u$, $64k_u$, $128k_u$, respectively. Here k_u is the channel multiplying factor to control the complexity of the complex field converter. The down-sampling and up-sampling operations between each stage use convolutional and transposed convolutional layers to realize 2x down/up-sampling, respectively. The down/up-sampling stages with the same spatial dimension have skip connections. In each stage, the features are processed by 2 Res-Blocks [1]. Before/after the processing of U-Net, the input/output images are passed through a convolutional layer to get the desired channel dimensions. We used $\text{LeakyReLU}(\cdot)$ activation function after every convolutional layer, and the negative slope was set to 0.2.

As shown in **Supplementary Fig. S20**, we investigated the performance of the digital encoder under different numbers of trainable parameters. In our implementation, (x, y, c) equals $(80, 80, 4)$ for each model and the factors to control the model size are listed in **Table S1** below:”

“...Therefore, the reverse process of the Denoising Diffusion Probabilistic Model (DDPM) is to gradually reconstruct the target data distribution by iteratively removing the noise ϵ_t and adding the perturbation from timestep t to $t - 1$, which can be formulated as:

$$\mathcal{J}_{t-1} = \frac{1}{\sqrt{\bar{\alpha}_t}} \left(\mathcal{J}_t - \frac{1 - \alpha_t}{\sqrt{1 - \bar{\alpha}_t}} \epsilon_t \right) + \sigma_t z \quad (21),$$

where $\bar{\alpha}_t = \prod_{s=1}^t \alpha_s$ and $\alpha_t = 1 - \beta_t$, with $\beta_t = \beta_{start} + t/T \cdot (\beta_{end} - \beta_{start})$ being a linear function of the timestep $t \in [1, T]$. $\sigma_t^2 = (1 - \bar{\alpha}_{t-1}/1 - \bar{\alpha}_t) \cdot \beta_t$. $z \sim \mathcal{N}(0, I)$ for $t > 1$, $z = 0$ when $t = 1$. The DDPM uses a U-Net model to predict ϵ_t from the noised sample \mathcal{J}_t . Therefore, the objective of the U-Net is defined as follows:

$$\mathcal{L}(\theta_{U-Net}) = \min_{\theta_{U-Net}} E_{t \sim [1, T], \mathcal{J}_0 \sim p_{data}(\mathcal{J}), \epsilon \sim \mathcal{N}(0, I)} \left[\left\| \epsilon_t - \epsilon_{\theta_{U-Net}}(\mathcal{J}_t, t) \right\|^2 \right] \quad (22),$$

where θ_{U-Net} represents the parameters of the U-Net model, T is the total timestep in the denoising scheduler, \mathcal{J}_0 represents the original inputs sampled from the target data distribution, ϵ_t is the Gaussian noise sampled in each noising process, $\epsilon_{\theta_{U-Net}}(\mathcal{J}_t, t)$ is the operation to predict the additive noise according to the noised sample \mathcal{J}_t and timestep t . For a given timestep t , the noised sample \mathcal{J}_t is calculated by $\mathcal{J}_t = \sqrt{\bar{\alpha}_t} \mathcal{J}_0 + \sqrt{1 - \bar{\alpha}_t} \epsilon_t$.”

3, Another concern lies in that, compared to the previous implementation of optical neural networks, what is new in this work? It looks like that they are all composed of several feed-forward layers. And if it is true, the credit should be purely given to diffusion model, and its hardware implementation is straightforward as previous works.

-- We thank the referee for raising this important point, which provides an opportunity for us to clarify the innovative contributions of our work compared to previous implementations of optical neural networks. **It is crucial to highlight that this is the first demonstration of novel image generation using a diffractive optical network.** Previous optical networks were primarily applied to tasks such as imaging, classification, and deterministic operations. **In contrast, our work distinguishes itself by enabling the generation of diverse images from various noise inputs, a capability not addressed in prior studies.** Furthermore, the phase encoding model employed in our approach utilizes an effective nonlinear information encoding mechanism and significantly enhances the capabilities of the diffractive decoder.

Accordingly, we included these points in the **Introduction section and the Discussion section:**

“...The presented framework is highly flexible since different generative optical models targeting different data distributions share the same optical architecture with an optimized diffractive decoder that is fixed/static for each task, synthesizing countless novel images using optical generative seeds phase-encoded from random noise.”

“...This work presents the demonstration of snapshot optical image generation from noise patterns by leveraging a diffractive network architecture inspired by diffusion models. Earlier free-space-based optical networks primarily focused on tasks such as computational imaging, object/scene classification, or deterministic operations [68-78]. In contrast, our framework introduces the ability to optically generate diverse novel images from noise, showcasing a highly desired "creative" image generation capability that extends beyond the scope of prior studies.”

*“...Moreover, it is important to note that the phase encoding strategy employed by the optical generative model provides an effective nonlinear information encoding mechanism since linear combinations of phase patterns at the input do not create complex fields or intensity patterns at the output that can be represented as a linear superposition of the individual outputs. In fact, this phase encoding strategy significantly enhances the capabilities of the diffractive decoding layer; for comparison, we trained optical generative models using amplitude encoding or intensity encoding, as presented in **Supplementary Fig. S26**, which further highlights the advantages of phase encoding with its superior performance as quantified by the lower FID scores on generated handwritten digit images. Similarly, for the generation of Van Gogh-style novel artworks, the optical generative models using amplitude encoding or intensity encoding failed to produce consistent high-quality and high-resolution output images, as shown in **Supplementary Fig. S26**, whereas the phase encoding strategy successfully generated Van Gogh-style novel artworks. These comparisons clearly underscore the critical role of phase encoding in the optical generative model.”*

We would like also to note that, while the digital diffusion model (serving as the teacher model) is indeed important, the role of the optical diffractive decoder is also crucial. Furthermore, the teacher diffusion model typically takes 1000 steps to generate a novel image, whereas the optical generative model results are achieved in a snapshot. As shown in Supplementary Fig. S6b as well as Supplementary Figs. S18-S19, models without an optical decoder (i.e., using just free-space propagation) fail to generate novel images despite the presence of the digital diffusion model-based teacher and the same digital encoder, which demonstrates that the diffractive decoder surface plays a vital role in improving the visual quality of the snapshot-generated novel optical images.

Additionally, as presented in the newly added **Supplementary Fig. 26**, we investigated various optical generative models utilizing different encoding strategies, including phase encoding, amplitude encoding, and intensity encoding. Our findings indicate that these approaches yield suboptimal generation performance compared to the phase encoding strategy. Specifically, for the MNIST image generation task, we calculated the FID for each encoding method, demonstrating that **phase encoding consistently achieves superior results**. Furthermore, **our evaluations on generating Van Gogh-style novel artworks reinforce this observation, with phase encoding delivering the most compelling artistic outputs with valid semantic information**.

These points have also been stated in the **Results section** of the manuscript:

*"... We also compared the architecture of our snapshot optical generative models with respect to a free-space propagation-based optical decoding model, where the diffractive decoder was removed; see **Supplementary Figs. S6a-b**. The results of this comparison demonstrate that the diffractive decoder surface plays a vital role in improving the visual quality of the generated novel images."*

*"...Additional comparisons in **Supplementary Fig. S18** reveal the diffractive decoder's superior performance over free-space-based image decoding, both using the same digital encoder architecture. Notably, while the free-space-based decoder completely failed in some cases, the diffractive decoder achieved stable image generation, with much better image quality at the output. As expected, we observed a numerical aperture-related minor degradation in image resolution when increasing the SLM-to-decoder distance to match our experimental conditions (see **Supplementary Fig. S18** vs. **Supplementary Fig. S19**); however, the diffractive decoder-based approach still maintains stable image generation performance compared to free-space-based decoding, which fails image generation in various cases despite using the same digital encoder, as shown in **Supplementary Fig. S19**."*

We have also added the related points to the **Discussion section** of the revised manuscript:

*"...Moreover, it is important to note that the phase encoding strategy employed by the optical generative model provides an effective nonlinear information encoding mechanism since linear combinations of phase patterns at the input do not create complex fields or intensity patterns at the output that can be represented as a linear superposition of the individual outputs. In fact, this phase encoding strategy significantly enhances the capabilities of the diffractive decoding layer; for comparison, we trained optical generative models using amplitude encoding or intensity encoding, as presented in **Supplementary Fig. S26**, which further highlights the advantages of*

phase encoding with its superior performance as quantified by the lower FID scores on generated handwritten digit images. Similarly, for the generation of Van Gogh-style novel artworks, the optical generative models using amplitude encoding or intensity encoding failed to produce consistent high-quality and high-resolution output images, as shown in Supplementary Fig. S26, whereas the phase encoding strategy successfully generated Van Gogh-style novel artworks. These comparisons clearly underscore the critical role of phase encoding in the optical generative model.”

4, Regarding the model, I believe some concise equations explaining the details should be helpful to readers.

-- We thank the referee for the valuable suggestions. We consolidated some of the detailed information into the supplementary information file to ensure clarity and readability of the main text. In response, we expanded **Supplementary Information Section 1** to include additional key equations of the digital encoder that more clearly illustrate the model's formulation and functionality. Besides, to enhance readability, we relocated some equations about the diffusion model training equations to **Supplementary Information Section 2**. These structural adjustments allow the revised **Methods** section to concisely present the technical details without compromising methodological rigor. We believe these modifications improved both the logical flow and technical completeness of our revised manuscript.

5, The diffractive decoder is the key in optical generative model. How did you prepare this decoder, and how much effort and cost are needed?

-- In the improved image generation experiments reported in our manuscript, we employed a spatial light modulator (SLM) as the reconfigurable decoding layer. The use of an SLM allows for convenient modification of phase patterns for different tasks and styles of image generation without requiring any changes to the optical architecture or realignment of components. Besides, the device can provide an 8 bit-depth (256 levels) within $0 - 2\pi$ phase modulation range, which is sufficient for our design. This capability is demonstrated in the **revised Fig. 7, the newly added Figs. 9-11, the first row of Supplementary Fig. S9, as well as Supplementary Figs. S21-24**, where the optical generation model produced high-quality images.

As we emphasized in our main text, the presented framework is highly flexible since different generative optical models targeting different data distributions share the same optical architecture with **an optimized diffractive decoder that is fixed/static for each task, synthesizing countless novel images using optical generative seeds phase-encoded from random noise**. Beyond reconfigurable decoders that can cover multiple image generation tasks, advanced fabrication techniques such as two-photon polymerization (TPP) 3D printing or optical lithography can also be used to fabricate the passive/static decoding layer for a given image generation task, e.g., artwork generation.

To assess the potential impact of phase bit-depth limitations of an optical generative model set-up, we conducted additional numerical analyses by imposing constraints on the phase patterns and the structure of the diffractive decoding layer. To be specific, we set the bit depth of the SLM and/or the

diffractive decoder to 4 bits (i.e., 16 discrete levels) during the testing phase. As shown in **Supplementary Fig. S9**, although the models were trained without any bit-depth constraints, they demonstrated acceptable image generation performance. Additionally, we explored a fabrication scenario involving a static decoding layer with only **3 available phase levels (e.g., 0 , $\pi/3$, $2\pi/3$)**. In this case, our model still exhibited robust image generation performance during blind testing, indicating the system's resilience to reduced phase levels. Finally, to mitigate the impact of this physical constraint in the set-up of an optical generative model (for example, in a resource-limited setting), we trained an optical generative model using an 8-bit SLM and a **static decoding layer with 3 discrete phase levels (0 , $\pi/3$, $2\pi/3$)**, which showed notable performance improvements (as shown in **Supplementary Fig. S9**) compared to models trained **without** considering bit-depth limitations of the set-up.

The above points were merged into the **Results section** and **Discussion section** of the revised manuscript:

*“...In addition to these results, we also experimentally evaluated the snapshot optical image generation under a limited phase encoding space (e.g., 0 - $\pi/2$ vs. 0 - 2π) and a limited decoder bit depth (e.g., 4 bit-depth vs. 8 bit-depth) using a restricted optical setup; see **Supplementary Figs. S14a, S14b and S15**. The performance comparisons between the experimental results of **Fig. 7** and **Supplementary Fig. S14** show the significance of employing a large phase bit depth for the diffractive decoder as well as increasing the encoding phase range of the input SLM.”*

*“...Furthermore, in **Supplementary Fig. S9**, we explored the impact of limited phase modulation levels (i.e., a limited phase bit-depth per feature) at the optical generative seed plane and the diffractive decoder. These comparisons revealed that novel image generation results could be improved by including the modulation bit depth limitation (due to, e.g., inexpensive SLM hardware or surface fabrication limitations) in the forward model of the training process. Such a training strategy using a limited phase bit-depth revealed that the fixed/static decoder surface could work with 4 phase bit-depth and even 3 discrete levels of phase (e.g., 0 , $\pi/3$, $2\pi/3$) per feature to successfully generate novel images through its decoder phase function; see **Supplementary Fig. S9**. This is important and highly desired since most two-photon polymerization or optical lithography-based fabrication methods [52-54] can routinely fabricate surfaces with 2-16 discrete phase levels per feature, which could help replace the decoder SLM with a passive fabricated surface structure.”*

*“...Despite some of these appealing features discussed above, there are also some challenges associated with optical generative models in general. Potential misalignments and physical imperfections within the optical hardware/setup present challenges, which might degrade the novel image generation performance. Another challenge is the limited phase bit depth of the optical modulator devices or surfaces that physically represent the generative optical seeds and the decoder layer(s), which might constrain the upper-performance limit of novel image representation and generation. To investigate this, we numerically analyzed three scenarios with varying phase bit-depth levels and evaluated their impact by imposing these constraints during the testing phase, as shown in **Supplementary Fig. S9**. Remarkably, models trained without such constraints or limitations were still able to generate novel handwritten digits despite the imposed phase bit-depth limitations during testing. To better mitigate these physical challenges*

*in the optical setup, one can integrate these limitations directly into the training process of the optical generative model, which will accordingly shape both the digital encoder and the diffractive decoder, making the in silico-optimized system align better with the physical limits and the capabilities of the local hardware of the optical generative model. As demonstrated in **Supplementary Fig. S9**, this strategy led to notable performance improvements compared to models that did not account for such bit-depth limitations during training. A key insight of this analysis is that a relatively simpler decoder surface with just 3 discrete phase levels (covering only 0, $\pi/3$ or $2\pi/3$ per feature) would be sufficient in our optical generative models, opening the door to replacing the decoder architecture with a passive, thin surface fabricated by e.g., two-photon polymerization or optical lithography based nano-fabrication techniques [52-54]. This would further simplify the physical setup of the local optical generative model, also making its hardware more compact, lightweight and cost-effective.”*

*“...We also investigated the influence of the number of diffractive layers and the performance limitations arising from potential misalignments in the fabrication or assembly of a multi-layer diffractive decoder trained for optical image generation. **Supplementary Fig. S8d** reveals that the quality of iterative optical image generation without a digital encoder exhibits a degradation with a reduced number of layers. Furthermore, as shown in **Supplementary Fig. S11**,”*

6, In all optical models, the optical-digital conversion should be considered when benchmarking the performance. Is it considered in this work? For instance, for the results in Fig. 4. BTW, the top-panel plot in Fig. 4 is really confusing, as all bars are the same for the two indexes.

-- We thank the referee for raising these important points, which helped us enhance the clarity and precision of our revised manuscript.

First, regarding the top-panel plot in **Fig. 4**: to address this issue, we expanded the figure to **Supplementary Fig.S2-S5** to better enhance data representation and ensure clear communication of the intended information. Specifically, for the blue bar representing inference FLOPs, we now exclusively display the digital inference FLOPs of our digital encoder, as optical computing inherently does not involve floating-point operations. Regarding the orange bar that represents the number of trainable parameters, we specified that it included the optimizable parameters from both the digital encoder and optical decoder side. These modifications effectively resolved the issues raised by the referee, and these points were added to the **Results section** of the revised manuscript, quoted below:

*“...We also conducted performance comparisons between snapshot optical generative models and all-digital deep-learning-based models formed by stacked FC layers, trained on the same image generation task. **Supplementary Figs. S2-S5** present different configurations of these optical and all-digital generative models. In these figures, we report their computing operations (i.e., the floating-point operations, FLOPs), training parameters, average IS values, and examples of the generated novel images, providing a comprehensive comparison of these approaches. The digital generative models in **Supplementary Fig. S2** were trained in an adversarial manner, which revealed that when the depth of the all-digital deep-learning-based generative model is shallow, the output image quality cannot capture the whole distribution of the target dataset, resulting in failures or repetitive generations. On the other hand, the snapshot*

optical generative model with a shallow digital encoder is able to realize a statistically comparable novel image generation performance, matching the performance of a deeper digital generative model stacked with nine FC layers (see **Supplementary Fig. S2**). To provide additional comparisons, the digital models in **Supplementary Figs. S3-S4** were trained using the same teacher DDPM as used in the training of the optical generative models, and the results showed similar conclusions. In **Supplementary Fig. S5**, we also show comparisons using the digital DDPM, where the number of parameters of the U-Net in DDPM was reduced to match that of our shallow digital encoder, which resulted in some image generation failures in the outputs of the digital DDPM (despite using 1000 denoising steps), which are exemplified with the red squares in **Supplementary Fig. S5c**. Overall, our findings reported in **Supplementary Figs. S2-S5** suggest that using a large DDPM as a teacher for the optical generative models can realize stable synthesis of novel images in a single snapshot through a shallow digital phase encoder followed by the optical diffractive decoder.”

Regarding digital-to-optical and optical-to-digital conversion and related interfaces, we have taken into account various factors with quantitative reports of energy consumption (including e.g., the SLM, illumination source and camera etc.) and added a comprehensive discussion of these factors in our revised manuscript, Discussion section:

“...The presented optical generative models comprise four primary components: the electronic encoder network, the input SLM, the illumination light, and the diffractive decoder, which are collectively optimized for novel image display. The electronic encoder used for MNIST and Fashion MNIST datasets consists of three fully connected layers, and it requires 6.29 MFLOPs per image, with an energy cost of $\sim 0.5\text{-}5.5$ pJ/FLOP [80-81], resulting in an energy consumption of 0.003-0.033 mJ per image. This energy consumption increases to $\sim 0.28\text{-}3.08$ J and $\sim 1.13\text{-}12.44$ J per image for the Van Gogh-style artworks reported in **Fig. 9** and **Figs. 10-11**, respectively. The input SLM, with a power range of 1.9–3.5 W, consumes $\sim 30\text{-}58$ mJ per image at a 60 Hz refresh rate. This SLM-related energy consumption can be reduced to <2.5 mJ per image using a state-of-the-art SLM [56-58]. The diffractive decoder has a similar energy consumption if a second SLM is used; however, its contribution would become negligible if a static decoder (e.g., a passive fabricated surface/layer) is employed. As for the illumination light, the energy consumption per wavelength channel can be estimated to be less than 0.8 mJ per image [82], which is negligible compared to other factors. If the generated images were to be digitized by an image sensor chip (e.g., a 5-10 mega-pixel **CMOS imager**), this would also add an extra energy consumption of $\sim 2\text{-}4$ mJ per image. Consequently, the overall energy consumption for generated images intended for human perception — excluding the need for a digital camera — is dominated by the SLM-based power in lower resolution image generation, whereas the digital encoder power consumption becomes the dominant factor for higher resolution image generation tasks such as the Van Gogh-style artworks. In contrast, GPU-based generative systems using a DDPM model have different energy characteristics that are dominated by the diffusion and successive denoising processes (involving e.g., 1000 steps). For example, the computational requirements for generating MNIST/Fashion MNIST and Van Gogh-style artwork images using a digital DDPM model amount to approximately 287.68 GFLOPs and 530.26 TFLOPs, respectively, corresponding to about 0.14–1.58 J per image for MNIST/Fashion-MNIST and 265–2916 J per image for Van Gogh-style artworks. Furthermore,

if the generated images must be displayed on a monitor for human perception, additional energy consumption is incurred—typically between ~13–500 mJ per image at a 60 Hz refresh rate.

Overall, these comparisons reveal that if the novel image information to be generated will be stored and processed/harnessed in the digital domain, optical generative models would face additional power and speed penalties due to the digital-to-analog and analog-to-digital conversion steps that would be involved in the optical setup. However, if the image information to be generated will remain in the analog domain for direct visualization by human observers (e.g., in a near-eye or head-mounted display), the optical generative seeds can be pre-calculated with a modest energy consumption per seed, as detailed above. Furthermore, the static diffractive decoder surface can be fabricated using optical lithography or two-photon polymerization-based nano-fabrication methods, which would optically generate snapshot novel images within the display setup. This could enable compact and cost-effective image generators—such as "optical artists"—by replacing the back-end diffractive decoder with a fabricated passive surface. This setup would allow for the snapshot generation of countless novel images, including various forms of artwork, using simpler local optical hardware. From the perspective of a digital generative model, for comparison, one could also use a standard image display along with pre-computed and stored novel images created through, e.g., a digital DDPM model; this, however, requires significantly more energy consumption per image generation through the diffusion and successive denoising processes, as discussed earlier. Exploration of optical generative architectures using nano-fabricated surfaces would enable various new applications, especially for image and near-eye display systems, including head-mounted and wearable setups.”

Additionally, we have revised the manuscript to include various performance comparisons between the snapshot optical generative models and all-digital deep-learning-based models, with several new figures added; quoted from our revised manuscript, Results section:

“...We also conducted performance comparisons between snapshot optical generative models and all-digital deep-learning-based models formed by stacked FC layers, trained on the same image generation task. **Supplementary Figs. S2-S5** present different configurations of these optical and all-digital generative models. In these figures, we report their computing operations (i.e., the floating-point operations, FLOPs), training parameters, average IS values, and examples of the generated novel images, providing a comprehensive comparison of these approaches. The digital generative models in **Supplementary Fig. S2** were trained in an adversarial manner, which revealed that when the depth of the all-digital deep-learning-based generative model is shallow, the output image quality cannot capture the whole distribution of the target dataset, resulting in failures or repetitive generations. On the other hand, the snapshot optical generative model with a shallow digital encoder is able to realize a statistically comparable novel image generation performance, matching the performance of a deeper digital generative model stacked with nine FC layers (see **Supplementary Fig. S2**). To provide additional comparisons, the digital models in **Supplementary Figs. S3-S4** were trained using the same teacher DDPM as used in the training of the optical generative models, and the results showed similar conclusions. In **Supplementary Fig. S5**, we also show comparisons using the digital DDPM, where the number of parameters of the U-Net in DDPM was reduced to match that of our shallow digital encoder, which resulted in some image generation failures in the

*outputs of the digital DDPM (despite using 1000 denoising steps), which are exemplified with the red squares in **Supplementary Fig. S5c**. Overall, our findings reported in **Supplementary Figs. S2-S5** suggest that using a large DDPM as a teacher for the optical generative models can realize stable synthesis of novel images in a single snapshot through a shallow digital phase encoder followed by the optical diffractive decoder.”*

*“...For the generation of **colorful Van Gogh-style artworks**, we also conducted performance comparisons for the optical generative model, reduced-size diffusion model (matching the size of our phase encoder), and the pretrained teacher diffusion model, as shown in **Supplementary Fig. S25**. Compared to our optical generative model, the reduced-size diffusion model that matches the size of our phase encoder produced inferior images with limited semantic details despite using 1000 inference steps. The optical generative model outputs closely match the teacher diffusion model (which also used 1.07B trainable parameters with 1000 inference steps).”*

7, When compared to electronic systems, the storage of parameters in memory is usually overlooked in optical computing systems. How should you consider this aspect, especially in the case of performance benchmarking?

-- Please see our former response to Question #6 above. In our snapshot image generation set-up, the generative phase seed can be pre-calculated and stored on a PC or in the cloud, which then transmits the corresponding phase pattern to the SLM for snapshot novel image generation and diffractive decoding. For a human observer, image display or projection is a necessary task and would also be needed for all-digital generative models, such as the teacher diffusion model. Therefore, this is not a relative disadvantage for an optical generative model that communicates with a human observer as part of, e.g., a head-mounted and wearable display.

As we emphasized in our main text, the presented framework is highly flexible since different generative optical models targeting different data distributions share the same optical architecture with **an optimized diffractive decoder that is fixed/static for each task**, synthesizing countless novel images using optical generative seeds phase-encoded from random noise. This means that the decoder surface is fixed for a given task and does not change from one novel image to another novel image generation unless the image generation task itself is changed to a new distribution. Beyond reconfigurable decoders that can cover multiple image generation tasks, advanced fabrication techniques such as two-photon polymerization (TPP) 3D printing or optical lithography can also be used to fabricate the passive/static decoding layer for a given image generation task, e.g., artwork generation. This fabrication approach requires a one-time effort, and in that case, the parameters would be inherently stored within the fabricated layer.

We have added a comprehensive discussion of these factors in our revised manuscript, Discussion section:

“...The presented optical generative models comprise four primary components: the electronic encoder network, the input SLM, the illumination light, and the diffractive decoder, which are collectively optimized for novel image display. The electronic encoder used for MNIST and Fashion MNIST datasets consists of three fully connected layers, and it requires 6.29 MFLOPs per image, with an energy cost of ~0.5-5.5 pJ/FLOP [80-81], resulting in an energy consumption

of 0.003-0.033 mJ per image. This energy consumption increases to ~0.28-3.08 J and ~1.13-12.44 J per image for the Van Gogh-style artworks reported in **Fig. 9** and **Figs. 10-11**, respectively. The input SLM, with a power range of 1.9–3.5 W, consumes ~30–58 mJ per image at a 60 Hz refresh rate. This SLM-related energy consumption can be reduced to <2.5 mJ per image using a state-of-the-art SLM [56-58]. The diffractive decoder has a similar energy consumption if a second SLM is used; however, its contribution would become negligible if a static decoder (e.g., a passive fabricated surface/layer) is employed. As for the illumination light, the energy consumption per wavelength channel can be estimated to be less than 0.8 mJ per image [82], which is negligible compared to other factors. If the generated images were to be digitized by an image sensor chip (e.g., a 5-10 mega-pixel **CMOS imager**), this would also add an extra energy consumption of ~2–4 mJ per image. Consequently, the overall energy consumption for generated images intended for human perception — excluding the need for a digital camera — is dominated by the SLM-based power in lower resolution image generation, whereas the digital encoder power consumption becomes the dominant factor for higher resolution image generation tasks such as the Van Gogh-style artworks. In contrast, GPU-based generative systems using a DDPM model have different energy characteristics that are dominated by the diffusion and successive denoising processes (involving e.g., 1000 steps). For example, the computational requirements for generating MNIST/Fashion MNIST and Van Gogh-style artwork images using a digital DDPM model amount to approximately 287.68 GFLOPs and 530.26 TFLOPs, respectively, corresponding to about 0.14–1.58 J per image for MNIST/Fashion-MNIST and 265–2916 J per image for Van Gogh-style artworks. Furthermore, if the generated images must be displayed on a monitor for human perception, additional energy consumption is incurred—typically between ~13–500 mJ per image at a 60 Hz refresh rate.

Overall, these comparisons reveal that if the novel image information to be generated will be stored and processed/harnessed in the digital domain, optical generative models would face additional power and speed penalties due to the digital-to-analog and analog-to-digital conversion steps that would be involved in the optical setup. However, if the image information to be generated will remain in the analog domain for direct visualization by human observers (e.g., in a near-eye or head-mounted display), the optical generative seeds can be pre-calculated with a modest energy consumption per seed, as detailed above. Furthermore, the static diffractive decoder surface can be fabricated using optical lithography or two-photon polymerization-based nano-fabrication methods, which would optically generate snapshot novel images within the display setup. This could enable compact and cost-effective image generators—such as "optical artists"—by replacing the back-end diffractive decoder with a fabricated passive surface. This setup would allow for the snapshot generation of countless novel images, including various forms of artwork, using simpler local optical hardware. From the perspective of a digital generative model, for comparison, one could also use a standard image display along with pre-computed and stored novel images created through, e.g., a digital DDPM model; this, however, requires significantly more energy consumption per image generation through the diffusion and successive denoising processes, as discussed earlier. Exploration of optical generative architectures using nano-fabricated surfaces would enable various new applications, especially for image and near-eye display systems, including head-mounted and wearable setups.”

8, Some references are not appropriate, e.g., those about emerging approaches for large generative AI models. Please check.

-- Following the referee's suggestions, we have carefully reviewed and revised the references throughout the manuscript. Additionally, we have included new relevant references to ensure coverage of related literature and to provide a more proper introduction to our work.

Referee #2:

In their manuscript Optical Generative Models, Chen et al report on the implementation of a digital-electronic / analog-optical generative network using off the shelf optical components. The results the authors report are indeed very surprising and fascinating in terms of generality. Using their concept, the authors demonstrate the generation of hand written digits, fashion items, butterflies and faces with high quality and robustness. The experimental results reported are sound, I do have some objections regarding claims made by the authors. Regarding a recommendation I am undecided at the moment. The quality and novelty to the work contrasts with some of the claims and a lack of important analysis.

-- We sincerely thank the referee for the constructive and comprehensive feedback, which helped us further improve the quality and clarify of our revised manuscript.

My major points:

1. Ultra fast inference time

The authors claim that inference time is extremely short – 1 ns. I strongly object to this claim. Firstly, this is only true for the optical propagation time and does not include (i) the implementation of the shallow encoder neither (ii) the response time of the SLM used to encode the injection phases and neither (iii) the camera frame rate. This would be like comparing the modulation speed of a transistor and ignoring the impact of the clock-network in an electronic chip. The contribution of (iii) one might ignore since the image is created without the camera. However, (i) and (ii) are essential ingredients to image creation and hence have to be included. Hence, we are rather talking about 100ms.

-- We thank the referee for raising this important concern regarding the inference time of our optical generative models. We acknowledge that “<1 ns” indeed refers specifically to the optical propagation time from the optical seed plane to the image plane, **indicating a thin optical volume for the diffractive decoder**, which excludes other factors such as the digital encoder's computation time and the response time of the SLM. **To avoid confusing our readers, we have removed such statements in our revised manuscript.** We have also revised the **Introduction section** and **Results section** of our manuscript accordingly to better emphasize these points, quoted below:

“...; the refresh rate is limited by the frame-rate of the SLM that displays the pre-calculated optical generative seeds. This image synthesis process from the optical seed plane to the output image requires power for the illumination light and the SLM operation, without additional computing power needed. The optical part of the computation required for snapshot image generation is

carried out entirely through free-space propagation of light via an optimized and fixed (i.e., static) diffractive decoder.”

“Since the encoded phase patterns can be pre-calculated by feeding random Gaussian noise to a shallow encoder network, they can be randomly accessed on demand, where novel images can be locally synthesized by the optical generative model. The overall inference time is constrained by the SLM loading time, which can be minimized by employing faster phase light modulators (PLMs) or SLMs with frame rates exceeding 1 kHz [56-58]. ...”

2. Focus on FLOPs and Parameters for comparison

As another argument the authors claim that their system needs 3 times less FLOPs. However, FLOPs are only of indirect importance as they link to energy consumption. Considering a FLOP uses <1 pJ I think it is fair to assume that the SLM will use more power than one is able to safe.

-- We thank the referee for raising this important discussion regarding energy consumption. Indeed, while FLOPs provide an indirect measure of computational energy efficiency and remain a standard metric for evaluating the complexity of generative models, they do not fully capture the energy requirements of the system—particularly those related to, for example, the SLM and illumination light.

To address the referee’s points, we have taken into account various factors with quantitative reports of energy consumption (including e.g., the SLM, illumination source and camera etc.) and added a comprehensive and balanced discussion of these factors in our revised manuscript, Discussion section:

*“...The presented optical generative models comprise four primary components: the electronic encoder network, the input SLM, the illumination light, and the diffractive decoder, which are collectively optimized for novel image display. The electronic encoder used for MNIST and Fashion MNIST datasets consists of three fully connected layers, and it requires 6.29 MFLOPs per image, with an energy cost of ~0.5-5.5 pJ/FLOP [80-81], resulting in an energy consumption of 0.003-0.033 mJ per image. This energy consumption increases to ~0.28-3.08 J and ~1.13-12.44 J per image for the Van Gogh-style artworks reported in **Fig. 9** and **Figs. 10-11**, respectively. The input SLM, with a power range of 1.9–3.5 W, consumes ~30–58 mJ per image at a 60 Hz refresh rate. This SLM-related energy consumption can be reduced to <2.5 mJ per image using a state-of-the-art SLM [56-58]. The diffractive decoder has a similar energy consumption if a second SLM is used; however, its contribution would become negligible if a static decoder (e.g., a passive fabricated surface/layer) is employed. As for the illumination light, the energy consumption per wavelength channel can be estimated to be less than 0.8 mJ per image [82], which is negligible compared to other factors. If the generated images were to be digitized by an image sensor chip (e.g., a 5-10 mega-pixel CMOS imager), this would also add an extra energy consumption of ~2–4 mJ per image. Consequently, the overall energy consumption for generated images intended for human perception — excluding the need for a digital camera — is dominated by the SLM-based power in lower resolution image generation, whereas the digital encoder power consumption becomes the dominant factor for higher resolution image generation tasks such as the Van Gogh-style artworks. In contrast, GPU-based generative systems using a DDPM model have different energy characteristics that are dominated by the diffusion and successive denoising processes (involving e.g., 1000 steps). For example, the computational requirements for generating MNIST/Fashion MNIST and Van Gogh-style artwork images using a **digital DDPM model** amount to approximately*

287.68 GFLOPs and 530.26 TFLOPs, respectively, corresponding to about **0.14–1.58 J** per image for MNIST/Fashion-MNIST and **265–2916 J** per image for Van Gogh-style artworks. Furthermore, if the generated images must be displayed on a monitor for human perception, additional energy consumption is incurred—typically between ~13–500 mJ per image at a 60 Hz refresh rate.

Overall, these comparisons reveal that if the novel image information to be generated will be stored and processed/harnessed in the digital domain, optical generative models would face additional power and speed penalties due to the digital-to-analog and analog-to-digital conversion steps that would be involved in the optical setup. However, if the image information to be generated will remain in the analog domain for direct visualization by human observers (e.g., in a near-eye or head-mounted display), the optical generative seeds can be pre-calculated with a modest energy consumption per seed, as detailed above. Furthermore, the static diffractive decoder surface can be fabricated using optical lithography or two-photon polymerization-based nano-fabrication methods, which would optically generate snapshot novel images within the display setup. This could enable compact and cost-effective image generators—such as "optical artists"—by replacing the back-end diffractive decoder with a fabricated passive surface. This setup would allow for the snapshot generation of countless novel images, including various forms of artwork, using simpler local optical hardware. From the perspective of a digital generative model, for comparison, one could also use a standard image display along with pre-computed and stored novel images created through, e.g., a digital DDPM model; this, however, requires significantly more energy consumption per image generation through the diffusion and successive denoising processes, as discussed earlier. Exploration of optical generative architectures using nano-fabricated surfaces would enable various new applications, especially for image and near-eye display systems, including head-mounted and wearable setups.”

3. Reason for the better performance in comparison to the digital decoder

The optical diffractive decoder implements a single layer with a linear matrix and an absolute square nonlinearity due to the photo-electric effect at the camera. This is a very simple operation – which is not necessarily a downside. However, I find it very confusing that using the electronic decoder the authors require more nonlinear layers to achieve the same results.

I would like to ask the authors to compare to a single layer with the same amount of parameters as implemented by the phase-diffraction SLM and to compare these results with the optical decoder.

-- We thank the referee for raising these important points.

To address the referee’s comments, we have revised the manuscript to include various performance comparisons between the snapshot optical generative models and all-digital deep-learning-based models, with several new figures added; quoted from our revised manuscript, Results section:

*“...We also conducted performance comparisons between snapshot optical generative models and all-digital deep-learning-based models formed by stacked FC layers, trained on the same image generation task. **Supplementary Figs. S2-S5** present different configurations of these optical and all-digital generative models. In these figures, we report their computing operations (i.e., the floating-point operations, FLOPs), training parameters, average IS values, and examples of the generated novel images, providing a comprehensive comparison of these*

approaches. The digital generative models in **Supplementary Fig. S2** were trained in an adversarial manner, which revealed that when the depth of the all-digital deep-learning-based generative model is shallow, the output image quality cannot capture the whole distribution of the target dataset, resulting in failures or repetitive generations. On the other hand, the snapshot optical generative model with a shallow digital encoder is able to realize a statistically comparable novel image generation performance, matching the performance of a deeper digital generative model stacked with nine FC layers (see **Supplementary Fig. S2**). To provide additional comparisons, the digital models in **Supplementary Figs. S3-S4** were trained using the same teacher DDPM as used in the training of the optical generative models, and the results showed similar conclusions. In **Supplementary Fig. S5**, we also show comparisons using the digital DDPM, where the number of parameters of the U-Net in DDPM was reduced to match that of our shallow digital encoder, which resulted in some image generation failures in the outputs of the digital DDPM (despite using 1000 denoising steps), which are exemplified with the red squares in **Supplementary Fig. S5c**. Overall, our findings reported in **Supplementary Figs. S2-S5** suggest that using a large DDPM as a teacher for the optical generative models can realize stable synthesis of novel images in a single snapshot through a shallow digital phase encoder followed by the optical diffractive decoder.”

“...For the generation of **colorful Van Gogh-style artworks**, we also conducted performance comparisons for the optical generative model, reduced-size diffusion model (matching the size of our phase encoder), and the pretrained teacher diffusion model, as shown in **Supplementary Fig. S25**. Compared to our optical generative model, the reduced-size diffusion model that matches the size of our phase encoder produced inferior images with limited semantic details despite using 1000 inference steps. The optical generative model outputs closely match the teacher diffusion model (which also used 1.07B trainable parameters with 1000 inference steps).”

We should also emphasize that the optical diffractive decoder is not merely performing a linear matrix operation with square nonlinearity. Instead, our optical generative model employs phase encoding, which provides an effective nonlinear information encoding mechanism since linear combinations of phase patterns at the input do not create complex fields or intensity patterns at the output that can be represented as a linear superposition of the individual outputs. The effectiveness of and performance improvements due to this phase encoding strategy was demonstrated in **Supplementary Fig. S6b and S26** where we compared it with the amplitude and intensity encoding of input information. The visual quality and the lower FID metrics indicate the superior performance of phase encoding on MNIST generation. For higher resolution Van Gogh-style artwork generation, the amplitude encoding or intensity encoding failed to produce consistent high-quality results. To reflect these, we have revised the **Results section** to include these points and provide a more detailed comparison:

“...We also compared the architecture of our snapshot optical generative models with respect to a free-space propagation-based optical decoding model, where the diffractive decoder was removed; see **Supplementary Figs. S6a-b**. The results of this comparison demonstrate that the diffractive decoder surface plays a vital role in improving the visual quality of the generated novel images. We also analyzed the class embedding feature in the digital encoder, as illustrated in **Supplementary Fig. S6c**; this additional analysis reveals that the snapshot image generation quality of an optical model without class embedding is lower, indicating that this additional

information conditions the optical generative model to better capture the overall structure of the underlying target data distribution”

*“...Moreover, it is important to note that the phase encoding strategy employed by the optical generative model provides an effective nonlinear information encoding mechanism since linear combinations of phase patterns at the input do not create complex fields or intensity patterns at the output that can be represented as a linear superposition of the individual outputs. In fact, this phase encoding strategy significantly enhances the capabilities of the diffractive decoding layer; for comparison, we trained optical generative models using amplitude encoding or intensity encoding, as presented in **Supplementary Fig. S26**, which further highlights the advantages of phase encoding with its superior performance as quantified by the lower FID scores on generated handwritten digit images. Similarly, for the generation of Van Gogh-style novel artworks, the optical generative models using amplitude encoding or intensity encoding failed to produce consistent high-quality and high-resolution output images, as shown in **Supplementary Fig. S26**, whereas the phase encoding strategy successfully generated Van Gogh-style novel artworks. These comparisons clearly underscore the critical role of phase encoding in the optical generative model.”*

With these points I find that the technological relevance of the system is hard to justify for the highest level of publication targeted by the authors. However, I could be convinced if the authors make a clear case for their technology. Fundamentally speaking I find the work fascinating.

-- We thank the referee for their thoughtful evaluation and for acknowledging the fascinating aspects of our work. We have carefully revised the manuscript to address all the questions and concerns raised. We hope these revisions can further strengthen the clarity and the significance of our work.

Minor comments and suggestions:

1. “This image generation process from the optical generative seed plane to the output image plane does not consume computing power except for the illumination wave.

I disagree, this needs the SLM too.

-- Please see our earlier responses on the same topic. To address this, we have revised the relevant sections of the manuscript:

“...the refresh rate is limited by the frame-rate of the SLM that displays the pre-calculated optical generative seeds. This image synthesis process from the optical seed plane to the output image requires power for the illumination light and the SLM operation, without additional computing power needed. The optical part of the computation required for snapshot image generation is carried out entirely through free-space propagation of light via an optimized and fixed (i.e., static) diffractive decoder.”

Furthermore, we have added a comprehensive energy analysis in the **Discussion** section of our revised manuscript, taking into account various factors with quantitative reports of energy consumption (including, e.g., the SLM, illumination source and camera etc.), quoted below:

“...The presented optical generative models comprise four primary components: the electronic encoder network, the input SLM, the illumination light, and the diffractive decoder, which are collectively optimized for novel image display. The electronic encoder used for MNIST and Fashion MNIST datasets consists of three fully connected layers, and it requires 6.29 MFLOPs per image, with an energy cost of $\sim 0.5\text{--}5.5$ pJ/FLOP [80-81], resulting in an energy consumption of 0.003-0.033 mJ per image. This energy consumption increases to $\sim 0.28\text{--}3.08$ J and $\sim 1.13\text{--}12.44$ J per image for the Van Gogh-style artworks reported in **Fig. 9** and **Figs. 10-11**, respectively. The input SLM, with a power range of 1.9–3.5 W, consumes $\sim 30\text{--}58$ mJ per image at a 60 Hz refresh rate. This SLM-related energy consumption can be reduced to <2.5 mJ per image using a state-of-the-art SLM [56-58]. The diffractive decoder has a similar energy consumption if a second SLM is used; however, its contribution would become negligible if a static decoder (e.g., a passive fabricated surface/layer) is employed. As for the illumination light, the energy consumption per wavelength channel can be estimated to be less than 0.8 mJ per image [82], which is negligible compared to other factors. If the generated images were to be digitized by an image sensor chip (e.g., a 5-10 mega-pixel CMOS imager), this would also add an extra energy consumption of $\sim 2\text{--}4$ mJ per image. Consequently, the overall energy consumption for generated images intended for human perception — excluding the need for a digital camera — is dominated by the SLM-based power in lower resolution image generation, whereas the digital encoder power consumption becomes the dominant factor for higher resolution image generation tasks such as the Van Gogh-style artworks. In contrast, GPU-based generative systems using a DDPM model have different energy characteristics that are dominated by the diffusion and successive denoising processes (involving e.g., 1000 steps). For example, the computational requirements for generating MNIST/Fashion MNIST and Van Gogh-style artwork images using a **digital DDPM model** amount to approximately 287.68 GFLOPs and 530.26 TFLOPs, respectively, corresponding to about **0.14–1.58 J** per image for MNIST/Fashion-MNIST and **265–2916 J** per image for Van Gogh-style artworks. Furthermore, if the generated images must be displayed on a monitor for human perception, additional energy consumption is incurred—typically between $\sim 13\text{--}500$ mJ per image at a 60 Hz refresh rate.

Overall, these comparisons reveal that if the novel image information to be generated will be stored and processed/harnessed in the digital domain, optical generative models would face additional power and speed penalties due to the digital-to-analog and analog-to-digital conversion steps that would be involved in the optical setup. However, if the image information to be generated will remain in the analog domain for direct visualization by human observers (e.g., in a near-eye or head-mounted display), the optical generative seeds can be pre-calculated with a modest energy consumption per seed, as detailed above. Furthermore, the static diffractive decoder surface can be fabricated using optical lithography or two-photon polymerization-based nano-fabrication methods, which would optically generate snapshot novel images within the display setup. This could enable compact and cost-effective image generators—such as “optical artists”—by replacing the back-end diffractive decoder with a fabricated passive surface. This setup would allow for the snapshot generation of countless novel images, including various forms of artwork, using simpler local optical hardware. From the perspective of a digital generative model, for comparison, one could also use a standard image display along with pre-computed and stored novel images created through, e.g., a digital DDPM model; this, however, requires significantly more energy consumption per image generation through the diffusion and successive denoising processes, as discussed earlier. Exploration of optical generative architectures using nano-fabricated surfaces would enable

various new applications, especially for image and near-eye display systems, including head-mounted and wearable setups.”

2. “To experimentally demonstrate the proof-of-concept[...]”

Starting from there till the end of the section I find the text quite repetitive from what was said before.

-- We carefully reviewed the related section and revised it accordingly to avoid repetitions.

3. Figure. -> Figure? Typo

-- We have corrected this typo throughout the manuscript to ensure consistency and accuracy.

4. and are projected onto a spatial light -> and are loaded onto a ...

-- We have revised the text accordingly.

5. a reconfigurable diffractive decoder

I suggest the authors remove jargon and simply call this another spatial light modulator. This is also true for the following paragraph. I would strongly prefer the authors call it here an SLM and the reconfigurable diffractive decoder in the sections that are more conceptual and less technical

-- We thank the referee for this thoughtful suggestion. In this revision, we improved the experimental setup, where the diffractive decoder (formerly a phase light modulator, PLM) was replaced by a spatial light modulator. To improve clarity, we have updated our text to use “*another SLM*” in sections that describe the experimental details, while retaining “reconfigurable diffractive decoder” in conceptual sections to emphasize the potential use of fabricated surfaces and/or more complex multi-layer diffractive designs. Accordingly, we have revised the manuscript in the **Results section** as follows:

“...After passing through a beam splitter, the optical fields modulated by the encoded phase patterns corresponding to the generative optical seeds are processed by another SLM serving as a fixed/static decoder layer.”

6. p-values: what are p-values? I don't think they were introduced

-- We thank the referee for this valuable suggestion. P-values are associated with t-test and are used to assess the statistical significance of a potential improvement of our optical generative model in terms of a performance metric. Following the referee's suggestion, we have accordingly revised the corresponding parts in the **Results section**:

“...We also carried out a t-test [63] between the optically generated image data and the original dataset, with p-values used to assess the statistical significance of improvements in the IS metric (see Fig. 3c). The higher IS values, combined with small p-values [63] of <0.05, indicate that our snapshot optical image generative models created statistically more diverse images compared to the original datasets.

7. Testing the accuracy of the optical output

In order to evaluate the validity of the newly optically generated data, the authors apply it to a network that was previously trained for MNIST classification and argue that the small drop in classification accuracy show that the generated data is representative of the original data.

I get the basic of the argument, and I would say it is kind of valid – yet it is a bit indirect. Why did the authors not take a generative adversarial approach? Did you try and what was the result?

8. ‘on the original MNIST dataset exhibit some overfitting [...]’

I am not sure I agree with this line of argument. If the drop by 5.8% of the generated images is due to overfitting, then the same overfitting should happen using the original MNIST data. Otherwise this drop has to be associated to the fact that the generated data are not exactly as the original in their distribution?

-- We thank the referee for these insightful comments (#7 and #8) regarding the evaluation of our optically generated image data; these 2 related comments will be addressed together. We initially chose to use pre-trained MNIST classification networks because they offer a straightforward and **interpretable** way to assess the representativeness of the generated image data based on the image classification performance. This approach provides a practical metric for evaluating the effectiveness of our optical generative model. In addition, we employed widely recognized metrics—the Fréchet Inception Distance (FID) and Inception Score (IS)—to directly assess the quality and diversity of the generated images.

For the **generative adversarial** approach to evaluate the generated data, we directly used the discriminator of the FC digital generator with $L_d = 9$ in the **Supplementary Fig. S2**. However, through a comprehensive evaluation of this discriminator, we observed that the discriminator exhibited overfitting and was specifically optimized for its corresponding generator, thus demonstrating limited generalizability in assessing the quality of the generated data across different contexts.

In order to provide a more direct, comprehensive and easier-to-follow assessment, **we revised our testing method in the main text by training three sets of ten binary classifiers**, each based on a convolutional neural network architecture. **The first set was trained using 100% standard MNIST data, while the second set was trained on a 50%-50% mixed dataset composed of standard and optically generated image data and the third set was trained on 100% optically generated image data.** Each classifier was tasked with recognizing a specific handwritten digit, and all the training sets contained the same number of samples to be fair. These classifiers were then evaluated/tested **exclusively on the standard MNIST dataset**, with the resulting classification accuracies presented in **Fig. 3e**. The model trained with the mixed dataset or purely generated image dataset exhibited only a negligible drop in accuracy, indicating that the handwritten digits generated by the snapshot optical generative model can indeed replace standard data (the original dataset) for pattern recognition tasks.

These points have been added to the **Results section** of our revised manuscript, quoted below:

“...To further evaluate the effectiveness of the snapshot optical generative models, we trained three sets of ten binary classifiers, each based on a convolutional neural network architecture [33]. The first set of classifiers was trained using only the standard MNIST training data; the second set was trained on a 50-50% mixed dataset composed of standard and optically generated image data, while the third set was trained on 100% optical generated data (see the

Methods section). Each classifier was tasked with recognizing a specific handwritten digit, and all the training sets had the same number of samples. These classifiers were then blindly evaluated exclusively on the standard MNIST dataset, with the classification accuracies presented in Fig. 3e. The classifiers trained using 100% generated image data achieved on average a classification accuracy of 99.18% (green curve in Fig. 3e), showing a small average decrease of 0.4% compared to the standard MNIST data-based training results (blue curve in Fig. 3e)."

"...To evaluate the quality of snapshot optical image generation, IS [49] and FID [50] indicators were used to quantify the diversity and fidelity of the generated images compared to the original distributions. For the class conditioned generation, e.g., handwritten digits, we further examined the effectiveness of snapshot optically generated images by comparing the classification accuracy of individual binary classifiers trained on different dataset compositions. As shown in Fig. 3e, each binary classifier, based on the same convolutional neural network architecture, was trained to determine whether a given handwritten digit belongs to a specific digit/class. The standard MNIST dataset, the 50%-50% mixed dataset, and the optically generated image dataset each contained 5K images per target digit and 5K non-target digits, where the non-target digits were sampled uniformly from the remaining classes..."

9. Training of the electronic decoder

Did the authors train the decoder as well every time they trained an electronic decoder, or did they use the encoder optimized for optical decoding?

-- We thank the referee for this question. To clarify, our system does not include an electronic decoder. Instead, it consists of an electronic encoder and an optical decoder, which were jointly trained to optimize the overall performance of the generative model. This joint training ensures efficient and high-quality image generation of our optical generative model. This training procedure is described in the **Results section**:

"..., where we first train a teacher digital generative model based on the Denoising Diffusion Probabilistic Model (DDPM) to learn the target data distribution [4]. Once trained, the learned DDPM is frozen, and it continuously generates the noise/image data pairs that are used to train the snapshot optical generative model. The shallow digital phase encoder and the optical generative model are jointly trained, enabling the model to efficiently learn the target distribution with a simple and reconfigurable architecture."

Referee #3:

A. Summary of the key results

This manuscript presents an optical neural network model implemented with a spatial light modulator (SLM) and phase light modulator (PLM), leveraging free-space light propagation for image generation.

The authors demonstrate novel image synthesis across several datasets and report a 3-fold reduction

in floating-point operations (FLOPs) compared to fully-digital fully-connected (FC) neural networks of greater depth. They claim an inference time of less than 1 ns for generating a single 32×32 image.

Additionally, the manuscript introduces the concept of multi-color and iterative optical generative models through numerical simulations.

The proposed architecture comprises a digital encoder that maps Gaussian noise to SLM phase patterns, followed by light propagation through an optimized diffractive optical layer (implemented with PLM) to produce images. The training strategy involves using a diffusion model to learn the data distribution, generating noise-image training pairs, and subsequently optimizing the digital encoder and diffractive layer in a supervised scheme.

-- We sincerely thank the reviewer for their careful review of our manuscript and for providing constructive and comprehensive feedback which helped us further improve our manuscript.

B. Originality and significance

Conceptually, the model is very similar to the previous thread of work in diffractive optical neural network, e.g. as described in this review:

*Hu, J., Mengu, D., Tzarouchis, D.C. et al. Diffractive optical computing in free space. Nat Commun **15**, 1525 (2024). <https://doi.org/10.1038/s41467-024-45982-w>*

Specifically, **Fig. 7: Hybrid optical-electronic networks for intelligent free-space processing** shows very similar model structure. All utilizing the combination of electronic neural network (FC perceptron) and optical neural network (diffractive layers).

-- We thank the referee for raising this important point, which provides an opportunity for us to better clarify the innovative contributions of our work compared to previous implementations, including the ones that are covered by our former Review paper cited above by the referee.

It is crucial to highlight that this is the first demonstration of novel image generation using a diffractive optical network. Previous optical networks were primarily applied to tasks such as imaging, classification, and deterministic operations. **In contrast, our work distinguishes itself by enabling the generation of diverse novel images from various noise inputs, a capability not addressed in prior studies.** Furthermore, the optical generative model also can be trained as a DDPM, without the guidance of a teacher DDPM, as shown in **Figs. 5 and 6**. Through this successive denoising process, a lightweight optical generative model was shown to capture the data distribution and generate high quality images. In addition to these, the phase encoding model employed in our approach provides an effective nonlinear information encoding mechanism (since linear combinations of phase patterns at the input do not create complex fields or intensity patterns at the output that can be represented as a linear superposition of the individual outputs) and this significantly enhances the capabilities of the diffractive decoding layer. The effectiveness of and performance improvements due to this phase encoding strategy was demonstrated in **Supplementary Figs. S6b and S26**, where we compared it with the amplitude and intensity encoding of input information. The visual quality and the lower FID

metrics indicate the superior performance of phase encoding on MNIST generation. For higher resolution Van Gogh-style artwork generation, the amplitude encoding or intensity encoding failed to produce consistent high-quality results, with phase encoding delivering the most compelling artistic outputs.

Accordingly, we included these points in the **Introduction section and Discussion section**:

“...The presented framework is highly flexible since different generative optical models targeting different data distributions share the same optical architecture with an optimized diffractive decoder that is fixed/static for each task, synthesizing countless novel images using optical generative seeds phase-encoded from random noise.”

“...This work presents the demonstration of snapshot optical image generation from noise patterns by leveraging a diffractive network architecture inspired by diffusion models. Earlier free-space-based optical networks primarily focused on tasks such as computational imaging, object/scene classification, or deterministic operations [68-78]. In contrast, our framework introduces the ability to optically generate diverse novel images from noise, showcasing a highly desired "creative" image generation capability that extends beyond the scope of prior studies.”

*“...Moreover, it is important to note that the phase encoding strategy employed by the optical generative model provides an effective nonlinear information encoding mechanism since linear combinations of phase patterns at the input do not create complex fields or intensity patterns at the output that can be represented as a linear superposition of the individual outputs. In fact, this phase encoding strategy significantly enhances the capabilities of the diffractive decoding layer; for comparison, we trained optical generative models using amplitude encoding or intensity encoding, as presented in **Supplementary Fig. S26**, which further highlights the advantages of phase encoding with its superior performance as quantified by the lower FID scores on generated handwritten digit images. Similarly, for the generation of Van Gogh-style novel artworks, the optical generative models using amplitude encoding or intensity encoding failed to produce consistent high-quality and high-resolution output images, as shown in **Supplementary Fig. S26**, whereas the phase encoding strategy successfully generated Van Gogh-style novel artworks. These comparisons clearly underscore the critical role of phase encoding in the optical generative model.”*

We would like to also note that, while the digital diffusion model (serving as the teacher model) is indeed important, the role of the optical diffractive decoder is also crucial. Furthermore, the teacher diffusion model typically takes 1000 steps to generate a novel image, whereas the optical generative model results are achieved in a snapshot. As shown in Supplementary Fig. S6b as well as Supplementary Figs. S18-S19, models without an optical decoder (i.e., utilize free-space propagation only) fail to generate novel images despite the presence of the digital diffusion model-based teacher and the same digital encoder, which demonstrates that the diffractive decoder surface plays a vital role in improving the visual quality of the snapshot-generated novel optical images.

These points have also been stated in the **Results section** of the revised manuscript:

*"...We also compared the architecture of our snapshot optical generative models with respect to a free-space propagation-based optical decoding model, where the diffractive decoder was removed; see **Supplementary Figs. S6a-b**. The results of this comparison demonstrate that the diffractive decoder surface plays a vital role in improving the visual quality of the generated novel images."*

*"...Additional comparisons in **Supplementary Fig. S18** reveal the diffractive decoder's superior performance over free-space-based image decoding, both using the same digital encoder architecture. Notably, while the free-space-based decoder completely failed in some cases, the diffractive decoder achieved stable image generation, with much better image quality at the output. As expected, we observed a numerical aperture-related minor degradation in image resolution when increasing the SLM-to-decoder distance to match our experimental conditions (see **Supplementary Fig. S18** vs. **Supplementary Fig. S19**); however, the diffractive decoder-based approach still maintains stable image generation performance compared to free-space-based decoding, which fails image generation in various cases despite using the same digital encoder, as shown in **Supplementary Fig. S19**."*

Finally, our new numerical and experimental results in the revised manuscript provide strong evidence of the model's advantages. For instance, we have demonstrated high-resolution image generation in **the style of Van Gogh paintings/artworks**—both monochrome and color. These results, along with comprehensive comparisons between diffractive and free-space decoders (see **Supplementary Figs. S17–S26**), further validate the effectiveness and creative capacity of our approach.

These new results have been added to the **Results section** and **Discussion section**:

*"...We further extended our experimental results for the optical generative models to create higher resolution images in **the style of Van Gogh artworks**, which were also demonstrated using the same setup shown in **Fig. 7b**. As illustrated in **Fig. 9**, we experimentally demonstrated snapshot monochrome image generation for Van Gogh-style novel artworks using a digital encoder paired with a jointly-trained diffractive decoder. The architecture of the digital encoder and the processing pipeline are shown in **Supplementary Fig. S17**. Additional comparisons in **Supplementary Fig. S18** reveal the diffractive decoder's superior performance over free-space-based image decoding, both using the same digital encoder architecture. Notably, while the free-space-based decoder completely failed in some cases, the diffractive decoder achieved stable image generation, with much better image quality at the output. As expected, we observed a numerical aperture-related minor degradation in image resolution when increasing the SLM-to-decoder distance to match our experimental conditions (see **Supplementary Fig. S18** vs. **Supplementary Fig. S19**); however, the diffractive decoder-based approach still maintains stable image generation performance compared to free-space-based decoding, which fails image generation in various cases despite using the same digital encoder, as shown in **Supplementary Fig. S19**."*

*By further increasing the number of digital encoder parameters (see **Supplementary Table S1**), we can improve the resolution and image quality of the optically generated Van Gogh-style novel artworks that are created in a snapshot; detailed comparisons are provided in **Supplementary***

Fig. S20 for the number of trainable parameters spanning 44M to 580M. Figures 10 and 11 show our experimental results for higher-resolution monochrome and color (RGB) image generation using a digital encoder with 580M parameters. The monochrome images of Van Gogh-style novel artworks were generated with an illumination wavelength of 520 nm, while the color images used sequential wavelengths of {450,520,638} nm for B, G and R channels, respectively. In Fig. 10, the left three columns display the results where the snapshot images created by the optical generative model in a single pass closely resemble those produced by the digital diffusion model (i.e., the teacher model with 1.07 Billion trainable parameters and 1000 inference steps per image), demonstrating the consistency of our image generation process with respect to the teacher diffusion model. Conversely, the three right columns, highlighted within the orange box, showcase the optical model's ability to generate diverse images that differ from those of the teacher digital diffusion model, illustrating its creative variability at the output; also see additional experimental results for more Van Gogh-style novel artworks supporting our conclusions in **Supplementary Figs. S21-S22**.

For the multi-color Van Gogh-style novel artwork generation, phase-encoded generative seed patterns at each wavelength channel were generated and sequentially loaded onto the SLM. Under the illumination of corresponding wavelengths, multi-color images were generated through a fixed/static diffractive decoder and merged digitally; stated differently, the same decoder state was shared across all the illumination wavelengths for all the novel image generation. Figure 11 presents the multi-color Van Gogh-style novel artwork generation results, including artistic examples that either match or differ from the outputs of the teacher digital diffusion model, which used 1.07 Billion trainable parameters and 1000 inference steps per image generation. Although slight chromatic aberrations were observed likely due to the achievable color space of the selected wavelengths and the camera's response function — the generated high-resolution color images maintained high quality. Additional experimental results of Van Gogh-style novel color artworks are provided in **Supplementary Figs. S23-S24**.

For the generation of colorful Van Gogh-style artworks, we also conducted performance comparisons for the optical generative model, reduced-size diffusion model (matching the size of our phase encoder), and the pretrained teacher diffusion model, as shown in **Supplementary Fig. S25**. Compared to our optical generative model, the reduced-size diffusion model that matches the size of our phase encoder produced inferior images with limited semantic details despite using 1000 inference steps. The optical generative model outputs closely match the teacher diffusion model (which also used 1.07B trainable parameters with 1000 inference steps).

As for significance, my primary concern lies in the framing of this work as “image generation.” The task presented here is more accurately described as learning a mapping from noise to images, with the hope of achieving some degree of generalization. While this is a valid machine learning task, it does not meet the standard expectations for true image generation in the context of modern generative models.

The diffusion model provides a theoretical foundation for approximating the target data distribution. However, the authors' approach essentially adds another layer of approximation, this time using an end-to-end supervised training scheme with a basic FC perceptron.

-- We thank the reviewer for their thoughtful feedback and for raising this important point. We agree

that our approach involves learning a mapping from noise to images, and we believe this mapping itself represents a fundamental capability of novel image generation. This process is central to many modern generative frameworks, including generative adversarial models, diffusion models, etc., where the ultimate goal is to approximate the target data distribution from a general source distribution.

In our work, we extended this concept by implementing an optical generative model trained by being guided by a digital diffusion model. **Importantly, our new results in the revised manuscript both experimentally and numerically demonstrate the distinctiveness of the outputs produced by the optical generative model (student) compared to those of the diffusion model (teacher). As shown in the orange boxes of the newly added Figs. 10-11 as well as Supplementary Fig. S21, the image generation results of the optical model exhibit high quality and valid semantic information while differing meaningfully from the teacher model's outputs.** These experimental and numerical results demonstrate that our framework achieves true novel image generation beyond mere replication of the teacher model.

These points have been added to the **Results section**:

*“...**Figures 10 and 11** show our experimental results for higher-resolution monochrome and color (RGB) image generation using a digital encoder with 580M parameters. The monochrome images of Van Gogh-style novel artworks were generated with an illumination wavelength of 520 nm, while the color images used sequential wavelengths of {450, 520, 638} nm for B, G and R channels, respectively. In **Fig. 10**, the left three columns display the results where the snapshot images created by the optical generative model in a single pass closely resemble those produced by the digital diffusion model (i.e., the teacher model with 1.07 Billion trainable parameters and 1000 inference steps per image), demonstrating the consistency of our image generation process with respect to the teacher diffusion model. Conversely, the three right columns, highlighted within the orange box, showcase the optical model's ability to generate diverse images that differ from those of the teacher digital diffusion model, illustrating its creative variability at the output; also see additional experimental results for more Van Gogh-style novel artworks supporting our conclusions in **Supplementary Figs. S21-S22.**”*

*“...For the multi-color Van Gogh-style novel artwork generation, phase-encoded generative seed patterns at each wavelength channel were generated and sequentially loaded onto the SLM. Under the illumination of corresponding wavelengths, multi-color images were generated through a fixed/static diffractive decoder and merged digitally; stated differently, the same decoder state was shared across all the illumination wavelengths for all the novel image generation. **Figure 11** presents the multi-color Van Gogh-style novel artwork generation results, including artistic examples that either match or differ from the outputs of the teacher digital diffusion model, which used 1.07 Billion trainable parameters and 1000 inference steps per image generation. Although slight chromatic aberrations were observed likely due to the achievable color space of the selected wavelengths and the camera's response function — the generated high-resolution color images maintained high quality. Additional experimental results of Van Gogh-style novel color artworks are provided in **Supplementary Figs. S23-S24.**”*

*For the generation of colorful Van Gogh-style artworks, we also conducted performance comparisons for the optical generative model, reduced-size diffusion model (matching the size of our phase encoder), and the pretrained teacher diffusion model, as shown in **Supplementary Fig. S25**. Compared to our optical generative model, the reduced-size diffusion model that matches the size of our phase encoder produced inferior images with limited semantic details despite using 1000 inference steps. The optical generative model outputs closely match the teacher diffusion model (which also used 1.07B trainable parameters with 1000 inference steps).”*

Furthermore, **Fig. 8 along with Supplementary Fig. S16 and Supplementary Videos 3-9** also help us visualize the interpolation results between different noise inputs. As stated in the manuscript, these interpolation results “*exhibit smooth and coherent transitions between different handwritten digits, indicating that the snapshot optical generative model learned a continuous and well-organized latent space representation*”. This interpolation behavior further validates the effectiveness of our framework and indicates the knowledge acquired by the teacher model has been distilled to the optical generative model.

The authors demonstrate advantages of the optical model in terms of reducing FLOPs for this specific task. However, this does not compensate for the lack of theoretical rigor and poor performance metrics. For example, for the MNIST dataset, the reported Frechet Inception Distance (FID) scores in this study (~196 in Fig. 3d) are over an order of magnitude worse than those achieved by modern GAN models (~13 in table II):

Lazcano, D., Franco, N. F., & Creixell, W. (2021). Hgan: Hyperbolic generative adversarial network. IEEE Access, 9, 96309-96320.

-- We have improved the performance of our models. With an optimized training strategy outlined in our revised Methods section that better utilized the phase encoding space, our optical generative models could create higher contrast images. For example, we were able to reduce the FID score for the MNIST dataset to <90, as shown in the **revised Fig. 3** (numerical results). Using our improved experimental set-up, the FID scores for the experimental image generation were calculated as 131.08 for the MNIST dataset and 180.57 for the Fashion MNIST dataset, as shown in the **revised Fig. 7**. While these experimental values are slightly higher than those obtained from numerical simulations, they still demonstrate the effectiveness of the optical generative model and validate its practical applicability.

We also report detailed comparisons to show the improvement of the optimized optical generative experimental set-up, as quoted from our revised manuscript:

*“... we also experimentally evaluated the snapshot optical image generation under a limited phase encoding space (e.g., $0-\pi/2$ vs. $0-2\pi$) and a limited decoder bit depth (e.g., 4 bit-depth vs. 8 bit-depth) using a restricted optical setup; see **Supplementary Figs. S14a, S14b and S15**. The performance comparisons between the experimental results of **Fig. 7 and Supplementary Fig. S14** show the significance of employing a large phase bit depth for the diffractive decoder as well as increasing the encoding phase range of the input SLM.”*

A key component of our results is that the image generation is achieved in a snapshot. For example, for the generation of colorful Van Gogh-style artworks, we conducted performance comparisons for the optical generative model, reduced-size digital diffusion model (matching the size of our phase encoder), and the pretrained teacher diffusion model, as shown in **Supplementary Fig. S25**. Compared to our optical generative model, the reduced-size diffusion model that matches the size of our phase encoder produced inferior images with limited semantic details despite using 1000 inference steps. The optical generative model outputs closely match the teacher diffusion model (which also used 1.07B trainable parameters with 1000 inference steps).

See Sec. E for more questions about the claimed advantages regarding time performance, scalability, and energy efficiency.

C. Data & methodology: validity of approach, quality of data, quality of presentation

I think all results generated through numerical simulation should be explicitly labeled and clearly distinguished from experimental results. Are the results in Figs. 3, 5, and 7 based on digital simulations of the diffractive decoding behavior without hardware implementation? If it is the case, please label it; if not, then the section titled “Experimental demonstration of snapshot optical generative models” and the label in Fig. 8c as “Experimental snapshot generation” are confusing and misleading.

-- We thank the reviewer for raising this important point. Accordingly, we have revised the text and figure captions (marked yellow) to explicitly specify whether the results are numerical or experimental. Additionally, we revised **Figs. 7 and 8** and included new **Figs. 9-11 as well as Supplementary Figs 21-24**, which all report experimental results compared with numerical simulations to better illustrate the performance of our optical generative model, both numerically and experimentally.

Some of these new results are included in the **Results section** of the revised manuscript:

*“...We further extended our experimental results for the optical generative models to create higher resolution images in the style of Van Gogh artworks, which were also demonstrated using the same setup shown in **Fig. 7b**. As illustrated in **Fig. 9**, we experimentally demonstrated snapshot monochrome image generation for Van Gogh-style novel artworks using a digital encoder paired with a jointly-trained diffractive decoder. The architecture of the digital encoder and the processing pipeline are shown in **Supplementary Fig. S17**. Additional comparisons in **Supplementary Fig. S18** reveal the diffractive decoder's superior performance over free-space-based image decoding, both using the same digital encoder architecture. Notably, while the free-space-based decoder completely failed in some cases, the diffractive decoder achieved stable image generation, with much better image quality at the output. As expected, we observed a numerical aperture-related minor degradation in image resolution when increasing the SLM-to-decoder distance to match our experimental conditions (see **Supplementary Fig. S18** vs. **Supplementary Fig. S19**); however, the diffractive decoder-based approach still maintains stable image generation performance compared to free-space-based decoding, which fails image generation in various cases despite using the same digital encoder, as shown in **Supplementary Fig. S19**.*”

By further increasing the number of digital encoder parameters (see **Supplementary Table S1**), we can improve the resolution and image quality of the optically generated Van Gogh-style novel artworks that are created in a snapshot; detailed comparisons are provided in **Supplementary Fig. S20** for the number of trainable parameters spanning 44M to 580M. **Figures 10 and 11** show our experimental results for higher-resolution monochrome and color (RGB) image generation using a digital encoder with 580M parameters. The monochrome images of Van Gogh-style novel artworks were generated with an illumination wavelength of 520 nm, while the color images used sequential wavelengths of {450, 520, 638} nm for B, G and R channels, respectively. In **Fig. 10**, the left three columns display the results where the snapshot images created by the optical generative model in a single pass closely resemble those produced by the digital diffusion model (i.e., the teacher model with 1.07 Billion trainable parameters and 1000 inference steps per image), demonstrating the consistency of our image generation process with respect to the teacher diffusion model. Conversely, the three right columns, highlighted within the orange box, showcase the optical model's ability to generate diverse images that differ from those of the teacher digital diffusion model, illustrating its creative variability at the output; also see additional experimental results for more Van Gogh-style novel artworks supporting our conclusions in **Supplementary Figs. S21-S22**.

For the multi-color Van Gogh-style novel artwork generation, phase-encoded generative seed patterns at each wavelength channel were generated and sequentially loaded onto the SLM. Under the illumination of corresponding wavelengths, multi-color images were generated through a fixed/static diffractive decoder and merged digitally; stated differently, the same decoder state was shared across all the illumination wavelengths for all the novel image generation. **Figure 11** presents the multi-color Van Gogh-style novel artwork generation results, including artistic examples that either match or differ from the outputs of the teacher digital diffusion model, which used 1.07 Billion trainable parameters and 1000 inference steps per image generation. Although slight chromatic aberrations were observed — likely due to the achievable color space of the selected wavelengths and the camera's response function — the generated high-resolution color images maintained high quality. Additional experimental results of Van Gogh-style novel color artworks are provided in **Supplementary Figs. S23-S24**.

For the generation of colorful Van Gogh-style artworks, we also conducted performance comparisons for the optical generative model, reduced-size diffusion model (matching the size of our phase encoder), and the pretrained teacher diffusion model, as shown in **Supplementary Fig. S25**. Compared to our optical generative model, the reduced-size diffusion model that matches the size of our phase encoder produced inferior images with limited semantic details despite using 1000 inference steps. The optical generative model outputs closely match the teacher diffusion model (which also used 1.07B trainable parameters with 1000 inference steps).”

As for performance evaluation:

- **Fig. 3e***: The main text reported a 94.2% correctness, but the figure caption mentions “classification confidence,” yet the method of calculation is unclear. The authors should specify how this metric is defined.

-- We thank the referee for this valuable request for clarification. We initially chose to use pre-trained MNIST classification networks because they offer a straightforward and **interpretable** way to assess the representativeness of the generated image data based on the image classification performance. This approach provides a practical metric for evaluating the effectiveness of our optical generative model. In addition, we employed widely recognized metrics—the Fréchet Inception Distance (FID) and Inception Score (IS)—to directly assess the quality and diversity of the generated images.

In order to provide a more comprehensive and easier-to-follow assessment, we have revised our testing method in the main text by training three sets of ten binary classifiers, each based on a convolutional neural network architecture. The first set was trained using 100% standard MNIST data, while the second set was trained on a 50%-50% mixed dataset composed of standard and optically generated image data and the third set was trained on 100% optically generated image data. Each classifier was tasked with recognizing a specific handwritten digit, and all the training sets contained the same number of samples to be fair. These classifiers were then evaluated/tested **exclusively on the standard MNIST dataset**, with the resulting classification accuracies presented in **Fig. 3e**. **The model trained with the mixed dataset or purely generated image dataset exhibited only a negligible drop in accuracy, indicating that the handwritten digits generated by the snapshot optical generative model can indeed replace standard data (the original dataset) for pattern recognition tasks.**

These points have been added to the **Results section** of our revised manuscript, quoted below:

*“...To further evaluate the effectiveness of the snapshot optical generative models, we trained three sets of ten binary classifiers, each based on a convolutional neural network architecture [33]. The first set of classifiers was trained using only the standard MNIST training data; the second set was trained on a 50-50% mixed dataset composed of standard and optically generated image data, while the third set was trained on 100% optical generated data (see the Methods section). Each classifier was tasked with recognizing a specific handwritten digit, and all the training sets had the same number of samples. These classifiers were then blindly evaluated exclusively on the standard MNIST dataset, with the classification accuracies presented in **Fig. 3e**. **The classifiers trained using 100% generated image data achieved on average a classification accuracy of 99.18% (green curve in Fig. 3e), showing a small average decrease of 0.4% compared to the standard MNIST data-based training results (blue curve in Fig. 3e).**”*

*“...To evaluate the quality of snapshot optical image generation, IS [49] and FID [50] indicators were used to quantify the diversity and fidelity of the generated images compared to the original distributions. For the class conditioned generation, e.g., handwritten digits, we further examined the effectiveness of snapshot optically generated images by comparing the classification accuracy of individual binary classifiers trained on different dataset compositions. As shown in **Fig. 3e**, each binary classifier, based on the same convolutional neural network architecture, was trained to determine whether a given handwritten digit belongs to a specific digit/class. The standard MNIST dataset, the 50%-50% mixed dataset, and the optically generated image dataset each contained 5K images per target digit and 5K non-target digits, where the non-target digits were sampled uniformly from the remaining classes...”*

- **Fig. 5f**: The criteria for defining “generation failure” are not sufficiently explained. While Fig. 5b and 5c qualitatively illustrate failure cases, some amount of color noise (not as strong) is also observed in the other generated samples. The main text does not clearly state how generation failure is defined, nor does it address whether these criteria are applied consistently. A quantitative criterion or threshold for failure should be provided for reproducibility and clarity.

-- To address this important comment of the referee, we have incorporated a noise variance-based metric to identify failure cases. Specifically, we use the noise variance (σ^2) of the generated images as the failure detection criterion. Generated samples with a noise variance exceeding an empirical threshold of $\sigma^2 > 0.015$ are classified as generation failures. We observed that this metric consistently identified failure cases, providing a reproducible and objective measure to evaluate image generation quality.

We have included this criterion in the **Results section** of the revised manuscript and **updated Fig. 4(c)** accordingly to reflect this quantitative threshold; we have also added a new **Supplementary Fig. S10** to better explain these points. Quoted from our revised Results section:

*“...Additionally, some failed image generation cases are highlighted with red boxes in the bottom-right corner of **Figs. 4b and c**. These rare cases were automatically identified based on a noise variance criterion, where the generated images with estimated noise variance (σ^2) exceeding an empirical threshold of 0.015 were classified as generation failures (see **Supplementary Fig. S10**) [67].”*

- The claim in the main text that image inference time is less than 1 ns requires further justification. How was this value determined? Does it also include the time cost by the shallow digital encoder and SLM loading pattern, in addition to light propagation? Also, the A/D conversion time in the camera also potentially needs to be accounted for. If these additional factors were not explicitly discussed and addressed, the claim of <1 ns should be revised or clarified to avoid overstating the system’s performance.

-- We thank the reviewer for raising this important point. We note that “<1 ns” refers specifically to the optical propagation time from the optical seed plane to the image plane, **indicating a thin optical volume for the diffractive decoder**, which excludes other factors such as the digital encoder’s computation time and the response time of the SLM. To avoid confusing our readers, we have removed the emphasis on this travel time (<1 ns) and accordingly revised the **Introduction section** and **Results section** of our manuscript to better explain these points, quoted below:

*“...; **the refresh rate is limited by the frame-rate of the SLM that displays the pre-calculated optical generative seeds**. This image synthesis process from the optical seed plane to the output image requires power for the illumination light and the SLM operation, without additional computing power needed. The optical part of the computation required for snapshot image generation is carried out entirely through free-space propagation of light via an optimized and fixed (i.e., static) diffractive decoder.”*

“...Since the encoded phase patterns can be pre-calculated by feeding random Gaussian noise to a shallow encoder network, they can be randomly accessed on demand, where novel images can be locally synthesized by the optical generative model. **The overall inference time is constrained by the SLM loading time**, which can be minimized by employing faster phase light modulators (PLMs) or SLMs with frame rates exceeding 1 kHz [56-58]. ”

Furthermore, regarding additional factors related to digital-to-optical and optical-to-digital conversion and related interfaces, we have added a comprehensive discussion of these factors in our revised manuscript, Discussion section:

“...The presented optical generative models comprise four primary components: the electronic encoder network, the input SLM, the illumination light, and the diffractive decoder, which are collectively optimized for novel image display. The electronic encoder used for MNIST and Fashion MNIST datasets consists of three fully connected layers, and it requires 6.29 MFLOPs per image, with an energy cost of $\sim 0.5\text{-}5.5$ pJ/FLOP [80-81], resulting in an energy consumption of 0.003-0.033 mJ per image. This energy consumption increases to $\sim 0.28\text{-}3.08$ J and $\sim 1.13\text{-}12.44$ J per image for the Van Gogh-style artworks reported in **Fig. 9** and **Figs. 10-11**, respectively. The input SLM, with a power range of 1.9–3.5 W, consumes $\sim 30\text{-}58$ mJ per image at a 60 Hz refresh rate. This SLM-related energy consumption can be reduced to <2.5 mJ per image using a state-of-the-art SLM [56-58]. The diffractive decoder has a similar energy consumption if a second SLM is used; however, its contribution would become negligible if a static decoder (e.g., a passive fabricated surface/layer) is employed. As for the illumination light, the energy consumption per wavelength channel can be estimated to be less than 0.8 mJ per image [82], which is negligible compared to other factors. If the generated images were to be digitized by an image sensor chip (e.g., a 5-10 mega-pixel **CMOS imager**), this would also add an extra energy consumption of $\sim 2\text{-}4$ mJ per image. Consequently, the overall energy consumption for generated images intended for human perception — excluding the need for a digital camera — is dominated by the SLM-based power in lower resolution image generation, whereas the digital encoder power consumption becomes the dominant factor for higher resolution image generation tasks such as the Van Gogh-style artworks. In contrast, GPU-based generative systems using a DDPM model have different energy characteristics that are dominated by the diffusion and successive denoising processes (involving e.g., 1000 steps). For example, the computational requirements for generating MNIST/Fashion MNIST and Van Gogh-style artwork images using a digital DDPM model amount to approximately 287.68 GFLOPs and 530.26 TFLOPs, respectively, corresponding to about 0.14–1.58 J per image for MNIST/Fashion-MNIST and 265–2916 J per image for Van Gogh-style artworks. Furthermore, if the generated images must be displayed on a monitor for human perception, additional energy consumption is incurred—typically between $\sim 13\text{-}500$ mJ per image at a 60 Hz refresh rate.

Overall, these comparisons reveal that if the novel image information to be generated will be stored and processed/harnessed in the digital domain, optical generative models would face additional power and speed penalties due to the digital-to-analog and analog-to-digital conversion steps that would be involved in the optical setup. However, if the image information to be generated will remain in the analog domain for direct visualization by human observers (e.g., in a near-eye or head-mounted display), the optical generative seeds can be pre-calculated with a modest energy consumption per seed, as detailed above. Furthermore, the static diffractive decoder surface can be fabricated using optical lithography or two-photon polymerization-based nano-fabrication methods, which would optically generate snapshot novel images within the display setup. This could enable compact and cost-effective image

generators—such as "optical artists"—by replacing the back-end diffractive decoder with a fabricated passive surface. This setup would allow for the snapshot generation of countless novel images, including various forms of artwork, using simpler local optical hardware. From the perspective of a digital generative model, for comparison, one could also use a standard image display along with pre-computed and stored novel images created through, e.g., a digital DDPM model; this, however, requires significantly more energy consumption per image generation through the diffusion and successive denoising processes, as discussed earlier. Exploration of optical generative architectures using nano-fabricated surfaces would enable various new applications, especially for image and near-eye display systems, including head-mounted and wearable setups."

- Is the FID score calculated against the original dataset used to train the DDPM? When you refer to "novel" images, are you comparing these to the data used to train the DDPM (denoising diffusion probabilistic model) or to the data generated by the DDPM that was subsequently used to train the optical generative model? I think it is essential to clearly separate the contributions of the DDPM and the optical generative model proposed in this work. If the optical generative model merely imitates the behavior of the DDPM and cannot generate new samples beyond the training samples provided by the DDPM, it would be misleading to describe it as a true generative model.

-- We sincerely appreciate the reviewer for raising this important point. Since the fundamental objective of a generative model is to accurately capture the data distribution of real-world samples, we calculate the FID scores against the original dataset. These points have been added to the **Methods Section**:

"...To evaluate the quality of snapshot optical image generation, IS [49] and FID [50] indicators were used to quantify the diversity and fidelity of the generated images compared to the original distributions."

Furthermore, our new results in the revised manuscript both experimentally and numerically demonstrate the distinctiveness of the outputs produced by the optical generative model (student) compared to those of the digital diffusion model (teacher). As shown in the orange boxes of the newly added Figs. 10-11 as well as Supplementary Fig. S21, the generation results of the optical model exhibit high quality and valid semantic information while differing meaningfully from the teacher diffusion model's outputs. These experimental and numerical results demonstrate that our framework achieves true image generation beyond a mere replication of the teacher model.

These points have been added to the **Results section**:

*"...**Figures 10 and 11** show our experimental results for higher-resolution monochrome and color (RGB) image generation using a digital encoder with 580M parameters. The monochrome images of Van Gogh-style novel artworks were generated with an illumination wavelength of 520 nm, while the color images used sequential wavelengths of {450, 520, 638} nm for B, G and R channels, respectively. In **Fig. 10**, the left three columns display the results where the snapshot images created by the optical generative model in a single pass closely resemble those produced by the digital diffusion model (i.e., the teacher model with 1.07 Billion trainable parameters and 1000 inference steps per image), demonstrating the consistency of our image generation process with respect to the teacher diffusion model. Conversely, the three right*

columns, highlighted within the orange box, showcase the optical model's ability to generate diverse images that differ from those of the teacher digital diffusion model, illustrating its creative variability at the output; also see additional experimental results for more Van Gogh-style novel artworks supporting our conclusions in **Supplementary Figs. S21-S22.**”

“...For the multi-color Van Gogh-style novel artwork generation, phase-encoded generative seed patterns at each wavelength channel were generated and sequentially loaded onto the SLM. Under the illumination of corresponding wavelengths, multi-color images were generated through a fixed/static diffractive decoder and merged digitally; stated differently, the same decoder state was shared across all the illumination wavelengths for all the novel image generation. **Figure 11** presents the multi-color Van Gogh-style novel artwork generation results, including artistic examples that either match or differ from the outputs of the teacher digital diffusion model, which used 1.07 Billion trainable parameters and 1000 inference steps per image generation. Although slight chromatic aberrations were observed likely due to the achievable color space of the selected wavelengths and the camera's response function — the generated high-resolution color images maintained high quality. Additional experimental results of Van Gogh-style novel color artworks are provided in **Supplementary Figs. S23-S24.**”

For the generation of colorful Van Gogh-style artworks, we also conducted performance comparisons for the optical generative model, reduced-size diffusion model (matching the size of our phase encoder), and the pretrained teacher diffusion model, as shown in **Supplementary Fig. S25.** Compared to our optical generative model, the reduced-size diffusion model that matches the size of our phase encoder produced inferior images with limited semantic details despite using 1000 inference steps. The optical generative model outputs closely match the teacher diffusion model (which also used 1.07B trainable parameters with 1000 inference steps).”

Furthermore, **Fig. 8 along with Supplementary Fig. S16 and Supplementary Videos 3-9** also help us visualize the interpolation results between different noise inputs. As stated in the manuscript, these interpolation results “*exhibit smooth and coherent transitions between different handwritten digits, indicating that the snapshot optical generative model learned a continuous and well-organized latent space representation*”. This interpolation behavior further validates the effectiveness of our framework and indicates the knowledge acquired by the teacher model has been distilled to the optical generative model.

We would like to also note that, while the digital diffusion model (serving as the teacher model) is indeed important, the role of the optical diffractive decoder is also crucial. Furthermore, the teacher diffusion model typically takes 1000 steps to generate a novel image, whereas the optical generative model results are achieved in a snapshot. As shown in Supplementary Fig. S6b as well as Supplementary Figs. S18-S19, models without an optical decoder (i.e., free-space propagation only) fail to generate novel images despite the presence of the digital diffusion model-based teacher and the same digital encoder, which demonstrates that the diffractive decoder surface plays a vital role in improving the visual quality of the snapshot-generated novel optical images.

Additionally, as presented in the **newly added Supplementary Fig. 26**, we investigated various optical generative models utilizing different encoding strategies, including phase encoding, amplitude encoding, and intensity encoding. Our findings indicate that these approaches yield suboptimal generation performance compared to the phase encoding strategy. Specifically, for the MNIST image generation task, we calculated the FID for each encoding method, demonstrating that phase encoding consistently achieves superior results. Furthermore, our evaluations on generating Van Gogh-style novel artworks reinforce this observation, with phase encoding delivering the most compelling artistic outputs.

These points have also been stated in the **Results section** of the manuscript:

*"...We also compared the architecture of our snapshot optical generative models with respect to a free-space propagation-based optical decoding model, where the diffractive decoder was removed; see **Supplementary Figs. S6a-b**. The results of this comparison demonstrate that the diffractive decoder surface plays a vital role in improving the visual quality of the generated novel images."*

*"...Additional comparisons in **Supplementary Fig. S18** reveal the diffractive decoder's superior performance over free-space-based image decoding, both using the same digital encoder architecture. Notably, while the free-space-based decoder completely failed in some cases, the diffractive decoder achieved stable image generation, with much better image quality at the output. As expected, we observed a numerical aperture-related minor degradation in image resolution when increasing the SLM-to-decoder distance to match our experimental conditions (see **Supplementary Fig. S18** vs. **Supplementary Fig. S19**); however, the diffractive decoder-based approach still maintains stable image generation performance compared to free-space-based decoding, which fails image generation in various cases despite using the same digital encoder, as shown in **Supplementary Fig. S19**."*

We have also added the related points to the **Discussion section** of the revised manuscript:

*"...Moreover, it is important to note that the phase encoding strategy employed by the optical generative model provides an effective nonlinear information encoding mechanism since linear combinations of phase patterns at the input do not create complex fields or intensity patterns at the output that can be represented as a linear superposition of the individual outputs. In fact, this phase encoding strategy significantly enhances the capabilities of the diffractive decoding layer; for comparison, we trained optical generative models using amplitude encoding or intensity encoding, as presented in **Supplementary Fig. S26**, which further highlights the advantages of phase encoding with its superior performance as quantified by the lower FID scores on generated handwritten digit images. Similarly, for the generation of Van Gogh-style novel artworks, the optical generative models using amplitude encoding or intensity encoding failed to produce consistent high-quality and high-resolution output images, as shown in **Supplementary Fig. S26**, whereas the phase encoding strategy successfully generated Van Gogh-style novel artworks. These comparisons clearly underscore the critical role of phase encoding in the optical generative model."*

D. Appropriate use of statistics and treatment of uncertainties

The manuscript demonstrates an appropriate treatment of uncertainties. The supplementary material includes an ablation study and provides a discussion on the influence of the SLM bit depth and misalignment.

-- We thank the reviewer for their positive feedback and for acknowledging our efforts in addressing uncertainties through ablation studies and related analyses and discussions.

E. Conclusions: robustness, validity, reliability

**Time Performance: ** First, see the comment in Sec. C. Data & methodology to clarify the time performance. While the introduction acknowledges that the refresh rate is constrained by the frame rate of the spatial light modulator (SLM), this limitation is not addressed in the conclusions. For instance, the HOLOEYE LC-R 2500 used in the experiments operates at only 75 Hz, which is orders of magnitude slower than the claimed GHz inference time. For the iterative optical generative models that involve sampling 1,000 timestamps, the total computation time would be at least tens of seconds, excluding the additional overhead for data transmission between the camera and the digital encoder. The conclusions should accurately reflect these constraints.

-- Please see our earlier responses related to some of these points that we have extensively discussed and addressed. Following the referee's recommendation, we have added a comprehensive discussion of these factors in our revised manuscript, Discussion section:

*“... The presented optical generative models comprise four primary components: the electronic encoder network, the input SLM, the illumination light, and the diffractive decoder, which are collectively optimized for novel image display. The electronic encoder used for MNIST and Fashion MNIST datasets consists of three fully connected layers, and it requires 6.29 MFLOPs per image, with an energy cost of ~0.5-5.5 pJ/FLOP [80-81], resulting in an energy consumption of 0.003-0.033 mJ per image. This energy consumption increases to ~0.28-3.08 J and ~1.13-12.44 J per image for the Van Gogh-style artworks reported in **Fig. 9** and **Figs. 10-11**, respectively. The input SLM, with a power range of 1.9–3.5 W, consumes ~30–58 mJ per image at a 60 Hz refresh rate. This SLM-related energy consumption can be reduced to <2.5 mJ per image using a state-of-the-art SLM [56-58]. The diffractive decoder has a similar energy consumption if a second SLM is used; however, its contribution would become negligible if a static decoder (e.g., a passive fabricated surface/layer) is employed. As for the illumination light, the energy consumption per wavelength channel can be estimated to be less than 0.8 mJ per image [82], which is negligible compared to other factors. If the generated images were to be digitized by an image sensor chip (e.g., a 5-10 mega-pixel **CMOS imager**), this would also add an extra energy consumption of ~2–4 mJ per image. Consequently, the overall energy consumption for generated images intended for human perception — excluding the need for a digital camera — is dominated by the SLM-based power in lower resolution image generation, whereas the digital encoder power consumption becomes the dominant factor for higher resolution image generation tasks such as the Van Gogh-style artworks. In contrast, GPU-based generative systems using a DDPM model have different energy characteristics that are dominated by the diffusion and successive denoising processes (involving e.g., 1000 steps). For example, the computational requirements for generating MNIST/Fashion MNIST and Van Gogh-style artwork images using a digital DDPM model amount to approximately 287.68 GFLOPs and 530.26 TFLOPs, respectively, corresponding to about 0.14–1.58 J per image for*

MNIST/Fashion-MNIST and 265–2916 J per image for Van Gogh-style artworks. Furthermore, if the generated images must be displayed on a monitor for human perception, additional energy consumption is incurred—typically between ~13–500 mJ per image at a 60 Hz refresh rate.

Overall, these comparisons reveal that if the novel image information to be generated will be stored and processed/harnessed in the digital domain, optical generative models would face additional power and speed penalties due to the digital-to-analog and analog-to-digital conversion steps that would be involved in the optical setup. However, if the image information to be generated will remain in the analog domain for direct visualization by human observers (e.g., in a near-eye or head-mounted display), the optical generative seeds can be pre-calculated with a modest energy consumption per seed, as detailed above. Furthermore, the static diffractive decoder surface can be fabricated using optical lithography or two-photon polymerization-based nano-fabrication methods, which would optically generate snapshot novel images within the display setup. This could enable compact and cost-effective image generators—such as "optical artists"—by replacing the back-end diffractive decoder with a fabricated passive surface. This setup would allow for the snapshot generation of countless novel images, including various forms of artwork, using simpler local optical hardware. From the perspective of a digital generative model, for comparison, one could also use a standard image display along with pre-computed and stored novel images created through, e.g., a digital DDPM model; this, however, requires significantly more energy consumption per image generation through the diffusion and successive denoising processes, as discussed earlier. Exploration of optical generative architectures using nano-fabricated surfaces would enable various new applications, especially for image and near-eye display systems, including head-mounted and wearable setups.”

We have also included additional text in the **Results section** and **Discussion section** of the revised manuscript to address these points:

“..., Since the encoded phase patterns can be pre-calculated by feeding random Gaussian noise to a shallow encoder network, they can be randomly accessed on demand, where novel images can be locally synthesized by the optical generative model. The overall inference time is constrained by the SLM loading time, which can be minimized by employing faster phase light modulators (PLMs) or SLMs with frame rates exceeding 1 kHz [56-58]. ”

****Scalability:**** The conclusion of scalability is unsubstantiated given the limited scope of the demonstrations. The work is restricted to generating 32×32 images, which is far from representative of modern high-resolution image generation tasks. Furthermore, the scalability of the system is fundamentally constrained by the resolution and bit depth of the SLM and PLM. These hardware limitations significantly impact the system's ability to scale to larger image sizes or more complex tasks.

-- To address these points, we built a new experimental setup with two SLMs (Meadowlark XY Phase Series, pixel pitch: 8 μ m, resolution: 1920 × 1200; HOLOEYE PLUTO-2.1, pixel pitch 8 μ m, resolution 1920 × 1080). With the smaller pixel pitch and 8 bit depth phase modulation of the new setup, better optical generated results were obtained with various levels of resolution. The comparisons between the new experimental results of **Fig. 7** and the older ones of **Supplementary Fig. S14** show the significance of increasing the representation capability of the optical set-up.

Furthermore, to assess the potential impact of phase bit-depth limitations of an optical generative model set-up, we conducted additional numerical analyses by imposing constraints on the phase patterns and the structure of the diffractive decoding layer. To be specific, we set the bit depth of the SLM and/or the diffractive decoder to 4 bits (i.e., 16 discrete levels) during the testing phase. As shown in **Supplementary Fig. S9**, although the models were trained without any bit-depth constraints, they demonstrated acceptable image generation performance. Additionally, we explored a fabrication scenario involving a static decoding layer with only **3 available phase levels (e.g., 0 , $\pi/3$, $2\pi/3$)**. In this case, our model still exhibited robust image generation performance during blind testing, indicating the system's resilience to reduced phase levels. Finally, to mitigate the impact of this physical constraint in the set-up of an optical generative model (for example, in a resource-limited setting), we trained an optical generative model using an 8-bit SLM and a **static decoding layer with 3 discrete phase levels (0 , $\pi/3$, $2\pi/3$)**, which showed notable performance improvements (as shown in **Supplementary Fig. S9**) compared to models trained without considering bit-depth limitations of the set-up.

To further illustrate the potential of the scalability of our method, we have also expanded our experimental results to include **the generation of higher-resolution images with 640×640 pixels**. Both numerical and experimental results for these higher-resolution images are now included in the revised manuscript, as shown in **Figs. 9-11, as well as Supplementary Figs. S21-S24**. These new experimental and numerical results demonstrate the capability of our system to scale to larger image sizes while maintaining quality and robustness.

Correspondingly, we included the related analysis and results in the **Results section** of the revised manuscript:

*“...We further extended our experimental results for the optical generative models to create higher resolution images in the style of Van Gogh artworks, which were also demonstrated using the same setup shown in **Fig. 7b**. As illustrated in **Fig. 9**, we experimentally demonstrated snapshot monochrome image generation for Van Gogh-style novel artworks using a digital encoder paired with a jointly-trained diffractive decoder. The architecture of the digital encoder and the processing pipeline are shown in **Supplementary Fig. S17**. Additional comparisons in **Supplementary Fig. S18** reveal the diffractive decoder's superior performance over free-space-based image decoding, both using the same digital encoder architecture. Notably, while the free-space-based decoder completely failed in some cases, the diffractive decoder achieved stable image generation, with much better image quality at the output. As expected, we observed a numerical aperture-related minor degradation in image resolution when increasing the SLM-to-decoder distance to match our experimental conditions (see **Supplementary Fig. S18** vs. **Supplementary Fig. S19**); however, the diffractive decoder-based approach still maintains stable image generation performance compared to free-space-based decoding, which fails image generation in various cases despite using the same digital encoder, as shown in **Supplementary Fig. S19**.*

*By further increasing the number of digital encoder parameters (see **Supplementary Table S1**), we can improve the resolution and image quality of the optically generated Van Gogh-style novel artworks that are created in a snapshot; detailed comparisons are provided in **Supplementary***

Fig. S20 for the number of trainable parameters spanning 44M to 580M. **Figures 10 and 11** show our experimental results for higher-resolution monochrome and color (RGB) image generation using a digital encoder with 580M parameters. The monochrome images of Van Gogh-style novel artworks were generated with an illumination wavelength of 520 nm, while the color images used sequential wavelengths of {450, 520, 638} nm for B, G and R channels, respectively. In **Fig. 10**, the left three columns display the results where the snapshot images created by the optical generative model in a single pass closely resemble those produced by the digital diffusion model (i.e., the teacher model with 1.07 Billion trainable parameters and 1000 inference steps per image), demonstrating the consistency of our image generation process with respect to the teacher diffusion model. Conversely, the three right columns, highlighted within the orange box, showcase the optical model's ability to generate diverse images that differ from those of the teacher digital diffusion model, illustrating its creative variability at the output; also see additional experimental results for more Van Gogh-style novel artworks supporting our conclusions in **Supplementary Figs. S21-S22**.

For the multi-color Van Gogh-style novel artwork generation, phase-encoded generative seed patterns at each wavelength channel were generated and sequentially loaded onto the SLM. Under the illumination of corresponding wavelengths, multi-color images were generated through a fixed/static diffractive decoder and merged digitally; stated differently, the same decoder state was shared across all the illumination wavelengths for all the novel image generation. **Figure 11** presents the multi-color Van Gogh-style novel artwork generation results, including artistic examples that either match or differ from the outputs of the teacher digital diffusion model, which used 1.07 Billion trainable parameters and 1000 inference steps per image generation. Although slight chromatic aberrations were observed — likely due to the achievable color space of the selected wavelengths and the camera's response function — the generated high-resolution color images maintained high quality. Additional experimental results of Van Gogh-style novel color artworks are provided in **Supplementary Figs. S23-S24**.

For the generation of colorful Van Gogh-style artworks, we also conducted performance comparisons for the optical generative model, reduced-size diffusion model (matching the size of our phase encoder), and the pretrained teacher diffusion model, as shown in **Supplementary Fig. S25**. Compared to our optical generative model, the reduced-size diffusion model that matches the size of our phase encoder produced inferior images with limited semantic details despite using 1000 inference steps. The optical generative model outputs closely match the teacher diffusion model (which also used 1.07B trainable parameters with 1000 inference steps).”

****Energy Efficiency:**** The claim of energy efficiency also requires reconsideration. While the optical component may theoretically operate with low power consumption, the overall system includes energy-intensive components such as the light source, SLM and camera. A more detailed analysis of the system's total power consumption, including both optical and electronic components, is necessary to validate this claim.

-- Following the referee's recommendation, we have added a comprehensive discussion of these factors in our revised manuscript, Discussion section:

“...The presented optical generative models comprise four primary components: the electronic encoder network, the input SLM, the illumination light, and the diffractive decoder, which are collectively optimized for novel image display. The electronic encoder used for MNIST and Fashion MNIST datasets consists of three fully connected layers, and it requires 6.29 MFLOPs per image, with an energy cost of $\sim 0.5\text{--}5.5$ pJ/FLOP [80-81], resulting in an energy consumption of 0.003-0.033 mJ per image. This energy consumption increases to $\sim 0.28\text{--}3.08$ J and $\sim 1.13\text{--}12.44$ J per image for the Van Gogh-style artworks reported in **Fig. 9** and **Figs. 10-11**, respectively. The input SLM, with a power range of 1.9–3.5 W, consumes $\sim 30\text{--}58$ mJ per image at a 60 Hz refresh rate. This SLM-related energy consumption can be reduced to <2.5 mJ per image using a state-of-the-art SLM [56-58]. The diffractive decoder has a similar energy consumption if a second SLM is used; however, its contribution would become negligible if a static decoder (e.g., a passive fabricated surface/layer) is employed. As for the illumination light, the energy consumption per wavelength channel can be estimated to be less than 0.8 mJ per image [82], which is negligible compared to other factors. If the generated images were to be digitized by an image sensor chip (e.g., a 5-10 mega-pixel CMOS imager), this would also add an extra energy consumption of $\sim 2\text{--}4$ mJ per image. Consequently, the overall energy consumption for generated images intended for human perception — excluding the need for a digital camera — is dominated by the SLM-based power in lower resolution image generation, whereas the digital encoder power consumption becomes the dominant factor for higher resolution image generation tasks such as the Van Gogh-style artworks. In contrast, GPU-based generative systems using a DDPM model have different energy characteristics that are dominated by the diffusion and successive denoising processes (involving e.g., 1000 steps). For example, the computational requirements for generating MNIST/Fashion MNIST and Van Gogh-style artwork images using a digital DDPM model amount to approximately 287.68 GFLOPs and 530.26 TFLOPs, respectively, corresponding to about 0.14–1.58 J per image for MNIST/Fashion-MNIST and 265–2916 J per image for Van Gogh-style artworks. Furthermore, if the generated images must be displayed on a monitor for human perception, additional energy consumption is incurred—typically between $\sim 13\text{--}500$ mJ per image at a 60 Hz refresh rate.

Overall, these comparisons reveal that if the novel image information to be generated will be stored and processed/harnessed in the digital domain, optical generative models would face additional power and speed penalties due to the digital-to-analog and analog-to-digital conversion steps that would be involved in the optical setup. However, if the image information to be generated will remain in the analog domain for direct visualization by human observers (e.g., in a near-eye or head-mounted display), the optical generative seeds can be pre-calculated with a modest energy consumption per seed, as detailed above. Furthermore, the static diffractive decoder surface can be fabricated using optical lithography or two-photon polymerization-based nano-fabrication methods, which would optically generate snapshot novel images within the display setup. This could enable compact and cost-effective image generators—such as “optical artists”—by replacing the back-end diffractive decoder with a fabricated passive surface. This setup would allow for the snapshot generation of countless novel images, including various forms of artwork, using simpler local optical hardware. From the perspective of a digital generative model, for comparison, one could also use a standard image display along with pre-computed and stored novel images created through, e.g., a digital DDPM model; this, however, requires significantly more energy consumption per image generation through the diffusion and successive denoising processes, as discussed earlier. Exploration of

optical generative architectures using nano-fabricated surfaces would enable various new applications, especially for image and near-eye display systems, including head-mounted and wearable setups.”

Additionally, although the main text does not state this explicitly, I suspect that the reported FID scores were measured only for the numerical simulations rather than the experimental results. This is suggested by Fig. 8, which shows a separate set of results labeled “experimental generation” without providing corresponding FID scores. If this is not the case, the authors should clarify this discrepancy, because the advantages of energy-efficiency and speed only hold true for the model implemented with real optics.

-- We have improved the performance of our models, as discussed earlier. With an optimized training strategy outlined in our revised Methods section that better utilized the phase encoding space, our optical generative models could create higher contrast images. For example, we were able to reduce the FID score for the MNIST dataset to <90, as shown in the **revised Fig. 3** (numerical results). Using our improved experimental set-up, the FID scores for the experimental image generation were calculated as 131.08 for the MNIST dataset and 180.57 for the Fashion MNIST dataset, as shown in the **revised Fig. 7**. While these experimental values are slightly higher than those obtained from numerical simulations, they still demonstrate the effectiveness of the optical generative model and validate its practical applicability.

Furthermore, we generated higher-resolution, Van Gogh–style artwork images to extend the demonstration of our model’s capabilities. Monochrome experimental results are presented in the newly added **Figs. 9 and 10 (as well as Supplementary Figs. S21-22)**, while **Fig. 11 and Supplementary Figs. S23-24** show experimental results for generating full-color images. These new experimental and numerical results provide a more comprehensive evaluation of our optical generative models.

These points have been included in the **Results section** of the revised manuscript:

*“...**Figure 7c** visualizes our experimental results for both models, which achieved experimental FID scores of 131.08 and 180.57 on the MNIST and Fashion-MNIST datasets, respectively.”*

*“...We further extended our experimental results for the optical generative models to create higher resolution images in the style of Van Gogh artworks, which were also demonstrated using the same setup shown in **Fig. 7b**. As illustrated in **Fig. 9**, we experimentally demonstrated snapshot monochrome image generation for Van Gogh-style novel artworks using a digital encoder paired with a jointly-trained diffractive decoder. The architecture of the digital encoder and the processing pipeline are shown in **Supplementary Fig. S17**. Additional comparisons in **Supplementary Fig. S18** reveal the diffractive decoder's superior performance over free-space-based image decoding, both using the same digital encoder architecture. Notably, while the free-space-based decoder completely failed in some cases, the diffractive decoder achieved stable image generation, with much better image quality at the output. As expected, we observed a numerical aperture-related minor degradation in image resolution when increasing the SLM-to-decoder distance to match our experimental conditions (see **Supplementary Fig. S18** vs. **Supplementary Fig. S19**); however, the diffractive decoder-based approach still maintains stable image generation performance compared to free-space-based decoding, which fails*

image generation in various cases despite using the same digital encoder, as shown in **Supplementary Fig. S19**.

By further increasing the number of digital encoder parameters (see **Supplementary Table S1**), we can improve the resolution and image quality of the optically generated Van Gogh-style novel artworks that are created in a snapshot; detailed comparisons are provided in **Supplementary Fig. S20** for the number of trainable parameters spanning 44M to 580M. **Figures 10 and 11** show our experimental results for higher-resolution monochrome and color (RGB) image generation using a digital encoder with 580M parameters. The monochrome images of Van Gogh-style novel artworks were generated with an illumination wavelength of 520 nm, while the color images used sequential wavelengths of {450,520,638} nm for B, G and R channels, respectively. In **Fig. 10**, the left three columns display the results where the snapshot images created by the optical generative model in a single pass closely resemble those produced by the digital diffusion model (i.e., the teacher model with 1.07 Billion trainable parameters and 1000 inference steps per image), demonstrating the consistency of our image generation process with respect to the teacher diffusion model. Conversely, the three right columns, highlighted within the orange box, showcase the optical model's ability to generate diverse images that differ from those of the teacher digital diffusion model, illustrating its creative variability at the output; also see additional experimental results for more Van Gogh-style novel artworks supporting our conclusions in **Supplementary Figs. S21-S22**.

For the multi-color Van Gogh-style novel artwork generation, phase-encoded generative seed patterns at each wavelength channel were generated and sequentially loaded onto the SLM. Under the illumination of corresponding wavelengths, multi-color images were generated through a fixed/static diffractive decoder and merged digitally; stated differently, the same decoder state was shared across all the illumination wavelengths for all the novel image generation. **Figure 11** presents the multi-color Van Gogh-style novel artwork generation results, including artistic examples that either match or differ from the outputs of the teacher digital diffusion model, which used 1.07 Billion trainable parameters and 1000 inference steps per image generation. Although slight chromatic aberrations were observed likely due to the achievable color space of the selected wavelengths and the camera's response function — the generated high-resolution color images maintained high quality. Additional experimental results of Van Gogh-style novel color artworks are provided in **Supplementary Figs. S23-S24**.

For the generation of colorful Van Gogh-style artworks, we also conducted performance comparisons for the optical generative model, reduced-size diffusion model (matching the size of our phase encoder), and the pretrained teacher diffusion model, as shown in **Supplementary Fig. S25**. Compared to our optical generative model, the reduced-size diffusion model that matches the size of our phase encoder produced inferior images with limited semantic details despite using 1000 inference steps. The optical generative model outputs closely match the teacher diffusion model (which also used 1.07B trainable parameters with 1000 inference steps).”

F. Suggested improvements: experiments, data for possible revision

Many suggestions for improvement have been provided in the earlier sections. The most critical issue is the last point in Section C—validating the capability of the proposed model to generate truly novel images by clearly separating its contribution from the pre-trained DDPM.

-- As detailed in our earlier responses, our new results in the revised manuscript both experimentally and numerically demonstrate the distinctiveness of the outputs produced by the optical generative model compared to those of the digital diffusion model (teacher). As shown in the orange boxes of the newly added Figs. 10-11 as well as Supplementary Fig. S21, the generation results of the optical model exhibit high quality and valid semantic information while differing meaningfully from the teacher diffusion model's outputs. These experimental and numerical results demonstrate that our framework achieves true image generation beyond a mere imitation of the teacher model.

One additional point to consider: In Fig. 4, the noticeably low contrast in the results of the optical generative models compared to fully digital models may worth further exploration. Is this contrast limitation an inherent drawback of the system caused by the zero-order effect in diffractive optics? If so, it would be valuable to explicitly address this in the text and discuss any potential strategies to mitigate this issue.

-- Indeed, the zero-order diffraction from the SLM disrupts the generated images and leads to contrast degradation, which can be mitigated by incorporating a 4F filtering system. In this revision, we have implemented a novel approach that predicts a scaling factor simultaneously with the electronic encoder's output of the optical random seeds, as detailed in the revised Methods section. Moreover, a larger phase bit depth for the diffractive decoder and the increased encoding phase range of the input SLM greatly improved the visual quality of the optically generated images in our revised manuscript. This enhancement, as illustrated in **Figs. 3 and 7**, significantly improves the contrast of the images generated through the diffractive decoder.

We also report detailed comparisons to show the improvement of the optimized optical generative set-up, as quoted from our revised manuscript:

*“... we also experimentally evaluated the snapshot optical image generation under a limited phase encoding space (e.g., $0-\pi/2$ vs. $0-2\pi$) and a limited decoder bit depth (e.g., 4 bit-depth vs. 8 bit-depth) using a restricted optical setup; see **Supplementary Figs. S14a, S14b and S15**. The performance comparisons between the experimental results of **Fig. 7 and Supplementary Fig. S14** show the significance of employing a large phase bit depth for the diffractive decoder as well as increasing the encoding phase range of the input SLM.”*

**G. References: appropriate credit to previous work?*

The references seem appropriate, and no significant omissions were noted.

-- Thanks.

H. Clarity and context: lucidity of abstract/summary, appropriateness of abstract, introduction and conclusions

The abstract, introduction, and conclusions are generally clear and appropriately contextualized.

-- Thanks.

Referee #3 (Remarks on code availability):

The code simply requires the PyTorch framework and is reproducible. It provides a sample that utilizes the trained parameters to simulate the behavior of the proposed optical generative model. It does not include the implementation for training the DDPM from dataset and use DDPM to train the optical generative model. It remains unclear that how many noise/image pairs are required and what the computational overhead is for training this model. The authors should either provide the corresponding code or elaborate the training parameters in the manuscript.

-- Following the referee's suggestion, we have added more details about the training process for the optical generative models in the **Methods Section** and the **Supplementary Information Sections 1 and 2**. The above points were included in the **Method Section** of the revised manuscript:

"...For the Van Gogh-style artwork generation, three class labels were used as the conditions for image generation: {"architecture", "plants", and "person"}. The input resolution (x, y) of latent noise was set to $(80, 80)$ and the size of the class label embedding l was 80. We added perturbations to the input noise so that the optical generative model was easier to cover the whole latent space [66]. The numerical simulations of monochrome and multi-color artwork generation shared the same physical distance and working wavelength as the lower-resolution optical image generation. The object plane size was $8\text{mm} \times 8\text{mm}$ with a resolution of 1000×1000 . The number of optimizable features on the decoding layer was 1000×1000 . On the image plane, the size and the resolution were $5.12\text{mm} \times 5.12\text{mm}$ and 640×640 , respectively. For the teacher DDPM used to generate Van Gogh-style artworks, we finetuned the pre-trained Stable Diffusion v1.5 [66] with the vangogh2photo dataset [64] and captioned it by a GIT-base model [65]. The total timestep T was set to 1000 steps and β_t was a linear function from 0.00085 ($t = 1$) to 0.012 ($t = T$). The models were trained and tested using PyTorch 2.21 with four NVIDIA RTX 4090 GPUs."

"...The extrinsic parameters of the digital encoder and the optical generative model were similar to the snapshot optical generative models except for the number (L_o) of the decoding layers, which was set to 5. The distance between decoding layers d_{l_o-1, l_o} was 20mm. In the training of the iterative optical generative models, the total timestep T was set to 1000. β_t is a linear function from $1e-3/1e-3$ ($t = 1$, Butterflies/Celeb-A) to $5e-3/0.01$ ($t = T$, Butterflies/Celeb-A)."

*"...For the optical image generation experiments corresponding to higher resolution monochrome Van Gogh-style artworks shown in **Figs. 9-10**, the same setup as in the previous experiments, was used, with adjustments made only to the resolution. The resolutions of the encoded phase pattern, the decoding layer, and the sensor plane were 1000×1000 , 1000×1000 , and 640×640 , respectively. For the multi-color artwork generation in, e.g., **Fig. 11**, the same setup was employed, with illumination wavelengths set to $\{450, 520, 638\}$ nm,*

applied sequentially. All the captured images were first divided by the bit depth of the sensor and normalized to [0, 1], and then we applied gamma correction ($\gamma = 0.454$) [83] to adapt to human vision.”

Also see **Supplementary Information Sections 1 and 2** for additional information that we included in our revised manuscript.

We sincerely thank the referees for their reviews and the constructive feedback that we have received on our manuscript “**Optical Generative Models**” submitted to *Nature* (Manuscript ID: **2024-09-20762A**).

As detailed below, in this 2nd round of reviews, we have revised our manuscript in response to the referees’ comments. The original referee comments are shown in black color, whereas for ease of communication, our answers are provided in blue. Our revisions have also been marked in the main text and supplementary information files using yellow highlighting.

Referee #1:

I appreciate the authors' continued efforts on this work, including the addition of more experimental results and comprehensive discussions.

-- We sincerely thank the referee for the positive assessment and constructive, valuable feedback, which helped us to enhance the quality and clarity of our manuscript.

However, several concerns remain: Authors claimed scalability of this optical system. However, the parameter scale considered in this paper is only in the millions, which is much smaller than the commonly used generative AI model. How may it scaled up?

-- We thank the referee for these valuable points. In our manuscript, we showed that by increasing the number of diffractive features in a single-layer diffractive decoder, optical generative models can achieve the generation of more complex and higher-resolution image content, such as novel artworks. From 400x400 diffractive features for Celeb-A generation to 1000x1000 for artwork generation, the optical generative model is able to scale up to high-resolution image generation tasks while the parameter of the digital encoder scales up from 27M to 44M. The results of the Celeb-A optical generation with 27M digital encoder and 400x400 diffractive features per decoding layer are shown in **Extended Data Figure 3**. The results of the Van Gogh-style artwork optical generation with 44M digital encoder and 1000x1000 diffractive features per decoding layer are shown in **Extended Data Figure 7**.

Beyond increasing the number of diffractive features in a single layer, we have further scaled the optical system to **multi-layer diffractive decoders**. For example, iterative optical generative models presented in **Supplementary Fig. S24** indicate that extending the decoding layers along the axial direction can further enhance the generation capabilities of optical generative models, quoted below:

“...Supplementary Fig. S24 further demonstrates the scalability of the diffractive decoder: as the number of decoding layers increases, the FID score on the Celeb-A dataset drops, indicating the enhanced generative capability of the iterative optical generative model.”

“Fig. S24: Performance evaluation of iterative optical generative models with different depths of the diffractive decoder → shown on the next page:

Fig. S24: Image generation ability of an iterative optical generative model with a multi-layer diffractive decoder ($L_o = 2, 5, 10$ layers), which shows the scalability of the diffractive decoder architecture.”

Therefore, increasing the complexity of the diffractive decoder layers both laterally and axially can further scale the image-creation ability of optical generative models. Another example of this scalability is demonstrated in the **Results** section, as part of **Supplementary Fig. S25**, where deeper diffractive decoders show resilience against random misalignments that may otherwise decrease the performance of the novel image generation.

Next, we consider the **scalability in diffractive decoder manufacturing**: our optical generative models can be adapted to very limited diffractive decoder bit depths (e.g., just 3 discrete phase levels per feature) for constrained manufacturing scenarios (as shown in **Supplementary Fig. S15**). This reduced bit depth per diffractive layer enables the use of mass manufacturing techniques, such as optical lithography-based nano-fabrication techniques and nanoimprint (or stamping) lithography, greatly facilitating the large-scale and cost-effective fabrication of optical generative model decoders. These points have been emphasized in the **Discussion** section of our manuscript:

“...in the optical setup, one can integrate these limitations directly into the training process of the optical generative model, which will accordingly shape both the digital encoder and the diffractive decoder, making the *in silico*-optimized system align better with the physical limits and the capabilities of the local hardware of the optical generative model. As demonstrated in **Supplementary Fig. S15**, this strategy led to notable performance improvements compared to models that did not account for such bit-depth limitations during training. **A key insight of this analysis is that a relatively simpler decoder surface with just 3 discrete phase levels (covering only $0, 2\pi/3$ or $4\pi/3$ per feature)** would be sufficient in our optical generative models, opening the door to replacing the decoder architecture with a passive, thin surface fabricated by

e.g., two-photon polymerization or optical lithography based nano-fabrication techniques. This would further simplify the physical setup of the local optical generative model, also making its hardware more compact, lightweight and cost-effective.”

“...Furthermore, in **Supplementary Fig. S15**, we explored the impact of limited phase modulation levels (i.e., a limited phase bit-depth per feature) at the optical generative seed plane and the diffractive decoder. These comparisons revealed that novel image generation results could be improved by including the modulation bit depth limitation (due to, e.g., inexpensive SLM hardware or surface fabrication limitations) in the forward model of the training process. **Such a training strategy using a limited phase bit-depth revealed that the fixed/static decoder surface could work with 4 phase bit-depth and even 3 discrete levels of phase (e.g., 0, $2\pi/3$, $4\pi/3$) per feature to successfully generate novel images through its decoder phase function; see Supplementary Fig. S15.** This is important and highly desired since most two-photon polymerization or optical lithography-based fabrication methods can routinely fabricate surfaces with 2-16 discrete phase levels per feature, which could help replace the decoder SLM with a passive fabricated surface structure.”

“Fig. S15: The impact of limited phase modulation levels on snapshot optical generative models.

Fig. S15: $l_{train}^\phi, l_{train}^D, l_{test}^\phi, l_{test}^D$ represent the discrete phase modulation levels of the SLM and the diffractive decoder during the training (train) and testing (test), respectively. The blind

testing performance can be improved significantly by including the modulation bit depth limitation of the snapshot optical generative model hardware during the training process. $M(l_{train}^{\phi}, l_{train}^D, l_{test}^{\phi}, l_{test}^D)$ refers to the snapshot optical generation model trained and tested under $l_{train}^{\phi}, l_{train}^D, l_{test}^{\phi}, l_{test}^D$.”

Additionally, as shown in **Extended Data Fig. 10**, we have further demonstrated the scalability of optical generative models through **spectral multiplexing**. In the scheme shown in **Extended Data Fig. 10a**, a single encoded phase pattern generated by a random seed is illuminated at different wavelengths, and the corresponding diffractive decoder can accurately reconstruct and reveal the intended information within its designated wavelength channel. **By increasing the number of trainable diffractive features in a given decoder architecture proportional to the number of illumination wavelengths, this spectral multiplexing capability can be scaled to include tens of wavelengths, where each unique decoder can only have access to one channel of information from the same/common encoder output.** Along with the newly added **Extended Data Fig. 10**, we have included these points in the **Multiplexed optical generative models** sub-section of the **Methods** section:

*“...We demonstrate in **Extended Data Fig. 10** the potential of the optical generative model as a privacy-preserving and multiplexed visual information generation platform. In the scheme shown in **Extended Data Fig. 10a**, a single encoded phase pattern generated by a random seed is illuminated at different wavelengths, and only the correctly paired diffractive decoder can accurately reconstruct and reveal the intended information within the corresponding wavelength channel. **This establishes secure content generation and simultaneous transmission of visual information to a group of viewers in a multiplexed manner, where the information presented by the digital encoder remains inaccessible to others unless the correct physical decoder is used (shown in **Extended Data Fig. 10b**). This is quite different from a free-space-based image decoding, which fails to multiplex different information channels using the same encoded pattern due to strong cross-talk among different channels of information, as shown in **Extended Data Fig. 10c**. By increasing the number of trainable diffractive features in a given decoder architecture proportional to the number of multiplexing wavelengths, this privacy-preserved multiplexing capability can be scaled to include many wavelengths, where each unique decoder can only have access to one channel of information from the same/common encoder output. This secure multiplexing capability through diffractive decoders does not need dispersion engineering of the decoder material, and can be further improved by including polarization diversity in the diffractive decoder system. Without spatially-optimized diffractive decoders, which act as physical security keys, a simple wavelength and/or polarization multiplexing scheme through free-space diffraction or a display would not provide real protection or privacy, as everyone would have wide access to the generated image content at a given wavelength and/or polarization combination.***

*In this context, the physical decoder architecture that is jointly trained with the digital encoder offers a naturally secure information processing pipeline for encryption and privacy preservation. **Since the encoder and the decoders are designed in tandem, and the individual decoders can be fabricated using various nano-fabrication methods, it is extremely difficult to reverse engineer or replicate the physical decoders unless access to the design files is available.** This physical protection and private multiplexing capability enabled by different physical decoders that receive signals from the same digital encoder is inherently difficult for conventional image display*

technologies to perform since they render content perceptible to any observer. For various applications such as secure visual communication to a group of users (e.g., in public), anti-counterfeiting, and personalized access control (e.g., dynamically adapting to the specific attributes or history of each user), private and multiplexed delivery of generated visual content would be highly desired. Although not demonstrated here, such a secure multiplexed optical generative model can also be designed to work with spatially partially-coherent light by appropriately including the desired spatial coherence diameter in the optical forward model, which would open up the presented framework to, e.g., light emitting diodes.”

“Extended Data Figure 10: Optical generative models for private and multiplexed information

generation. a, Demonstration of the privacy-preserving and multiplexed information/image projection using an optical generative model. A single encoded phase pattern generated from a random seed is illuminated with different wavelengths, and only the matching diffractive decoder can correctly reconstruct and reveal the intended information; with three unique diffractive decoders, each viewer can see a different optically generated image (i.e., Van Gogh style paintings of architecture, plants, person) from the same encoded phase pattern. **b**, Confusion matrices showing the decoded images as perceived by each viewer (V1, V2 or V3); a separate decoder (D1, D2 or D3) is given to each viewer. Rows represent different illumination wavelengths, while columns correspond to different diffractive decoders, D1, D2, D3. **c**, Free-space-based decoding (without a diffractive decoder) fails in multiplexed image generation. Furthermore, it does not enable protection or privacy of the displayed information since anyone can observe the resulting images at different wavelengths without the need for a physical decoder.”

As summarized above, from the perspective of the complexity of the diffractive decoders, the misalignment robustness, the manufacturing feasibility, and the wavelength multiplexing capabilities, we can scale optical generative models across multiple dimensions to accommodate increasingly complex novel image generation tasks.

In particular, authors have supplemented more experimental results, which, however, are apparently worse than the numerical results, e.g., for Van Gogh-style artworks. These findings suggest that inherent analog noise in the optical system may limit its scalability. The authors should address this issue explicitly.

Regarding the experimental results, the authors should also provide accuracy loss metrics, as done in the numerical section, to allow for better comparison between numerical, digital, and optical outcomes.

-- We thank the reviewer for this constructive feedback. In our setup, one of the sources of analog noise arises from speckle noise due to the high coherence of the laser source. To address this issue, various speckle reduction approaches¹⁻⁵ can be used. One of the simpler approaches to reduce speckle noise could be to use partially coherent light, such as an LED (light emitting diode) with an adjustable pinhole; by appropriately selecting the bandwidth of the LED and the diameter of the pinhole (as well as the axial propagation distance), we can precisely engineer both the spatial and temporal coherence of the illumination. This point has been included in the **Methods** section:

“...Although not demonstrated here, such a secure multiplexed optical generative model can also be designed to work with spatially partially-coherent light by appropriately including the desired spatial coherence diameter in the optical forward model, which would open the presented framework to, e.g., light emitting diodes.”

Furthermore, we quantified the PSNR (Peak Signal-to-Noise Ratio) image quality metric calculated between our numerical simulations and experimental results, which is reported in **Supplementary Fig. S13**. Our optical generative model experimental results achieved PSNR values of 17.70 dB and 17.28 dB on the snapshot artwork generation and multicolor artwork generation tasks, respectively. These PSNR values indicate a high similarity between the two sets of images (experimental vs. numerical), which was also added to our revised manuscript as quoted below:

*“...To quantify the fidelity of the experimental optical generative model, **Supplementary Fig. S13***

reports the peak signal-to-noise ratio (PSNR) values between the numerically simulated and the experimentally generated results. These quantitative comparisons for both the snapshot monochrome and the multicolor optical generative models demonstrate that the experimental outputs closely match their respective simulations.”

“Fig. S13: Fidelity comparisons between the numerical and experimental results of the optical generative model.

Fig. S13: We report the PSNR values between the numerically simulated and experimentally generated results. For the snapshot optical generative model, the results correspond to **Fig. 4, Supplementary Fig. S9, and Supplementary Fig. S10**. For the multicolor optical generative model, the evaluations correspond to the results from **Fig. 5, Supplementary Fig. S11, and Supplementary Fig. S12**.”

In addition to these analyses, we also showed the CLIP (contrastive language-image pretraining score) values of the numerical and experimental results of the Optical Generative Model, compared against the teacher digital diffusion model, indicating the high semantic alignment of our experimental results. These comparisons are mentioned in the **Results** section of the revised manuscript:

“Additionally, Supplementary Fig. S14 presents the CLIP score evaluations corresponding to the results shown in Figs. 4 - 5, highlighting the strong semantic consistency achieved by the optical generative model.”

“Fig. S14: CLIP score evaluation of the text-to-image alignment for Van Gogh style artwork generation.

Fig. S14: We present the contrastive language-image pre-training score (CLIP score) for both the

numerical and experimental results of the Optical Generative Model, compared against the teacher digital diffusion model (with 1.07 Billion trainable parameters and 1000 steps used for each image inference). The CLIP score quantifies the semantic alignment between the generated images and the reference text: “Van Gogh style painting of {architecture, plants, person}”. The CLIP score evaluation for the snapshot optical generative model (left) corresponds to **Fig. 4, Supplementary Fig. S9, and Supplementary Fig. S10**. The evaluation for the multicolor optical generative model (right) corresponds to **Fig. 5, Supplementary Fig. S11, and Supplementary Fig. S12.**”

Related to these comparisons, we should also note that there are only ~800 authenticated Van Gogh paintings available; therefore, the sample set is relatively small to capture the full data distribution. As a result, computing FID against this limited set provides unstable measurements, which we avoided.

Regarding the cost for diffractive decoder, I meant the training cost. This aspect has not been discussed. This generative optics system is only used for inference, but what about the training process? It seems to be hidden, like a lot of accelerator work, but I think the training cost should be explicitly presented, especially when the system is to be scaled up.

-- We thank the reviewer for raising this important point regarding the training efficiency of our optical generative models. These details are provided in the **Supplementary Sec. 3** of our revised manuscript:

“...The training was conducted on a server configured with an AMD Threadripper 3990X CPU, 128 GB of G.Skill DDR4 RAM (16 GB × 8), and four NVIDIA GeForce RTX 4090 GPUs. For training the teacher denoising diffusion probabilistic model (DDPM), we used a batch size of 200 and trained it for 300 epochs. The training time, which is a one-time effort, in GPU hours for each dataset is as follows: MNIST – 33 hours, Fashion-MNIST – 52 hours, Butterfly-100 – 48 hours, and CelebA – 58 hours. For fine-tuning the stable diffusion for artwork generation, we used a batch size of 8 and trained for 20 epochs, which took ~61 GPU hours. With more advanced GPUs, such as the NVIDIA A100, the training speed would be 3–8 times faster. This implies, using a state-of-the-art GPU, a reduction in training time for the teacher DDPM to approximately 6, 10, 9, and 11 GPU hours for MNIST, Fashion-MNIST, Butterfly-100, and CelebA, respectively.

For training the optical generative model, we used a batch size of 100 and trained for 100 epochs. In each iteration, we sampled a batch of random noise and fed it into the teacher model to obtain the data pair for training the optical generative model. We used 50 equally spaced steps of sampling to obtain the target distribution without significantly affecting the quality of the image generation. The training time of the optical generative model in GPU hours for each dataset is as follows: MNIST – 38 hours, Fashion-MNIST – 58 hours, Butterfly-100 – 59 hours, and CelebA – 70 hours. For training the optical generative model in artwork generation, we used a batch size of 4 and trained for 100 epochs (with 2500 iterations each epoch), which took approximately 82 GPU hours. Because the model we finetuned from stable diffusion shares the same latent representation for monochrome and colorful image generation, the training time consumption is similar for these two tasks. Once again, with higher-end GPUs, such as the NVIDIA A100, the training speed would be 3–8 times faster. This implies a reduction in training time for the teacher DDPM to approximately 7, 11, 11, and 13 GPU hours for MNIST, Fashion-MNIST, Butterfly-100, and CelebA, respectively.

While training a deep learning model can be computationally intensive and time-consuming, it is generally a one-time process that occurs during the model development phase. Once the model is

trained, it is fixed and reused across a wide range of deployment scenarios. In contrast, inference must be performed repeatedly—often in real-time or at large scale—during practical deployment. As a result, inference efficiency becomes much more critical for real-world utility, especially in applications requiring rapid response, low power consumption, or deployment on resource-constrained devices.”

Referee #2:

I would like to thank the authors for the extensive clarifications that are very transparent. I am finally leaning now towards recommending publication. I think the technological relevance is given for some particular potential applications, not in the context of general image generation.

However, the experiment illustrates in a beautiful manner what unconventional systems and optics in particular are capable of implementing. As such I expect the work to draw significant attention from a broad community.

-- We sincerely thank the referee for the constructive and comprehensive feedback in the first round of the revision, which helped us significantly improve the quality of the manuscript.

Referee #3:

Thanks to the authors for carefully addressing the previous comments. The paper has improved significantly over the earlier version, demonstrating the proposed optical neural network as a properly and successfully distilled variant of a diffusion model for image generation tasks. I appreciate the high-resolution Van Gogh image generation demonstration, and the additional details on training and evaluation make the work more robust. It is encouraging to see that the proposed model can indeed generate novel images that are meaningfully different from those produced by the teacher model, and the empirical validation of the phase encoding strategy adds strength to the contribution. I now find the method itself valid;

-- We sincerely thank the reviewer for the encouraging feedback and for recognizing the improvements made in our revised manuscript.

However, I remain somewhat unconvinced about its practical advantages or necessity. Fundamentally, even if we acknowledge that the distilled network structure is efficient, one must ask: if the ultimate goal is to digitally generate novel images, why not simply use the numerical version without implementing it physically with a laser and SLM? The phase encoding can be simulated digitally and integrated into the network structure. The FLOPs would be nearly identical (given that the digital encoder has ~100M params and the diffractive layer only has ~1M), and without relying on the physical SLM, the inference frame rate could actually be higher. In that case, the energy consumption would also remain comparable.

Viewed this way, the core value of the paper becomes the proposal of a network architecture with a physical interpretation to approximate or accelerate diffusion-based generative models. There is

already a substantial body of literature on accelerating diffusion models, and I believe it is necessary to acknowledge these efforts and, ideally, compare the proposed approach with them—particularly if the framing is focused on inference speed or energy efficiency.

Overall, the concept is interesting and the current demonstration is valid. However, I remain concerned that its practical value is limited, as it lacks compelling advantages over existing generative models. The need for a physical optical layer is not fully justified, especially given that the use of diffractive layers in optical neural networks is already a well-established technique. Without a clearer argument for why physical implementation is necessary or superior, adapting this structure to a new application may fall short of the novelty and impact expected for a top-tier journal.

-- We sincerely thank the reviewer for recognizing the validity of our demonstrations and also raising these important points listed above.

First, we would like to clarify some related points: when phase encoding and angular spectrum-based wave propagation are incorporated into a digital pipeline, the optical operations indeed account for only a small fraction of the total parameter count. However, their digital implementation incurs a relatively large number of FLOPs due to the computational demand of Fast Fourier Transforms (FFTs) involved in the digital implementation of wave optics. For a 2D spatial sampling grid of size (H, W) and a wave propagation distance of z , the digital implementations of the angular spectrum-based monochromatic wave propagation require a 2D FFT computation cost of $5N \log_2(N)$ FLOPs, where $N = H \times W$. The element-wise multiplication with the transfer function in the complex domain introduces $6N$ operations, followed by an inverse 2D FFT that adds another $5N \log_2(N)$ FLOPs. Consequently, the total number of FLOPs for the digital implementation of one-layer diffractive decoding at a single wavelength, as described in Eq. 5, amounts to $20N \log_2(N) + 18N$, accounting for two free-space propagation steps and one layer of modulation. These numbers would further scale up for multicolor image generation tasks as we would need to digitally perform wave processing at each wavelength channel separately. For example, consider the monochrome optical generative model for MNIST, which comprises a three-layer fully connected (FC) digital encoder and a one-layer diffractive decoder with an optimizable feature map of size $H \times W = 400 \times 400$. In this example, the digital replacement of the optical diffractive decoder (which only contains 0.16 million parameters) will need 58.24 million digital FLOPs (whereas the digital phase encoder only uses ~ 6.3 million FLOPs).

Furthermore, our optical generative model performs inference in a **single step**, achieving comparable performance with respect to the pre-trained teacher diffusion model that requires 1000 iterative inference steps, as further quantified in **Extended Data Fig. 8**. While prior works have accelerated diffusion model inference, as also emphasized by the referee, single-step inference is still desired. In our implementations, the snapshot optical generative model could generate high-quality novel artworks without losing semantic information, as quantitatively compared in **Extended Data Fig. 8**:

*“...For the generation of colorful Van Gogh-style artworks, we also conducted performance comparisons for the optical generative model, reduced-size diffusion model (matching the size of our phase encoder), and the pretrained teacher diffusion model, as shown in **Extended Data Fig. 8**. Compared to our optical generative model, the reduced-size diffusion model that matches the size of our phase encoder produced inferior images with limited semantic details despite using 1000 inference steps. The optical generative model outputs closely match the teacher diffusion model (which also used 1.07B trainable parameters with 1000 inference steps). Furthermore, the CLIP score evaluations suggest that the optically generated images exhibit strong alignment with the underlying semantic content.”* **The average CLIP scores of the**

multicolor optical generative model, the reduced-size diffusion model, and the pre-trained teacher diffusion model are 28.25, 24.45, and 28.72, respectively (the reference text: “Van Gogh style painting of {architecture, plants, person}”)

Extended Data Figure 8: Generation of colorful Van Gogh-style artworks. Numerical simulation results of the multicolor optical generative model (580M parameters) compared against a reduced-size diffusion model (shown in the middle) trained from scratch, whose U-Net has the same number of learnable parameters as the digital encoder in the optical generative model (580M parameters), and a pre-trained teacher diffusion model (with 1.07B parameters) which is shown on the right. **The average CLIP scores of the multicolor optical generative model, the reduced-size diffusion model, and the pre-trained teacher diffusion model are 28.25, 24.45, and 28.72, respectively (the reference text: “Van Gogh style painting of {architecture, plants, person}”).**

Following the referee’s suggestion, we have also included in our revised manuscript additional energy efficiency comparisons with some of the mainstream diffusion acceleration methods, added into the **‘Energy consumption and speed of optical generative models’** sub-section of the **Methods**, along with some additional references cited:

“...We also note that various prior works focused on accelerating diffusion models to improve their inference speed and energy efficiency. For example, the Denoising Diffusion Implicit Model (DDIM) enabled content generation up to 20 times faster than DDPM while maintaining comparable image quality^{Error! Reference source not found.-Error! Reference source not found.}. Under such an accelerated configuration, the estimated computational energy required for generating images using a digital DDIM would be ~7–79 mJ per image for MNIST/Fashion-MNIST and 13.25–145.8 J per image for Van Gogh-style artworks. Furthermore, if the generated images must be displayed on a monitor for human perception, additional energy consumption is incurred—typically between ~13–500 mJ per image at a 60 Hz refresh rate.”

The same sub-section in the **Methods** also discusses the energy consumption of our optical generative models under different settings and image generation tasks:

*“The electronic encoder used for MNIST and Fashion MNIST datasets consists of three fully connected layers, and it requires 6.29 MFLOPs per image, with an energy cost of ~0.5-5.5 pJ/FLOP, resulting in an energy consumption of 0.003-0.033 mJ per image. This energy consumption increases to ~1.13-12.44 J and ~0.28-3.08 J per image for the Van Gogh-style artworks reported in **Figs. 4-5** and **Extended Data Fig. 6**, respectively. The input SLM, with a power range of 1.9–3.5 W, consumes ~30–58 mJ per image at a 60 Hz refresh rate. This SLM-related energy consumption can be reduced to <2.5 mJ per image using a state-of-the-art SLM. The diffractive decoder has a similar energy consumption if a second SLM is used; however, its contribution would become negligible if a static decoder (e.g., a passive fabricated surface/layer) is employed. As for the illumination light, the energy consumption per wavelength channel can be estimated to be less than 0.8 mJ per image, which is negligible compared to other factors. If the generated images were to be digitized by an image sensor chip (e.g., a 5-10 mega-pixel CMOS imager), this would also add an extra energy consumption of ~2–4 mJ per image.”*

Overall, these comparisons reveal that **if the image information to be generated will remain in the analog domain for direct visualization by human observers (e.g., in a near-eye or head-mounted display), the optical generative seeds can be pre-calculated with a modest energy consumption per seed, as detailed above. Furthermore, the static diffractive decoder surface can be fabricated using optical lithography or two-photon polymerization-based nano-fabrication methods, which would optically generate snapshot novel images within the display setup. This could enable compact and cost-effective image generators by replacing the back-end diffractive decoder with a fabricated passive surface. This setup would allow for the snapshot generation of countless novel images, including various forms of artwork, using simpler local optical hardware.** From the perspective of a digital generative model, for comparison, one could also use a standard image display along with pre-computed and stored novel images created through, e.g., a digital DDPM model; this, however, requires significantly more energy consumption per image generation through the diffusion and successive denoising processes, as discussed earlier. **Optical generative architectures using nano-fabricated surfaces would enable various new applications, especially for image and near-eye display systems, including head-mounted and wearable setups.**

Therefore, we believe that the optical diffractive decoders provide various advantages, enabling exciting applications of the presented optical generative models using diffractive (physical) decoders, as emphasized in our revised manuscript.

In addition to these, in our revision, we have also demonstrated another important capability of optical generative models as a privacy-preserving and multiplexed information generation platform (see **Extended Data Fig. 10**). A single encoded phase pattern can be illuminated by various visible wavelengths, with only the correctly matched diffractive decoder able to accurately reconstruct the intended information within the corresponding wavelength channel. This multiplexed novel image generation mechanism enables secure content generation and transmission, where access to the information is contingent on the presence and correct pairing of both the digital encoder and the physical decoder. **In this context, the physical decoder architecture that is jointly trained with the**

digital encoder also offers a naturally secure information processing pipeline for encryption and privacy preservation. Since the encoder and the decoders are designed in tandem, and the individual decoders can be fabricated using various nano-fabrication methods, it is extremely difficult to reverse engineer or replicate the physical decoders unless access to the design files is available. This physical protection and private multiplexing capability enabled by different physical decoders that receive signals from the same digital encoder is inherently difficult for conventional image display technologies to perform since they render content perceptible to any observer. **For various applications such as secure visual communication to a group of users (e.g., in public), anti-counterfeiting, and personalized access control (e.g., dynamically adapting to the specific attributes or history of each user), private and multiplexed delivery of generated visual content would be highly desired.** Such a secure multiplexed optical generative model can also be designed to work with spatially partially-coherent light by appropriately including the desired spatial coherence diameter in the optical forward model, which would open up the presented framework to, e.g., light emitting diodes (LEDs).

These points have been included in the **Multiplexed optical generative models** of the **Methods section**, along with the newly added **Extended Data Fig. 10**:

*“...We demonstrate in **Extended Data Fig. 10** the potential of the optical generative model as a privacy-preserving and multiplexed visual information generation platform. In the scheme shown in **Extended Data Fig. 10a**, a single encoded phase pattern generated by a random seed is illuminated at different wavelengths, and only the correctly paired diffractive decoder can accurately reconstruct and reveal the intended information within the corresponding wavelength channel. This establishes secure content generation and simultaneous transmission of visual information to a group of viewers in a multiplexed manner, where the information presented by the digital encoder remains inaccessible to others unless the correct physical decoder is used (shown in **Extended Data Fig. 10b**). This is quite different from a free-space-based image decoding, which fails to multiplex different information channels using the same encoded pattern due to strong cross-talk among different channels of information, as shown in **Extended Data Fig. 10c**. By increasing the number of trainable diffractive features in a given decoder architecture proportional to the number of multiplexing wavelengths, this privacy-preserved multiplexing capability can be scaled to include many wavelengths, where each unique decoder can only have access to one channel of information from the same/common encoder output. This secure multiplexing capability through diffractive decoders does not need dispersion engineering of the decoder material, and can be further improved by including polarization diversity in the diffractive decoder system. Without spatially-optimized diffractive decoders, which act as physical security keys, a simple wavelength and/or polarization multiplexing scheme through free-space diffraction or a display would not provide real protection or privacy, as everyone would have wide access to the generated image content at a given wavelength and/or polarization combination.*

In this context, the physical decoder architecture that is jointly trained with the digital encoder offers a naturally secure information processing pipeline for encryption and privacy preservation. Since the encoder and the decoders are designed in tandem, and the individual decoders can be fabricated using various nano-fabrication methods, it is extremely difficult to reverse engineer or replicate the physical decoders unless access to the design files is available. This physical protection and private multiplexing capability enabled by different physical decoders that receive signals from the same digital encoder is inherently difficult for conventional image display technologies to perform since they render

content perceptible to any observer. For various applications such as secure visual communication to a group of users (e.g., in public), anti-counterfeiting, and personalized access control (e.g., dynamically adapting to the specific attributes or history of each user), private and multiplexed delivery of generated visual content would be highly desired. Although not demonstrated here, such a secure multiplexed optical generative model can also be designed to work with spatially partially-coherent light by appropriately including the desired spatial coherence diameter in the optical forward model, which would open up the presented framework to, e.g., light emitting diodes.”

“Extended Data Figure 10: Optical generative models for private and multiplexed information

generation. a, Demonstration of the privacy-preserving and multiplexed information/image projection using an optical generative model. A single encoded phase pattern generated from a random seed is illuminated with different wavelengths, and only the matching diffractive decoder can correctly reconstruct and reveal the intended information; with three unique diffractive decoders, each viewer can see a different optically generated image (i.e., Van Gogh style paintings of architecture, plants, person) from the same encoded phase pattern. b, Confusion matrices showing the decoded images as perceived by each viewer (V1, V2 or V3); a separate decoder (D1, D2 or D3) is given to each viewer. Rows represent different illumination wavelengths, while columns correspond to different diffractive decoders, D1, D2, D3. c, Free-space-based decoding (without a diffractive decoder) fails in multiplexed image generation. Furthermore, it does not enable protection or privacy of the displayed information since anyone can observe the resulting images at different wavelengths without the need for a physical decoder.”

Furthermore, some of the comparisons raise logical concerns. Specifically, in Figs. S18–S19, the authors state: “Models without an optical decoder (i.e., utilize free-space propagation only) fail to generate novel images despite the presence of the digital diffusion model-based teacher and the same digital encoder, which demonstrates that the diffractive decoder surface plays a vital role in improving the visual quality of the snapshot generated novel optical images.”

First, in Fig. S18, I wonder why Design II produces a completely gray output in the first row, yet manages to generate some meaningful structure in the second row? The comparison is also not exactly fair, because the two models do not have equivalent capacity. While both designs use the same digital encoder architecture, Design I includes an optimizable diffractive decoder layer, whereas Design II lacks additional trainable component.

-- We thank the reviewer for this insightful observation. First, **Supplementary Fig. S18** is now renumbered as **Supplementary Fig. S7** due to the shortening of the main text to comply with editorial requests. As for the comparison between Design I and Design II in this supplementary figure, the purpose is to highlight the importance of the physical diffractive decoder. While both designs share the same digital encoder, Design I benefits from an additional trainable optical decoding component, which significantly enhances its generative capacity. Design II lacks this element, and **the resulting performance gap underlines the diffractive decoder's critical role in enabling high-fidelity image generation.**

In this supplementary figure, some examples from Design II show partial structures, while others completely fail to generate meaningful outputs, as shown in different rows of the figure. We empirically considered the generated images with a CLIP score below 15 as a failure. Under this criterion, the failure rate of free-space-based image decoding (without the physical decoder) was quantified as 18.8%. Even among the successful cases (the remaining ~81.2%), the image generation quality using free-space decoding is inferior to that achieved with the jointly optimized diffractive decoder; indeed, *this is physically expected since the diffractive decoder has additional trainable degrees of freedom compared to the free-space-based image decoding.* To better clarify these points, we have revised the captions of the corresponding supplementary figure (now **Supplementary Fig. S7**) as well as the related text in the **Results** section, as quoted below:

“...Additional comparisons in **Supplementary Fig. S7** reveal the diffractive decoder's superior performance over free-space-based image decoding, both using the same digital encoder architecture. Notably, while the free-space-based decoder completely failed in some cases, achieving a contrastive

language-image pretraining score (CLIP score) below 10-15, the diffractive decoder achieved stable image generation, with much better image quality at the output; this is expected since the diffractive decoder has additional trainable degrees of freedom compared to the free-space-based image decoding.”

“Fig. S7: Comparison of diffractive decoder and free-space decoder on Van Gogh style artwork generation.

Fig. S7: (a) Schematics of Design I trained with the diffractive decoder, and Design II trained directly with the free-space decoder (i.e., without an optimized diffractive layer). The optical generative model has 85M parameters and D_1 is set to 1cm. (b) Generated images of Design I with or without the diffractive decoder and Design II using the free-space decoder. As shown in the 3rd row, some lower contrast image results fail for the free-space-based decoding, with a CLIP score below 15 empirically considered as a failure. Under this criterion, the failure rate of

free-space-based decoding is 18.8%. Even among the remaining cases, the image generation quality using free-space-based decoding is inferior to that achieved with the jointly optimized diffractive decoder, which is expected due to the additional degrees of trainable parameters available at the diffractive decoder.”

Fig. S25’s conclusion regarding is also questionable. There is no guarantee that a reduced-size diffusion model will learn the same noise-to-image mapping as the larger teacher model. Since the smaller model is trained from scratch, rather than distilled from the full model in an end-to-end manner, it is unsurprising that it fails to match the output distribution of the original. To support the claims made in this section, quantitative metrics such as FID scores should be reported. In fact, based on visual inspection alone, the results from the reduced-size DDPM do not appear significantly degraded to me.

-- To address the referee’s comments, we have now included CLIP score comparisons as a measure of semantic alignment between the generated images and their corresponding text prompts, as shown in **Extended Data Fig. 8** (originally **Supplementary Fig. S25** – which has been renumbered to shorten the main text and comply with editorial requests). Because there are only ~800 authenticated Van Gogh paintings available with limited digitization, the sample set is relatively small; as a result, computing FID score against this limited set provides unstable measurements, which we avoided.

Specifically, the CLIP scores for the multicolor optical generative model, the reduced-size diffusion model, and the pre-trained teacher diffusion model are 28.25, 24.45, and 28.72, respectively. **These results show that the optical generative model achieves a performance comparable to the pre-trained teacher diffusion model (which used 1000 steps) and outperforms the reduced-size diffusion model (which also used 1000 steps), indicating that the optically generated images exhibit strong alignment with the intended semantic content.**

Accordingly, we included these in the revised **Methods** and the captions of **Extended Data Figure 8**:

“...Furthermore, the CLIP Score evaluation suggests that the optically generated images exhibit strong alignment with the underlying semantic content.”

“...The average CLIP scores of the multicolor optical generative model, the reduced-size diffusion model, and the pre-trained teacher diffusion model are 28.25, 24.45, and 28.72, respectively (the reference text: “Van Gogh style painting of {architecture, plants, person}”).”

Finally, to focus our visual attention for better judgment, we copied below the last row of the generated images from Extended Data Figure 8, which clearly show the failures of image generation for the reduced-size diffusion model, despite using 1000 steps. See below, especially the failures around the features of the hat, jacket, arms and faces of the middle images:

Pretty similar to the last submission if I remember correctly. Provided checkpoint and test code but not training.

-- Thanks for pointing this out. We have included the training code within the current revised version.

References

1. B. Redding, A. Cerjan, X. Huang, M.L. Lee, A.D. Stone, M.A. Choma, & H. Cao, Low spatial coherence electrically pumped semiconductor laser for speckle-free full-field imaging, *Proc. Natl. Acad. Sci. U.S.A.* 112 (5) 1304-1309, <https://doi.org/10.1073/pnas.1419672112> (2015).
2. Akio Furukawa, Norihiro Ohse, Yoshifumi Sato, Daisuke Imanishi, Kazuya Wakabayashi, Satoshi Ito, Koshi Tamamura, Shoji Hirata, "Effective speckle reduction in laser projection displays," *Proc. SPIE* 6911, Emerging Liquid Crystal Technologies III, 69110T (29 January 2008); <https://doi.org/10.1117/12.760860>
3. J. J. Liu, C. Y. Lu, P. C. Pan, C. H. Chou, Y. C. Lee, L. H. Lai, L. W. Lai, Y. L. Ho, "1130nm wavelength vertical-cavity surface-emitting laser with single mode characteristics," *Proc. SPIE* 13384, Vertical-Cavity Surface-Emitting Lasers XXIX, 133840D (19 March 2025); <https://doi.org/10.1117/12.3039678>
4. Brian Chao, Manu Gopakumar, Suyeon Choi, Jonghyun Kim, Liang Shi, and Gordon Wetzstein. 2024. Large Étendue 3D Holographic Display with Content-adaptive Dynamic Fourier Modulation. In SIGGRAPH Asia 2024 Conference Papers (SA '24). Association for Computing Machinery, New York, NY, USA, Article 26, 1–12. <https://doi.org/10.1145/3680528.3687600>
5. Akram, Muhammad & Chen, Xuyuan. (2015). Speckle reduction methods in laser-based picture projectors. *Optical Review*. 23. 10.1007/s10043-015-0158-6.

We sincerely thank the referees for their reviews and the constructive feedback that we have received on our manuscript “***Optical Generative Models***” submitted to *Nature* (Manuscript ID: **2024-09-20762B**).

Following the editorial requests, we have shortened our main text to less than 4000 words to comply with the length requirements. We have also shortened our Methods section.

The original referee comments are shown in black color, whereas for ease of communication, our answers are provided in blue.

Referee 1

I appreciate and thank the authors for the extensive supplementary work they have done, and I believe the work is worthy of publication. However, I hope the authors can pay attention to their wording and remain objective. The CLIP score obtained in the experiment is significantly lower than that of the numerical calculation and the lightweight numerical calculation, rather than demonstrating "high alignment".

-- We sincerely thank the reviewer for the constructive and thorough feedback that we have received over the last 3 revision cycles that significantly improved our manuscript. We have accordingly toned down our claims to remain objective and eliminated the use of “high alignment” or other potentially subjective wording in our shortened manuscript.

Referee 3

Thanks to the authors for the clarifications. The reduction in FLOPs makes sense, it is intuitive that using a FC structure leads to a more compact generative neural network with fewer parameters, and that the optical implementation reduces the computational overhead of the FC layers. The methodology now presents a compelling story to me. (though I imagine the training process must be very costly given the FLOPs argument?)

-- We sincerely thank the reviewer for the constructive and thorough feedback that we have received over the last 3 revision cycles that significantly improved our manuscript.

Regarding the training process of our optical generative models, some of these details, including the training times for each model, are provided in **Supplementary Sec. 3** of our manuscript:

“...The training was conducted on a server configured with an AMD Threadripper 3990X CPU, 128 GB of G.Skill DDR4 RAM (16 GB × 8), and four NVIDIA GeForce RTX 4090 GPUs. For training the teacher denoising diffusion probabilistic model (DDPM), we used a batch size of 200 and trained it for 300 epochs. The training time, which is a one-time effort, in GPU hours for each dataset is as follows: MNIST – 33 hours, Fashion-MNIST – 52 hours, Butterfly-100 – 48 hours, and CelebA – 58 hours. For fine-tuning the stable diffusion for artwork generation, we used a batch size of 8 and trained for 20 epochs, which took ~61 GPU hours. With more advanced GPUs, such as the NVIDIA A100, the training speed would be 3–8 times faster. This implies, using a state-of-the-art GPU, a reduction in training time for the teacher DDPM to approximately 6, 10, 9, and 11 GPU hours for MNIST, Fashion-MNIST, Butterfly-100, and CelebA, respectively.

For training the optical generative model, we used a batch size of 100 and trained for 100 epochs. In each iteration, we sampled a batch of random noise and fed it into the teacher model to obtain the data pair for training the optical generative model. We used 50 equally spaced steps of sampling to

obtain the target distribution without significantly affecting the quality of the image generation. The training time of the optical generative model in GPU hours for each dataset is as follows: MNIST – 38 hours, Fashion-MNIST – 58 hours, Butterfly-100 – 59 hours, and CelebA – 70 hours. For training the optical generative model in artwork generation, we used a batch size of 4 and trained for 100 epochs (with 2500 iterations each epoch), which took approximately 82 GPU hours. Because the model we finetuned from stable diffusion shares the same latent representation for monochrome and colorful image generation, the training time consumption is similar for these two tasks. Once again, with higher-end GPUs, such as the NVIDIA A100, the training speed would be 3–8 times faster. This implies a reduction in training time for the teacher DDPM to approximately 7, 11, 11, and 13 GPU hours for MNIST, Fashion-MNIST, Butterfly-100, and CelebA, respectively.

While training a deep learning model can be computationally intensive and time-consuming, it is generally a one-time process that occurs during the model development phase. Once the model is trained, it is fixed and reused across a wide range of deployment scenarios. In contrast, inference must be performed repeatedly—often in real-time or at large scale—during practical deployment. As a result, inference efficiency becomes much more critical for real-world utility, especially in applications requiring rapid response, low power consumption, or deployment on resource-constrained devices.”

The potential of the optical generative model as a platform for privacy-preserving and multiplexed visual information generation is also quite interesting, and I appreciate the authors’ efforts in conducting such experiments.

-- We appreciate the constructive and encouraging feedback of the referee on our privacy-preserving and multiplexed visual information generation results.

My only remaining question is about the evaluation metrics: why is the CLIP score used for the Van Gogh dataset, rather than standard metrics like FID or IS as used for other datasets? Also, was any comparison made against the teacher diffusion model in terms of FID or IS? As far as I know, CLIP was originally introduced for text-to-image generation, where a target text description is provided. However, in this paper all generative models are performing unconditional generation without text guidance. While I agree it is a valid demonstration that the generated images align with the intended semantic content, the use of CLIP score alone feels unusual to me. I wonder is there a specific reason for not using standard metrics like FID or IS for the Van Gogh dataset, as was done for MNIST and CelebA in Extended Data Figs. 2?

Overall, I believe this work is suitable for publication, provided that the above points are addressed. Finally, I appreciate the authors for releasing the training code.

-- We appreciate the reviewer’s thoughtful question regarding the evaluation metrics. Since there are only ~800 authenticated Van Gogh paintings available, computing IS or FID indicators against a limited data distribution is not meaningful or consistent with the literature, and it will be less stable, potentially leading to misleading conclusions. In our manuscript, various evaluations on the Van Gogh-style artwork generation are presented in, for example, **Supplementary Figs. S13-S14**, where the PSNR and the CLIP scores are reported to demonstrate consistency at both the pixel level and semantic level. These points are included in our revised Methods, at the end of the “Performance analyses and comparisons” sub-section: “...Additional evaluations on the Van Gogh-style artwork generation are presented in Supplementary Figs. S13-S14, where the PSNR and the CLIP scores are reported to demonstrate consistency at both the pixel level and semantic level. Since there are only ~800 authenticated Van Gogh paintings available, computing IS or FID indicators against a limited data distribution is not meaningful and will be less stable.”